# Examining the role of common variants in rare neurodevelopmental conditions

Qin Qin Huang[1,14], Emilie M. Wigdor[1,14], Daniel S. Malawsky[1], Patrick Campbell[1,2], Kaitlin E. Samocha[3,4], V. Kartik Chundru[1,5], Petr Danecek[1], Sarah Lindsay[1], Thomas Marchant[1], Mahmoud Koko[1], Sana Amanat[1], Davide Bonfanti[1], Eamonn Sheridan[1,6,7], Elizabeth J. Radford[1,8], Jeffrey C. Barrett[1], Caroline F. Wright[5], Helen V. Firth[1,9], Varun Warrier[10,11], Alexander Strudwick Young[12,13], Matthew E. Hurles[1] & Hilary C. Martin[1✉]

Although rare neurodevelopmental conditions have a large Mendelian component[1], common genetic variants also contribute to risk[2,3]. However, little is known about how this polygenic risk is distributed among patients with these conditions and their parents nor its interplay with rare variants. It is also unclear whether polygenic background affects risk directly through alleles transmitted from parents to children, or whether indirect genetic effects mediated through the family environment[4] also play a role. Here we addressed these questions using genetic data from 11,573 patients with rare neurodevelopmental conditions, 9,128 of their parents and 26,869 controls. Common variants explained around 10% of variance in risk. Patients with a monogenic diagnosis had significantly less polygenic risk than those without, supporting a liability threshold model[5]. A polygenic score for neurodevelopmental conditions showed only a direct genetic effect. By contrast, polygenic scores for educational attainment and cognitive performance showed no direct genetic effect, but the non-transmitted alleles in the parents were correlated with the child's risk, potentially due to indirect genetic effects and/or parental assortment for these traits[4]. Indeed, as expected under parental assortment, we show that common variant predisposition for neurodevelopmental conditions is correlated with the rare variant component of risk. These findings indicate that future studies should investigate the possible role and nature of indirect genetic effects on rare neurodevelopmental conditions, and consider the contribution of common and rare variants simultaneously when studying cognition-related phenotypes.

Rare conditions affect 3.5–6% of the global population[6] and of these, most involve the central nervous system[7]. Whereas genomic sequencing has revolutionized the diagnosis of rare neurodevelopmental conditions, which typically include intellectual disability and/or developmental delay, a monogenic diagnosis is only identified for about 30–40% of patients[1,8]. Common variants also contribute to risk for rare neurodevelopmental conditions[2,3]. In particular, this common variant contribution overlaps with polygenic risk for schizophrenia and for predisposition to reduced educational attainment and cognitive performance[2]. Accordingly, rare damaging variants in constrained genes, which play a major role in risk of rare neurodevelopmental conditions, are also associated with increased risk of mental health conditions and reduced educational attainment and cognitive performance in UK Biobank[9–11]. In this work, we seek to address three fundamental questions (Extended Data Fig. 1). First, we aim to better understand the nature of common variant risk for

rare neurodevelopmental conditions, particularly its overlap with common variant risk for mental health and cognitive phenotypes. Second, we aim to explore the interplay between common and rare variants in the context of these conditions. Third, we aim to test whether there is an effect of common variants in the parents on their child's risk of these conditions, above and beyond the child's own genetics.

We begin by leveraging new, larger genome-wide association studies (GWASs) than were previously available[2] to explore the extent to which common variant effects on rare neurodevelopmental conditions are correlated with their effects on a broad range of mental health conditions. This is motivated by findings that some psychiatric conditions have a partial neurodevelopmental origin[12–14], and that people with rare neurodevelopmental conditions[15], as well as their relatives[16,17], are more likely to have psychiatric conditions. Some of this overlap seems to be driven by certain rare copy number variants with variable

[1]Wellcome Sanger Institute, Hinxton, UK. [2]Department of Medical and Molecular Genetics, King's College London, London, UK. [3]Center for Genomic Medicine, Massachusetts General Hospital, Boston, MA, USA. [4]Broad Institute of MIT and Harvard, Cambridge, MA, USA. [5]Institute of Biomedical and Clinical Science, University of Exeter, Exeter, UK. [6]Leeds Institute of Medical Research, University of Leeds, St. James's University Hospital, Leeds, UK. [7]Yorkshire Regional Genetics Service, Chapel Allerton Hospital, Leeds, UK. [8]Department of Paediatrics, University of Cambridge, Cambridge Biomedical Campus, Cambridge, UK. [9]Cambridge University Hospitals Foundation Trust, Addenbrooke's Hospital, Cambridge, UK. [10]Department of Psychiatry, University of Cambridge, Cambridge, UK. [11]Department of Psychology, University of Cambridge, Cambridge, UK. [12]University of California Los Angeles Anderson School of Management, Los Angeles, CA, USA. [13]Human Genetics Department, UCLA David Geffen School of Medicine, Los Angeles, CA, USA. [14]These authors contributed equally: Qin Qin Huang, Emilie M. Wigdor. ✉e-mail: hcm@sanger.ac.uk

expressivity[18,19], suggesting some shared aetiology between psychiatric and rare neurodevelopmental conditions. Here, to address our first aim, we explore whether shared common variant effects may also contribute to the overlap between these conditions, and whether this is independent of the genetic overlap between these conditions and cognitive traits.

Little is known about the interplay between rare and common variants in the context of rare neurodevelopmental conditions, and dissecting this will be key to fully understanding their genetic architecture and improving genetic diagnosis and risk prediction. As the second aim of our study, we set out to address two hypotheses in this space, testing the liability threshold model and whether common variants modify the penetrance of inherited rare variants. The liability threshold model predicts that an individual will develop a condition once the sum of independent genetic and environmental risk factors exceeds some threshold[5,20]. Under this model, one might expect that patients with neurodevelopmental conditions who have a highly penetrant damaging variant (constituting a monogenic diagnosis) would require, on average, less polygenic load to cross a diagnostic threshold than those without such variants (Extended Data Fig. 2a). We previously saw no significant difference in polygenic scores between patients with versus without a monogenic diagnosis[2], but in this work, we anticipated that increased sample size and improved diagnostic rate[1,21] might improve power to detect a difference. As rare variants associated with neurodevelopmental conditions seem to act additively with polygenic scores in affecting cognitive ability in UK Biobank[10,11], we hypothesized that polygenic background would modify the penetrance of these inherited rare variants in families with neurodevelopmental conditions, as it does, for example, in the context of *BRCA1/2* variants predisposing to breast cancer[22].

Finally, as our third aim, we explore whether common variants predisposing to rare neurodevelopmental conditions act directly on the affected individuals carrying them ('direct genetic effects'). Many studies have shown that genetic associations between common genetic variants and cognition-related phenotypes estimated in population-based samples shrink when estimated within families[4,23–26]. One possible explanation for this is that variants associated with these traits have indirect genetic effects, that is, they have some effect on the parents, and this then affects the offspring through the family or prenatal environment[4,26–28]. However, confounding factors may also contribute to population-based genetic effect estimates[4,29,30]. Studies of rare diseases have typically assumed implicitly that variants affecting risk have direct genetic effects on the affected individual. Given the genetic overlap with educational attainment and cognition, we hypothesized that the common variants associated with risk of rare neurodevelopmental conditions might not only reflect direct genetic effects.

We address these questions using two large UK-based cohorts of individuals with rare neurodevelopmental conditions, the Deciphering Developmental Disorders (DDD) study ($N = 7,955$ patients with genotype array and exome sequence data) and the Genomics England 100,000 Genomes project (GEL; $N = 3,618$ patients with genome sequence data), combined with several control cohorts (Supplementary Table 1). We have included a Frequently Asked Questions document in less technical language to explain the study, and to address some possible misunderstandings (Supplementary Note 1).

## GWAS and genetic correlations

We first sought to validate the role of common genetic variation in neurodevelopmental conditions by replicating the key findings from our previous work in DDD in a large independent cohort. We identified a subset of GEL rare disease families with neurodevelopmental conditions and removed families overlapping with the DDD study (Methods). Almost all probands with neurodevelopmental conditions in GEL (97%) had intellectual disability or global developmental delay, versus 88% of those in DDD. The cohorts were broadly phenotypically similar (Extended Data Fig. 3 and Supplementary Note 2).

When comparing 3,618 unrelated patients with neurodevelopmental conditions to 13,667 unrelated controls within GEL, polygenic scores for educational attainment ($PGS_{EA}$)[31], cognitive performance ($PGS_{CP}$)[31] and schizophrenia ($PGS_{SCZ}$)[32] each explained a significant but small amount of variance on the liability scale ($R^2 < 1\%$; logistic regression $P < 3.9 \times 10^{-4}$). This was similar to that observed when comparing 6,397 unrelated patients from DDD with 9,270 independent unrelated controls (Supplementary Table 2). The polygenic score for neurodevelopmental conditions derived from our GWAS in DDD[2] ($PGS_{NDC,DDD}$) was also associated with neurodevelopmental conditions within GEL ($P = 1.1 \times 10^{-6}$, $R^2 = 0.11\%$; Supplementary Table 2).

These results indicated that the polygenic contribution to rare neurodevelopmental conditions was similar between these two cohorts. Thus, to increase power to study common variant effects on these conditions, we conducted a GWAS in GEL, then meta-analysed the results with the DDD GWAS (Extended Data Fig. 4 and Supplementary Data 1–3). This meta-analysis revealed two genome-wide significant loci (Supplementary Note 3). Variants at one of these loci are associated with cognitive traits[31,33]. The fraction of phenotypic variance explained by genome-wide common variants (that is, the single-nucleotide polymorphism (SNP) heritability on the liability scale assuming a population prevalence of 1%) was estimated at 11.2% (8.5–13.8%) (Supplementary Table 3).

In pursuit of our first main aim, to test for possible shared genetic contributors between rare neurodevelopmental conditions and other brain-related traits and conditions, we calculated genetic correlations ($r_g$) between them using our own and published GWAS meta-analyses. We observed the expected negative genetic correlations between neurodevelopmental conditions and educational attainment[31] ($r_g = -0.65$ (−0.84, −0.47), $P = 4.9 \times 10^{-12}$) and cognitive performance[31] ($r_g = -0.56$ (−0.73, −0.39), $P = 1.6 \times 10^{-10}$), stronger in magnitude than those observed with the DDD GWAS alone, and a positive genetic correlation with schizophrenia[32] ($r_g = 0.27$ (0.13, 0.40), $P = 9.7 \times 10^{-5}$) (Fig. 1a and Supplementary Table 4). Furthermore, we detected significant genetic correlations ($P < 0.0038 = 0.05/13$; Bonferroni correction for 13 traits) with several other mental health and neurodevelopmental conditions including attention-deficit hyperactive disorder (ADHD)[34] ($r_g = 0.46$ (0.28, 0.64), $P = 5.2 \times 10^{-7}$), and with the 'non-cognitive component of educational attainment' derived from GWAS-by-subtraction (Non-CogEA)[35] ($r_g = -0.37$ (−0.52, −0.22), $P = 1.2 \times 10^{-6}$) (Fig. 1a). We hypothesized that the genetic correlations with brain-related conditions could be explained at least in part by their relationship with educational attainment[35,36], given the strong negative genetic correlation between that and neurodevelopmental conditions. To explore this, we estimated the genetic correlations conditioning on the educational attainment GWAS summary statistics (Fig. 1b). The genetic correlations with ADHD and depression were no longer significant after conditioning on educational attainment, whereas those with schizophrenia and Tourette's syndrome remained significant. The latent genetic component of neurodevelopmental conditions that was correlated with educational attainment explained 77% of the genetic correlation with ADHD, the highest among all tested conditions (Supplementary Fig. 1 and Supplementary Methods). These results confirmed that common variants collectively associate with rare neurodevelopmental conditions in two independent cohorts, and that these common variant effects are shared with other brain-related conditions and cognitive traits.

Below, we explore the extent and nature of the contribution of polygenic background to neurodevelopmental condition risk using $PGS_{NDC,DDD}$ and polygenic scores for the most significantly genetically correlated traits ($PGS_{EA}$, $PGS_{CP}$, $PGS_{NonCogEA}$, $PGS_{SCZ}$) from much larger published GWASs. Several of these polygenic scores are significantly correlated with each other (Supplementary Fig. 2), thus our correction for multiples of five tests is conservative. Below, we often use the term 'more polygenic risk' for neurodevelopmental conditions as a shorthand for having higher $PGS_{NDC,DDD}$ and/or $PGS_{SCZ}$, and/or lower $PGS_{EA}$, $PGS_{CP}$ and/or $PGS_{NonCogEA}$.

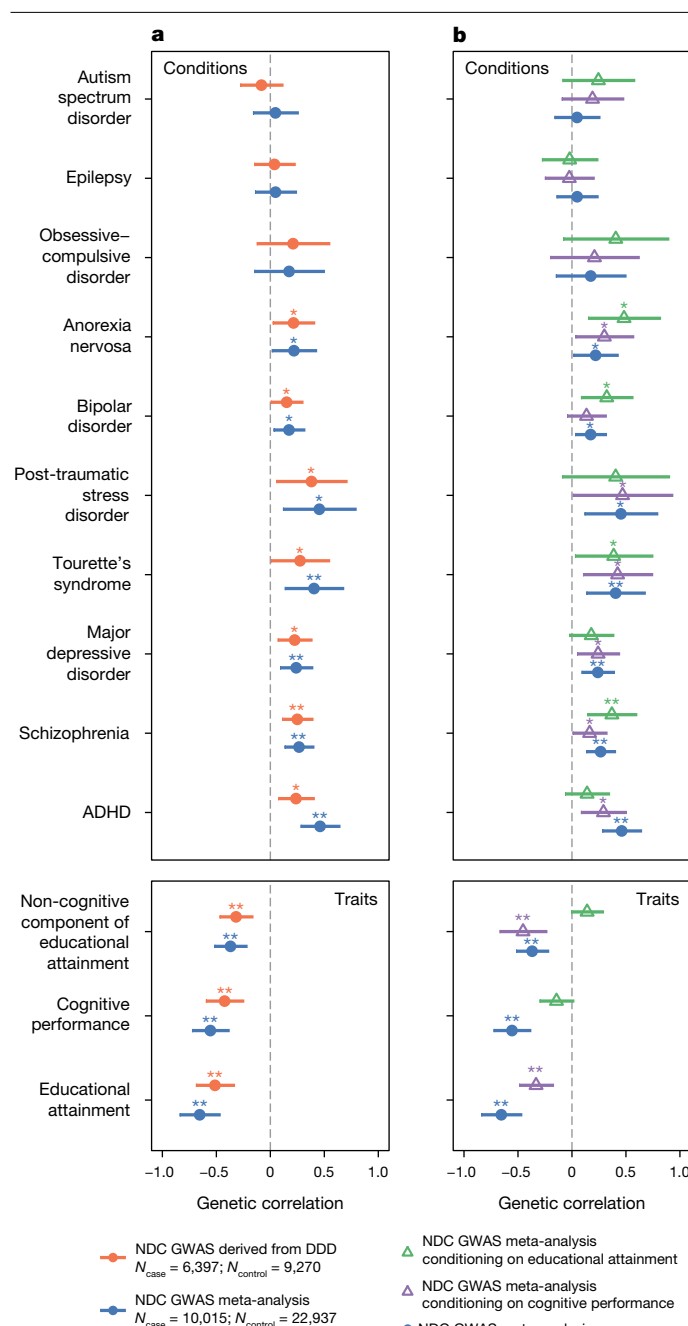

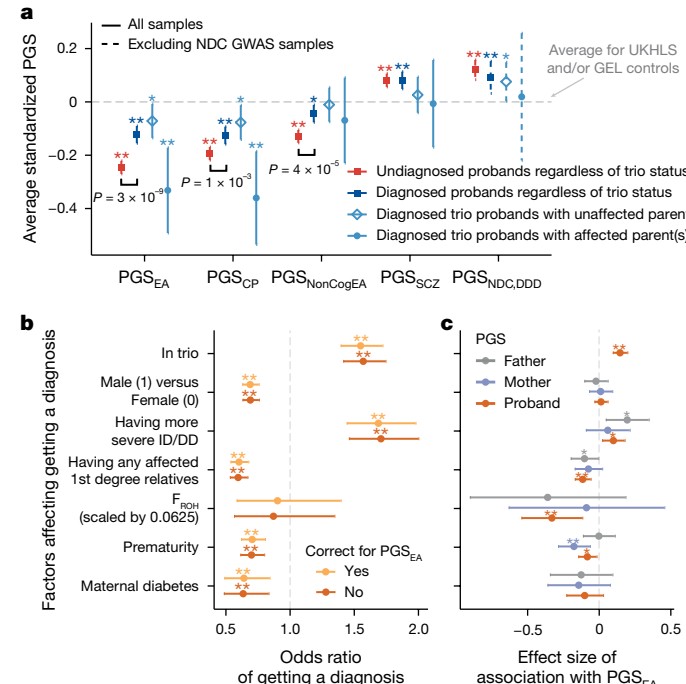

**Fig. 1 | Genetic correlations between neurodevelopmental conditions and other brain-related traits and conditions. a**, Points show the estimates from linkage disequilibrium score regression for the DDD GWAS (orange) and the meta-analysis of neurodevelopmental conditions (NDCs) between DDD and GEL (blue). **b**, Points show the estimates for the meta-analysis after conditioning on the GWAS summary statistics for educational attainment (green) or cognitive performance (purple) using GenomicSEM. Error bars show 95% confidence intervals. One asterisk indicates nominally significant results (*$P < 0.05$) and a double asterisk indicates significant results that passed the Bonferroni correction for 13 traits and conditions (**$P < 0.0038$). Exact estimates and two-sided $P$ values are reported in Supplementary Table 4.

## Less polygenic risk in diagnosed probands

Thirty-six percent of patients in these cohorts have a molecular monogenic diagnosis, including de novo, recessive, X-linked or inherited dominant diagnoses that involve rare (or novel) variants[1]. To address our second aim of investigating the interplay between common and rare genetic variants in these conditions, we tested whether these diagnosed patients differed from undiagnosed patients in terms of their

**Fig. 2 | Disentangling polygenic score associations with diagnostic status. a**, Average polygenic scores (PGSs) in probands with ('diagnosed'; $N = 3,821$; dark blue) versus without ('undiagnosed'; $N = 6,345$; red) a monogenic diagnosis, from DDD and GEL combined. Diagnosed probands from trios split by parental affectedness are in light blue. The scores have been standardized such that the controls have mean 0 and variance 1. Subgroups that have significantly different average polygenic score from controls (dashed line) are indicated by an asterisk (*$P < 0.05$) or double asterisk (**$P < 0.01$ after Bonferroni correction for five polygenic scores). Significant differences between diagnosed (dark blue) and undiagnosed (red) patients are annotated with $P$ values. See Supplementary Table 5 for results of two-sided $t$-tests comparing the various groups. UKHLS, UK Household Longitudinal Study. **b**, Associations between various factors and diagnostic status within the full DDD cohort[1], with or without correcting for the proband's $PGS_{EA}$ ($N = 7,549$), calculated within probands of GBR ancestry (individuals with genetic similarity to British individuals from the 1,000 Genomes Project) using logistic regression. An odds ratio (shown in points) greater than one indicates that that factor is associated with a higher chance of receiving a monogenic diagnosis. $F_{ROH}$, the fraction of the genome in runs of homozygosity; ID/DD, intellectual disability or developmental delay. **c**, Associations between these factors and DDD probands' ($N = 7,549$), mothers' or fathers' $PGS_{EA}$ ($N = 2,497$). Points show effect sizes assessed by linear regression. A double asterisk indicates that the association passed Bonferroni correction for seven factors (see Supplementary Table 8 for exact $P$ values). The expected value of $F_{ROH}$ is 0.0625 for individuals whose parents are first cousins. $P$ values in all panels are two-sided. Error bars show 95% confidence intervals.

polygenic risk. Consistent with the liability threshold model (Extended Data Fig. 2a), we observed significantly higher $PGS_{EA}$ (DDD and GEL combined; average difference $\Delta = 0.12$ standard deviations (s.d.), two-sided $t$-test $P = 3.0 \times 10^{-9}$), $PGS_{CP}$ ($\Delta = 0.068$ s.d., $P = 1.2 \times 10^{-3}$) and $PGS_{NonCogEA}$ ($\Delta = 0.085$ s.d., $P = 3.7 \times 10^{-5}$) in probands with versus without a monogenic diagnosis (Fig. 2a). Despite this, we observed that for all scores except for $PGS_{NonCogEA}$, the diagnosed probands still had significantly more polygenic risk than the controls ($P < 0.01 = 0.05/5$; Fig. 2a and Supplementary Table 5). Sensitivity analyses suggest that this observation is not driven by ascertainment bias in the controls, although the effect size is sensitive to the choice of control cohort, particularly for $PGS_{EA}$ (Supplementary Note 4 and Supplementary Table 6). To mitigate this, we developed a set of statistical weights adjusting for sampling and non-response bias in the Millenium Cohort Study (MCS), an extra control cohort, to calculate weighted average polygenic scores that should be representative of the full UK population (Supplementary Note 4

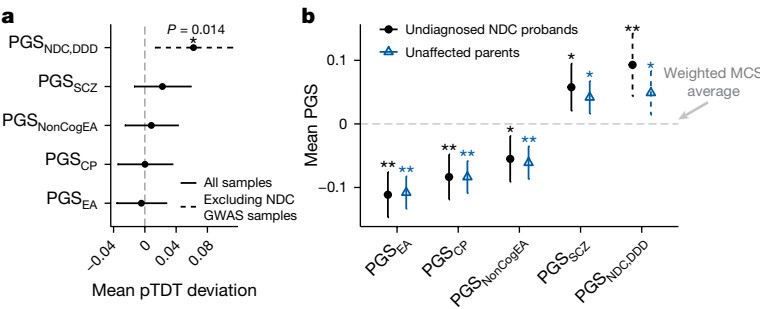

**Fig. 3 | Polygenic background in parents of patients with neurodevelopmental conditions. a**, pTDT in undiagnosed probands with unaffected parents. Plotted is the mean pTDT deviation (difference between the child's polygenic score and the mean parental score, in units of the s.d. of the latter) in trios from GEL and DDD ($N = 2,866$, or $N = 1,567$ for testing $PGS_{NDC,DDD}$). We tested whether this is significantly different from 0 using two-sided one-sample $t$-tests. **b**, Mean polygenic scores for undiagnosed probands or

their unaffected parents in the trios used in the pTDT analysis, standardized using the weighted MCS controls whose mean is indicated by the dotted line. See Supplementary Tables 9 and 7 for results of pTDT and two-sided $t$-tests, respectively. Subgroups that have a significantly different average polygenic score from controls are indicated by an asterisk (*$P < 0.05$) or double asterisk (**$P < 0.01$ after Bonferroni correction for five polygenic scores). Error bars in both plots show 95% confidence intervals.

and Extended Data Figs. 5 and 6). Both undiagnosed and diagnosed probands had a significantly lower average $PGS_{EA}$ than weighted MCS controls (0.17 and 0.049 s.d., respectively; Supplementary Table 7).

The difference between the diagnosed probands and controls is driven by those with affected parents (those reported by clinicians to show a similar phenotype to their child), who had significantly lower polygenic scores for educational attainment and cognitive performance than those with unaffected parents (for example, $PGS_{EA} \Delta = 0.26$ s.d., $P = 3.4 \times 10^{-3}$) (light blue points and diamonds in Extended Data Fig. 5). Diagnosed probands with unaffected parents did not show significantly different polygenic scores from the weighted MCS controls.

We next explored whether the difference in polygenic risk between diagnosed and undiagnosed probands was related to various technical, clinical and prenatal factors that are associated with receiving a monogenic diagnosis in DDD[1]. For example, diagnosed probands were more likely than undiagnosed to be in a trio (probably due to the ability to distinguish de novo from inherited variants) and to have severe intellectual disability, and less likely to have been born prematurely (a known risk factor for neurodevelopmental conditions[37,38]) (Fig. 2b and Supplementary Table 8). We hypothesized that some of these associations might be confounding, or be confounded by the association between $PGS_{EA}$ and diagnostic status, as, for example, single-parent households and premature birth are associated with higher levels of deprivation and/or lower parental educational attainment[39]. Indeed, we observed that the probands' $PGS_{EA}$ was significantly associated with several of these factors (Fig. 2c): a higher chance of being in a trio and having more severe intellectual disability, and a lower chance of being born prematurely and having any affected first-degree relatives (Extended Data Fig. 7a). However, it was not associated with sex (Supplementary Note 5 and Extended Data Fig. 8a) or maternal diabetes (Fig. 2c and Supplementary Table 8). Controlling for $PGS_{EA}$ minimally altered the association between these factors and diagnostic status (Fig. 2b). Similarly, after controlling for these factors, the association between $PGS_{EA}$ and diagnostic status remained significant with negligible change in effect size (Extended Data Fig. 7b). Thus, the observation that diagnosed patients tend to have lower polygenic risk than undiagnosed probably largely reflects the liability threshold model under which both common and rare variants contribute to risk (Extended Data Fig. 2a).

## Assessing transmission of polygenic risk

Most of the parents in our sample are reported by clinicians to be clinically unaffected (89.2% in DDD and 95.4% in GEL, although the clinical annotation of parental affected status may be imperfect). Given this, and results in autism[40], we hypothesized that probands without monogenic diagnoses might inherit more common variant risk for

neurodevelopmental conditions from unaffected parents than one would expect given their parents' mean risk, a phenomenon termed 'polygenic transmission disequilibrium'[40]. Applying the polygenic transmission disequilibrium test (pTDT)[40] to undiagnosed trios with unaffected parents (Fig. 3a), we saw nominally significant over-transmission of $PGS_{NDC,DDD}$ in 1,567 families not included in the original GWAS (pTDT deviation 0.062, paired $t$-test $P = 0.014$). This over-transmission was significant in females (pTDT deviation 0.10, $P = 0.0078$ in 589 trios) but not in males (pTDT deviation 0.036, $P = 0.27$ in 978 trios) (Extended Data Fig. 8c and Supplementary Note 5). However, we saw no significant transmission disequilibrium for the other polygenic scores (paired $t$-test $P > 0.05$) in either sex (Extended Data Fig. 8c) or in both sexes combined (Fig. 3a). Given the known over-transmission of $PGS_{EA}$ to autistic individuals[40], we excluded autistic individuals from our sample and repeated the pTDT, but still only saw significant transmission disequilibrium for $PGS_{NDC,DDD}$ (Supplementary Fig. 3a). Among probands with a monogenic genetic diagnosis, we saw no significant transmission disequilibrium for any polygenic score tested (Supplementary Fig. 3b).

To put the pTDT results in context, we compared average polygenic scores between unaffected parents of undiagnosed patients and controls. For all five scores tested, the parents had more polygenic risk than the weighted MCS controls ($P < 0.026$) (Fig. 3b and Supplementary Table 7). Given this observation and the results from the pTDT, we conclude that risk for neurodevelopmental conditions is affected both by familial polygenic background, or factors correlated with it, and by polygenic risk (specifically, $PGS_{NDC,DDD}$) that is over-transmitted from unaffected parents to affected children.

## Association with non-transmitted alleles

Given these findings, and to address our third aim, we next tested whether parental alleles are correlated with their child's risk of neurodevelopmental conditions independently of the alleles transmitted to the child: in other words, whether there is an effect of parental alleles that are not transmitted to the child ('non-transmitted alleles') on the child's phenotype. This could potentially be indicative of indirect genetic effects; that is, effects of alleles in parents on parental phenotypes that affect their offspring's risk through the family environment (otherwise known as 'genetic nurture'), as opposed to the direct genetic effects of alleles transmitted to the child. Indirect genetic effects have been argued to explain around 30–45% of the association between polygenic predictors of educational attainment and school grades[26,30] and educational attainment[4], although these inferences have been contested as confounded by parental assortment and population stratification[29,30]. To investigate the possible role of non-transmitted parental alleles in risk of neurodevelopmental conditions, we compared 2,866

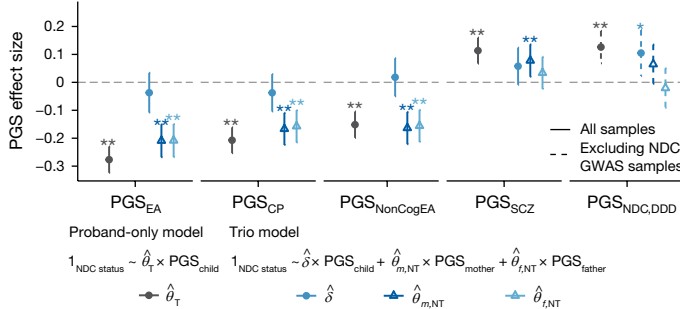

**Fig. 4 | Assessing direct genetic effects and associations with non-transmitted parental alleles.** The plot shows effect sizes of polygenic scores on case status, testing either the child's polygenic score alone (proband-only model) among trio probands or while also controlling for the parents' scores (trio model). Logistic regression models (annotated in the figure) were fitted to compare undiagnosed probands with neurodevelopmental conditions from 2,866 trios from DDD + GEL (or 1,567 trios for testing $PGS_{NDC,DDD}$) in which parents are unaffected with 4,804 control trios. $1_{NDC\,status}$ is an indicator for whether the proband has a neurodevelopmental condition (1) or is a control (0). In the trio model, the coefficients on the parental polygenic scores are referred to as the non-transmitted coefficients ($\hat{\theta}_{m,NT}$ and $\hat{\theta}_{f,NT}$), whereas the coefficient on the child's score is called the direct effect ($\hat{\delta}$). Error bars indicate 95% confidence intervals. One asterisk indicates nominally significant results (*$P < 0.05$) and double asterisk (**$P < 0.01$) indicates significant results that passed the Bonferroni correction for five polygenic scores. See Supplementary Table 10 for two-sided $P$ values.

affected trio probands from DDD + GEL whose parents are unaffected with 4,804 control trios from two UK birth cohorts ($N = 3,932$ trios) and from GEL ($N = 872$ trios without neurodevelopmental conditions). We first tested whether the child's polygenic scores for traits related to neurodevelopmental conditions were significantly associated with case status ('proband-only' model), and then whether this held after conditioning on the parents' polygenic scores ('trio model') (Fig. 4). The trio model removes the environmentally mediated portion of polygenic risk in the parents from the direct genetic effects of alleles transmitted to their children. We refer to the coefficients on the parental scores in the trio model as the 'non-transmitted coefficients' as they represent the association with non-transmitted parental alleles[24]. For more explanation and formal mathematical definition of this model, see the Methods section on 'Association with non-transmitted alleles' and the legend of Fig. 4.

For $PGS_{EA}$, $PGS_{CP}$ and $PGS_{NonCogEA}$, we found that undiagnosed probands' polygenic scores were no longer significantly associated with having a neurodevelopmental condition after conditioning on their parents' scores in the trio model. This implies limited or no direct genetic effects, whereas the non-transmitted coefficients were highly significant (Fig. 4 and Supplementary Table 10). This result held for $PGS_{EA}$ and $PGS_{NonCogEA}$ in sensitivity analyses of subsets of trios; $PGS_{CP}$ showed more equivocal results but the estimate of direct genetic effects was never significantly different from zero (Supplementary Fig. 4). We also observed a significant non-transmitted coefficient in the mother when using a polygenic score derived from a within-family GWAS for educational attainment[25] (Supplementary Note 6). This finding could imply that there are aspects of the environment—including the prenatal environment—that are correlated with these non-transmitted alleles and that affect risk of neurodevelopmental conditions, including genetically influenced parental phenotypes. However, our observations could also be due to the effects of parental assortment (that is, phenotypic correlation between partners), which we discuss further below.

For $PGS_{NDC,DDD}$, we found that the probands' polygenic scores were still nominally significantly associated with having a neurodevelopmental condition after controlling for their parents' scores in the trio model (Fig. 4). This implies that there is a direct genetic effect of $PGS_{NDC,DDD}$

on the probands' risk of neurodevelopmental conditions, consistent with the over-transmission observed in Fig. 3a. For $PGS_{SCZ}$, we saw no significant effect of the probands' score ($P = 0.089$) in the trio model, whereas the mothers' score was significant ($P = 8.6 \times 10^{-3}$) (Fig. 4). In summary, there is evidence for direct genetic effects of the polygenic score for rare neurodevelopmental conditions, but not for polygenic scores for related traits.

## Exploring the role of prenatal factors

We explored whether prenatal factors might mediate the effects of non-transmitted parental alleles on risk of neurodevelopmental conditions (Supplementary Note 7). These included preterm delivery, smoking, alcohol use, gestational hypertension and sleep apnoea. Among them, preterm delivery (that is, giving birth prematurely)[41], a risk factor for neurodevelopmental conditions in the offspring[37,38], showed the strongest genetic correlation with neurodevelopmental conditions ($r_g = 0.58$ (0.18, 0.97), $P = 0.004$) (Extended Data Fig. 9a and Supplementary Table 11), and was significantly genetically correlated with lower educational attainment ($r_g = -0.30$ (−0.39, −0.21), $P = 2.3 \times 10^{-10}$), mirroring the epidemiological association[42]. Premature birth was also associated with lower $PGS_{EA}$ in DDD ($P = 0.0125$; Extended Data Fig. 9d). However, controlling for prematurity or removing premature probands did not significantly change the non-transmitted coefficients in the trio model (Supplementary Note 7 and Supplementary Fig. 5). Thus, there is no significant evidence at present that prematurity explains the association between neurodevelopmental conditions and non-transmitted common variants in the parents that are associated with educational attainment.

## Correlated common and rare variant risk

Another factor that may contribute to the significant correlation between non-transmitted alleles in parents and neurodevelopmental conditions in their children is parental assortment, the phenomenon whereby people are more likely to choose partners with similar traits to themselves. Parental assortment is particularly strong for educational attainment and cognitive ability[43]. It is also observed for psychiatric conditions[43–45], including in parents of autistic individuals and of individuals with neurodevelopmental conditions due to the 16p12.1 deletion[46]. Parental assortment induces a correlation between alleles that act in the same direction on a trait, both between parents and, in their descendants, within and between loci[47] (Extended Data Fig. 2b). Thus, parental assortment on cognitive ability or correlated traits (for example, educational attainment) would be expected to lead to individuals with inherited rare variants associated with reduced cognitive ability[9,10,48,49] also having a polygenic background of common variants associated with reduced cognitive ability[46,47]. In the proband-only model in Fig. 4, the proband's polygenic score would statistically capture ('tag') the correlated effects of these rare variants (which causally affect neurodevelopmental conditions[50]). However, in the trio model, the proband's polygenic score would no longer capture effects of the rare variant component after conditioning on the parents' scores (Extended Data Fig. 10a). Instead, this correlation with the rare variant component would be reflected by the non-transmitted coefficients on the parents' polygenic scores[29].

To explore this potential genetic consequence of parental assortment in our cohorts, we tested whether the common and rare variant components contributing risk of neurodevelopmental conditions are indeed correlated. From the sequencing data in DDD and GEL, we extracted rare (minor allele frequency (MAF) $<1 \times 10^{-4}$) protein-truncating variants (PTVs) and damaging missense variants in genes intolerant of loss-of-function variation ('constrained genes'), which are associated with reduced cognitive ability[10] and risk of neurodevelopmental conditions[49,50]. Consistent with the effects of parental assortment,

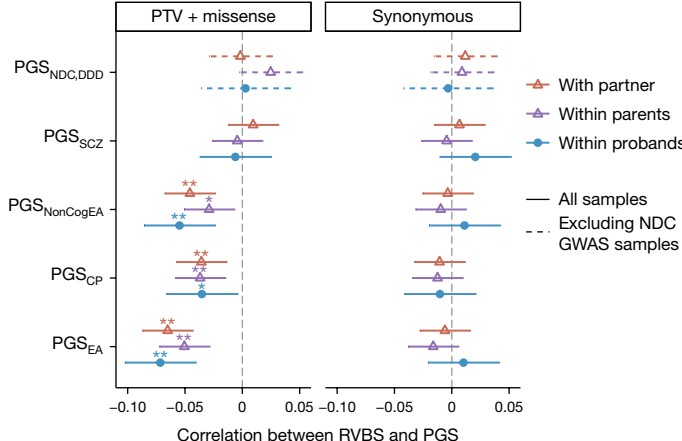

**Fig. 5 | Correlation between rare variant burden scores and polygenic scores.** Points represent Pearson correlation coefficients between the number of inherited rare damaging coding (left) or synonymous variants (right, negative control) in constrained genes and polygenic scores within or between different sets of individuals. In blue are the correlations within probands with neurodevelopmental conditions whose parents are unaffected (the child's rare variant burden score (RVBS) correlated with their own polygenic score), and in purple are the correlations within their parents. In orange is the cross-parental correlation (one parent's rare variant burden score correlated with the other parent's polygenic score). We calculated the correlations in trios with neurodevelopmental conditions from DDD and GEL ($N = 3,999$ or $2,553$ for $PGS_{NDC,DDD}$ excluding samples from the original GWAS[2]). Note that both the rare variant burden scores and polygenic scores have been corrected for 20 genetic principal components. Error bars represent 95% confidence intervals. One asterisk indicates nominally significant correlations (*$P < 0.05$) and the double asterisk indicates significant correlations that passed the Bonferroni correction for ten tests (five polygenic scores and two variant types) (**$P < 0.005$). See Supplementary Table 12 for exact estimates and two-sided $P$ values.

among unaffected parents of probands with neurodevelopmental conditions, we observed that the number of rare damaging coding variants in constrained genes (the 'rare variant burden score') in one parent was significantly negatively correlated with the other parent's $PGS_{EA}$ ($r = -0.065$, $P = 5.5 \times 10^{-9}$), $PGS_{CP}$ ($r = -0.036$, $P = 1.4 \times 10^{-3}$), and $PGS_{NonCogEA}$ ($r = -0.046$, $P = 4.3 \times 10^{-5}$) (orange points in Fig. 5 and Supplementary Table 12). As expected, a similar correlation was seen within the probands themselves, regardless of whether including all probands, undiagnosed probands or probands with de novo diagnoses (blue lines in Fig. 5 and Supplementary Fig. 6b,c, respectively), and if restricting rare variant burden score to haploinsufficient genes associated with developmental disorders (three leftmost columns in Supplementary Fig. 6). We also saw a similar result among control children from the MCS (pale blue points in Supplementary Fig. 7), indicating that this correlation is not only observed in patients with neurodevelopmental conditions. We saw no significant correlation between any of the polygenic scores and the burden of rare synonymous variants in tested gene sets (right-hand panel in Fig. 5, third and sixth columns in Supplementary Fig. 6), confirming that the result observed for deleterious variants is unlikely to be due to population structure artefacts. The correlations between polygenic scores and rare damaging variants may explain why we saw very limited evidence that these scores modify the penetrance of such variants in families with neurodevelopmental conditions (Supplementary Note 8 and Supplementary Fig. 8).

We next explored whether the correlation between common and rare variants associated with neurodevelopmental conditions could be driving the association between non-transmitted common alleles and children's risk shown in Fig. 4. We extended the trio model to control for the probands', mothers' and fathers' rare variant burden scores as well as polygenic scores when comparing trio probands with ($N = 1,343$) versus

without ($N = 872$) neurodevelopmental conditions in GEL (red boxes in Extended Data Fig. 10b). Correcting for rare variant burden scores did not change our original conclusion from the trio regression analysis of common variants. However, we cannot rule out that the association between neurodevelopmental conditions and non-transmitted common alleles is primarily driven by the assortment-induced correlation between common and rare variants, because the rare variant burden score we have used probably only captures a small proportion of the total rare variant component (just as the polygenic score only captures a small fraction of SNP heritability). Thus, further work and new datasets are needed to confirm whether the association between risk of neurodevelopmental conditions and the non-transmitted alleles is due to true indirect genetic effects and/or parental assortment.

## Discussion

Here we combined two large cohorts of patients with rare neurodevelopmental conditions to explore the contribution of common variants to risk. After first demonstrating that polygenic scores for neurodevelopmental conditions and several related traits were significantly associated with risk for neurodevelopmental conditions within both DDD and GEL (Supplementary Table 2), we conducted a GWAS meta-analysis of patients with neurodevelopmental conditions from the two cohorts and revealed significant genetic correlations with several psychiatric conditions that had not been previously reported[2] (Fig. 1a). Conditional genetic correlations show that these are only partially driven by the component of polygenic risk for neurodevelopmental conditions that is shared with educational attainment (for example, between 22% for Tourette's and 77% for ADHD; Supplementary Fig. 1). This suggests that these brain-related conditions share underlying biology with neurodevelopmental conditions that is partly independent of that captured by effects of common variants on educational attainment, although we acknowledge that estimates of genetic correlations can be biased by cross-trait parental assortment and other confounding factors[51]. Furthermore, although we observe a significant negative genetic correlation with what has been termed the non-cognitive component of educational attainment, we note that this could also contain elements of cognitive ability not captured in the GWAS for cognitive performance[31] that was used in the paper that derived it[35].

We showed that polygenic scores for several traits that are genetically correlated with neurodevelopmental conditions were significantly associated with having a monogenic diagnosis, with the strongest effect observed for educational attainment (Fig. 2a). Our previous work had found no such difference in polygenic background between diagnosed and undiagnosed probands in DDD[2], and it is likely that power has been improved here by our larger sample size and better definition of which probands truly have a monogenic diagnosis[1,21]. Our result is consistent with a liability threshold model for rare neurodevelopmental conditions, and consistent with recent findings in a population-based cohort, UK Biobank[11], and a rare disease cohort[52]. Children without a large-effect monogenic variant may require higher polygenic load (or a major environmental contribution such as a teratogenic infection, for example, Zika virus[53]) to move their phenotype over the threshold required to be clinically diagnosed with a neurodevelopmental condition (Extended Data Fig. 2a). Our findings suggest we can rule out a model whereby liability for neurodevelopmental conditions is conferred only by fully penetrant monogenic causes and environmental factors. Important for consideration in clinical settings, we find probands with more affected first-degree relatives had both a lower $PGS_{EA}$ (hence, more polygenic risk for neurodevelopmental conditions) and a lower chance of getting a monogenic diagnosis in DDD than probands with no affected relatives (Extended Data Fig. 7a). This emphasizes that if there are several first-degree relatives with neurodevelopmental conditions in a family, this may not necessarily be due to a monogenic cause. Our observation that diagnosed patients with affected parents (most of whom have

inherited dominant diagnoses), and their parents, have lower average PGS$_{EA}$ than those with unaffected parents (Extended Data Fig. 5) is consistent with the effects of parental assortment (Fig. 5).

As most parents of the patients we studied are annotated as clinically unaffected, we hypothesized that they might be over-transmitting polygenic risk to their affected offspring. We saw nominally significant over-transmission of PGS$_{NDC,DDD}$ from unaffected parents to undiagnosed probands, but saw no significant transmission disequilibrium for PGS$_{EA}$ or PGS$_{CP}$ (Fig. 3a), despite these polygenic scores explaining far more variance in risk than PGS$_{NDC,DDD}$ (Supplementary Table 2). Consistent with this, in a trio model (Fig. 4), we found evidence for a direct genetic effect of PGS$_{NDC,DDD}$ on risk of neurodevelopmental conditions, but not for other scores tested. Instead, we observed that the parents' PGS$_{EA}$, PGS$_{CP}$ and PGS$_{NonCogEA}$ were significantly associated with their children's risk even after controlling for the children's polygenic score, indicating a correlation between non-transmitted alleles and the children's phenotype. Thus, a key conclusion from this work is that the association between common variants and neurodevelopmental conditions is not entirely due to their having direct genetic effects on risk.

The correlation between non-transmitted alleles in the parents and neurodevelopmental conditions in the children may be due to indirect genetic effects, population stratification and/or the consequences of parental assortment[4,29,30,54]. Parental assortment induces a correlation between the polygenic score associated with the trait under assortment and the remaining genetic component of the phenotype. This includes the component due to rare variants, which could have a much stronger effect on risk of neurodevelopmental conditions than the common variant component. We demonstrated a correlation between the rare and common variant components that affect cognitive and educational outcomes, both between partners (one parent's rare variant burden score and the other parent's polygenic score), and within individuals (an individual's rare variant burden score and their own polygenic score, in both offspring and parents) (Fig. 5 and Supplementary Figs. 6 and 7). This supports the hypothesis that the association of PGS$_{EA}$ with lower risk of neurodevelopmental conditions is at least partly due to the assortment-induced correlation of PGS$_{EA}$ with rare variants affecting both neurodevelopmental conditions and educational attainment. Given that polygenic scores and our rare variant burden scores capture only small fractions of total common and rare variant components of risk, respectively, the actual correlation is substantially higher than the observed estimates. Very large whole-genome sequenced (WGS) datasets will be required to better characterize the total rare variant component of these traits and estimate this correlation more accurately.

With the current study design, we were unable to demonstrate the presence of indirect genetic effects on risk of neurodevelopmental conditions unambiguously, and nor could we test whether, if present, these are mediated by parenting behaviours. However, we did explore whether common genetic variants might influence risk by affecting prenatal risk factors (a form of indirect genetic effects). We found that educational attainment showed a significant negative genetic correlation with preterm delivery, whereas neurodevelopmental conditions showed a significant positive genetic correlation with it, of which only 35% was due to the educational attainment component (Extended Data Fig. 9b). This is consistent with epidemiological studies that found an association between prematurity and poorer cognitive outcomes even after controlling for socioeconomic confounders[37,55]. We saw no significant evidence that prematurity mediates the effects of non-transmitted common parental alleles associated with educational attainment (Supplementary Note 7). However, it may be that our analysis was simply underpowered at this sample size, as we did see some attenuation (albeit not significant) of the non-transmitted coefficients for PGS$_{EA}$ when removing premature probands (Supplementary Fig. 7). Nonetheless, our results emphasize how genetics may confound epidemiological associations between risk factors and neurodevelopmental

conditions[56,57], and also suggest that studies seeking to characterize indirect genetic effects on educational outcomes should consider the contribution of prenatal factors.

Our study has several limitations. First, the overall variance in risk of neurodevelopmental conditions explained by common variants is low (roughly 10%) and the polygenic scores tested here explain only a fraction of this. However, these polygenic scores are statistically significant predictors of neurodevelopmental conditions (Supplementary Table 2) and are likely to explain more variance as GWAS sample sizes grow. Second, the reported significance of detected polygenic score effects does not simply reflect the strength of the real associations, but also the power of the original GWAS from which SNP effect sizes were derived. Thus, one must be cautious when comparing effects between polygenic scores for different traits. We explored combining the different polygenic scores into a composite polygenic score to try to improve power; although this explained slightly more variance on the liability scale than PGS$_{EA}$ (Supplementary Table 2), results from the main analyses were very concordant between this composite polygenic score and PGS$_{EA}$ (which had the highest weight). Third, the phenotypic heterogeneity of the cohorts probably limits our power and may confound results. For example, missed diagnoses of autism among DDD and GEL participants with neurodevelopmental conditions (perhaps due to the young average age; Supplementary Note 2) could be confounding our result of there being no apparent under-transmission of PGS$_{EA}$ (Fig. 3a and Supplementary Fig. 3a), as PGS$_{EA}$ may be over-transmitted to autistic individuals[20,40] but under-transmitted to patients with intellectual disability who are not autistic. In future, larger cohorts with quantitative phenotype data (for example, on IQ or social responsiveness) may allow us to revisit these questions while subsetting to reduce phenotypic heterogeneity. Fourth, the fact that probands in trios tend to have higher PGS$_{EA}$ than those not in trios (Extended Data Fig. 6b) suggests that the trio probands are a non-random sample, which could potentially induce biases in trio-based analyses; for example, the undiagnosed trio probands may be enriched for monogenic causes in as-yet-undiscovered genes, which could reduce power when assessing over-transmission of polygenic risk (Fig. 3a). Furthermore, many of our analyses are predicated on the assumption that the 'unaffected parents' (those reported by the clinician not to have a similar phenotype to the proband) do not have phenotypes related to neurodevelopmental conditions. However, some of them may have (or may have had, earlier in life) relevant phenotypic features (for example, learning difficulty, speech delay) that were not detected and recorded by clinicians. The inclusion of these parents could be reducing power or confounding results in several analyses. Finally, the correlation between the rare and common variant components of neurodevelopmental conditions (Fig. 5), which is probably due to parental assortment, may have confounded several of these analyses.

In future, as GWAS discovery cohorts for both rare neurodevelopmental conditions and related traits increase in size, we will have more power to explore common variant effects on risk, penetrance and phenotypic expressivity of these conditions. These studies should seek to confirm whether there really are no direct genetic effects of common variants influencing educational attainment and cognitive performance on risk of neurodevelopmental conditions, or whether these are just small. To disentangle the contribution of indirect genetic effects and parental assortment to common variant associations with neurodevelopmental conditions, future studies will need to use extended genealogies and/or more sophisticated modelling of the influence of parental assortment on common and rare variants than is possible at present[29,30,54]. If these studies also had measures of epidemiological and prenatal risk factors such as prematurity, and of parental phenotypes and nurturing behaviours, one could explore how indirect genetic effects (if present) are mediated, which has potential implications for assessing the modifiability of risk. Larger

GWASs for neurodevelopmental conditions will also give us more power to explore the extent to which the common variants affecting these conditions are targeting different pathways and cell types from the rare variants (Supplementary Note 9). Finally, it will be important for future studies to explore the role of polygenic background in neurodevelopmental conditions in families with non-European genetic ancestries.

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

## Methods

### Cohort descriptions and phenotypes

**DDD.** The aim of the DDD study is to find molecular diagnoses for families and patients affected by previously genetically undiagnosed, severe developmental conditions. Recruitment was conducted from 2011 to 2015 across 24 clinical genetics services in the United Kingdom and Ireland[58]. The clinical inclusion criteria included neurodevelopmental conditions, congenital, growth or behavioural abnormalities and dysmorphic features. Probands were systematically phenotyped through DECIPHER[59] using Human Phenotype Ontology (HPO)[60] terms and a bespoke online questionnaire that collected information on developmental milestones, growth measurements, number of affected relatives, prematurity, maternal diabetes, and other clinically relevant parameters. The cohort has been described extensively[1,50,58,61].

We focused on probands in the DDD cohort who had neurodevelopmental conditions, which were defined previously by Niemi et al.[2] Briefly, these were probands who had at least one of the following neurodevelopmental HPO terms or their descendent terms: abnormality of higher mental function (HP:0011446), neurodevelopmental abnormality (HP:0012759), abnormality of the nervous system morphology (HP:0012639), behavioural abnormality (HP:0000708), seizures (HP:0001250), encephalopathy (HP:001298), abnormal synaptic transmission (HP:0012535), or abnormal nervous system electrophysiology (HP:0001311).

**GEL project.** The 100,000 Genomes project is an initiative by the UK Department of Health and Social Care to sequence the whole genomes of individuals with rare conditions or cancer in the National Health Service[62,63]. The rare disease branch of the project consists of sequencing data from roughly 72,000 patients with rare conditions and their relatives, in roughly 34,000 families with a variety of structures. There are more than 190 rare conditions represented in the cohort, and about 23% of the patients have neurodevelopmental conditions. The cohort was sequenced at around 35 times coverage, and variant calling and quality control (QC) were performed by Genomics England[63,64].

Patients from GEL with neurodevelopmental conditions were defined as those recruited under the 'Neurodevelopmental disorders' disease subcategory, or with more than one HPO term that was a descendant of 'Neurodevelopmental Abnormality' (HP:0012759). We removed probands whose age of onset was above 16 years or who had neurodegenerative conditions.

The set of unrelated GEL controls included patients with cancer above 30 years old ($N = 10,469$) and unaffected relatives ($N = 3,198$) of probands with rare conditions who were not in the neurodevelopmental condition set and did not have phenotypes similar to probands from DDD ('DDD-like'). The DDD-like probands were defined as those who:
1. were recruited into a disease model that was also used to recruit probands who had previously been recruited into DDD (section below on identifying probands overlapping between the two cohorts), or
2. had one the top five HPO terms used in DDD and their descendants, namely HP:0000729 (autistic behaviour), HP:0001250 (seizure), HP:0000252 (microcephaly), HP:0000750 (delayed speech and language development), and HP:0001263 (global developmental delay).

Probands recruited into the neurodegenerative disorders subcategory or with an age of onset greater than 16 years were removed from the DDD-like set, as were probands recruited into a disease subcategory for which the average age of probands was older than 16 years.

To define relatedness, we used a file generated by GEL consisting of a pairwise kinship matrix produced using the PLINK2 (refs. 65,66) implementation of the KING robust algorithm[67] and a --king-cutoff of 0.0442 (that is, $1/2^{4.5}$).

**Control cohorts.** The UK Household Longitudinal Study (UKHLS) cohort consists of a continuation of the British Household Panel Survey of individuals living in the United Kingdom[68,69]. The Avon Longitudinal Study of Parents and Children (ALSPAC) is a birth cohort study of children born in Avon, England with expected dates of delivery between 1 April 1991 and 31 December 1992 (ref. 70). Eligible pregnant women ($N = 13,761$) were recruited and their children have been phenotyped extensively over the past 30 years. Please note that the study website (http://www.bristol.ac.uk/alspac/researchers/our-data/) contains details of all the data that are available through a fully searchable data dictionary and variable search tool. The MCS is a birth cohort study of children born across the UK during 2000 and 2001 from 18,552 families[71,72]. Further information about recruitment of these cohorts is given in Supplementary Note 4.

**Ethics.** The DDD study has UK Research Ethics Committee approval (10/H0305/83, granted by the Cambridge South Research Ethics Committee and GEN/284/12, granted by the Republic of Ireland Research Ethics Committee). The 100,000 Genomes project was approved by the East of England−Cambridge Central Research Ethics Committee (REF 20/EE/0035). Ethical approval for ALSPAC was obtained from the ALSPAC Ethics and Law Committee and the Local Research Ethics Committees. Ethical approval for each sweep of MCS was obtained from NHS Research Ethics Committees (MREC). Ethical approval for the sixth MCS sweep, which included the collection of saliva samples from children and biological resident parents, was obtained from London-Central REC (MREC; 13/LO/1786).

### Preparation of genetic data

Individuals from DDD, UKHLS, ALSPAC and MCS were genotyped on various arrays, whereas GEL individuals were whole-genome sequenced. The available data are summarized here briefly:

A subset of the DDD cohort (all children and several thousand parents) was genotyped on three genotype array chips: the Illumina HumanCoreExome chip (CoreExome), the Illumina OmniChipExpress (OmniChip) and the Illumina Infinium Global Screening Array (GSA). Some probands were genotyped on more than one chip, as shown in Supplementary Fig. 9. In downstream analysis, we used the Core-Exome and OmniChip data for analyses of probands, and the GSA and OmniChip data for analyses of trios. QC of CoreExome (including DDD patients and 9,270 UKHLS controls genotyped on the same chip) and OmniChip data were performed by Niemi et al.[2] and we performed QC in the GSA data specifically for this paper (Supplementary Tables 13 and 14). The DDD cohort was also exome sequenced, and those data were used for the analyses involving rare variants.

GEL individuals were whole-genome sequenced with 150 bp paired-end reads using Illumina HiSeqX. Variant calling and QC were performed by Genomics England. We used 78,195 post-QC germline genomes from the Aggregated Variant Calls (aggV2) prepared by the GEL team. We kept variants that passed the QC filters shown in Supplementary Table 15.

Data we received from ALSPAC were processed in two batches[69]. In the first batch, we received post-QC array data for G0 mothers ($N = 8,884$) who were genotyped on the Illumina Human 660W chip and G1 children ($N = 8,932$) genotyped on the HumanHap550 quad chip. In the second batch, we received another 2,198 parents (G0 mothers and G0 partners[73]) who were genotyped on the CoreExome array.

We received data for 21,181 MCS samples who were genotyped using the GSA array chip[74].

We applied standard QC filters in each dataset separately, described further in Supplementary Methods. We used the maximum subset of unrelated individuals that passed QC. We did not use any statistical methods to predetermine sample sizes.

**Genetically predicted ancestry.** To avoid spurious results due to population stratification, all genetic analyses were conducted in a

genetically homogeneous subset of individuals with genetic similarity to British individuals from the 1,000 Genomes Project[75], henceforth referred to as having GBR ancestry. The Supplementary Methods provide detailed information on ancestry inference, but we summarize it briefly here. The identification of GBR-ancestry samples from the DDD CoreExome and OmniChip data was described previously[2]. To identify individuals of genetically inferred GBR ancestry in DDD GSA samples, we first projected post-QC samples onto 1,000 Genomes phase 3 individuals[75] (Supplementary Fig. 10). We then performed another principal component analysis (PCA) within the loosely defined European ancestry subset and identified a homogeneous subgroup (Supplementary Fig. 11) using uniform manifold approximation and projection (UMAP)[76]. As we merged parent–offspring trios genotyped on GSA and OmniChip array chips in downstream analysis, we kept GSA individuals who were similar to OmniChip individuals in terms of genetic ancestry in PCA space (Supplementary Fig. 12). In GEL, we used individuals with genetically inferred European ancestry, which were identified by the GEL bioinformatics team. We further restricted to a homogeneous subset ($N = 56,249$) that represents White British individuals (Supplementary Fig. 13). Array data received from the ALSPAC all had genetically predicted European ancestry, so we did not perform any filtering based on genetic ancestry. We performed similar PCA and UMAP clustering to identify individuals of GBR ancestry in MCS (Supplementary Figs. 14 and 15), and further filtered to individuals who self-reported as being of White ethnicity.

**Relatives within and across cohorts.** Within each dataset, we identified up to third-degree relatives (kinship coefficient greater than 0.0442 by KING v.2.2.4 (ref. 67) using post-QC genotyped array data or WGS data. We always used a subset of unrelated individuals (that is, more distant than third-degree relatives) in downstream analysis. In analyses using trios, we made sure probands in trios were unrelated and parents were unrelated with parents from other families.

In analyses combining DDD and GEL, we removed from GEL any participants who were also recruited into DDD and/or who were related to DDD participants, and also removed Scottish samples from DDD as we were unable to check whether GEL samples were related to them (Supplementary Methods). We removed individuals from the two birth cohorts who were related to each other or to DDD participants, which left 1,434 and 2,498 trios from ALSPAC and MCS, respectively (Supplementary Methods).

**Imputation and post-imputation QC.** Imputation of array data was performed in each genotyped cohort separately using the maximum number of variants available after QC. Before imputation, we removed palindromic SNPs, SNPs that were not in the imputation reference panel, and SNPs with mismatched alleles. DDD samples and UKHLS controls who were genotyped on the CoreExome array were imputed with the HRC r1.1 reference panel by Niemi et al.[2] DDD GSA and OmniChip samples and ALSPAC samples were imputed to the TOPMed r2 reference panel using the TOPMed imputation server, and the MCS samples to the HRC r1.1 reference panel[77–79]. We kept well-imputed common variants with Minimac4 $R^2 > 0.8$ and MAF > 1%. For polygenic score analyses, we subsequently restricted to common variants that passed these QC filters in all genotyped cohorts and also passed QC in the GEL WGS data.

**Extraction and QC of rare variants.** QC of DDD exome sequencing data and extraction of rare single-nucleotide variants, and insertion and deletions (indels) is summarized in Supplementary Table 16. Indels in the same gene and sample were removed (4% of indels with MAF < 1%), as these were often part of complex mutational events that would require haplotype-aware annotation.

For GEL, details of the QC of single-nucleotide variants and indels in the WGS data are provided by the GEL team[63,64] and variant QC is summarized in Supplementary Table 15. We use a custom python script to extract rare variants from GEL aggregated WGS variant call format files (aggV2). We filtered genotypes to those with genotype quality (GQ) ≥ 20 and read depth (DP) ≥ 10. We removed heterozygous genotypes that did not pass a binomial test of balanced REF and ALT alleles ($P < 1 × 10^{-3}$) or for which ALT/(REF + ALT) (AB ratio) was not between 0.2 and 0.8. We further removed variants with missing high-quality genotypes in more than 5% of all samples in aggV2. We removed indels in the same gene and sample for the same reason described above for DDD.

For MCS, details of the QC of exome sequencing data are in Supplementary Methods.

**Defining monogenic diagnoses in patients**

**DDD.** The DDD study identified clinically relevant rare variants from exome sequencing and microarray data using a filtering procedure described in ref. 58. The procedure focuses on identifying rare damaging variants that fit an appropriate inheritance mode in a set of genes that cause developmental disorders (DDG2P, https://www.deciphergenomics.org/ddd/ddgenes). Variants that pass clinical filtering are uploaded to DECIPHER[59], where the patients' clinicians are asked to classify them as definitely pathogenic, likely pathogenic, uncertain, likely benign or benign. We defined 'diagnosed' probands as those with one or more variants either annotated as pathogenic or likely pathogenic in DECIPHER by their referring clinician, or predicted as pathogenic or likely pathogenic using diagnoses autocoded following the American College of Medical Genetics and Genomics guidelines as described in ref. 1. All remaining probands were classed as 'undiagnosed'. Probands with a de novo diagnosis are those with a de novo mutation in a monoallelic or X-linked DDG2P gene that was either annotated or predicted as pathogenic or likely pathogenic.

**GEL.** The probands assigned diagnostic status were those included in the Genomic Medicine Service exit questionnaire, in which a clinician evaluated the pathogenicity of variants of interest identified through GEL's custom pipeline. We defined diagnosed probands as those that had a pathogenic or likely pathogenic variant that is annotated as partially or fully explaining their phenotype in this exit questionnaire. Probands with a de novo diagnosis are those whose pathogenic or likely pathogenic variants from the exit questionnaire were annotated as de novo protein-truncating or missense variants in DDG2P monoallelic or X-linked genes. We defined undiagnosed probands as those that were present in the exit questionnaire but not annotated as having a pathogenic or likely pathogenic variant and not annotated as 'yes' or 'partially' in the 'case_solved_family' column. We further removed from this undiagnosed set any probands who have potential diagnoses in the Diagnostic Discovery data in GEL, which is a list of variants submitted by researchers that are thought probably to be pathogenic by the GEL clinical team.

**Defining trio sample sets in DDD and GEL**
The procedure used for filtering trios used in DDD and GEL is shown in Supplementary Fig. 16. Briefly, in DDD, we combined data across GSA and OmniChip arrays and kept trios in which all three members had GBR ancestry and the proband had a neurodevelopmental condition. We excluded trios recruited from Scottish centres and kept unrelated trios. We then split trios into those with both parents unaffected and those with one or both parents affected. These were then categorized as genetically diagnosed or undiagnosed. We applied similar filtering in GEL trios. See Supplementary Methods for more information.

**GWAS of neurodevelopmental conditions**
We used PLINK v.1.9 to conduct a GWAS comparing individuals with neurodevelopmental conditions ($N = 3,618$) to controls ($N = 13,667$) in GEL, controlling for 20 genetic principal components, age and sex. Before running the GWAS, we removed variants with MAF < 1%, missingness > 2% or Hardy–Weinberg equilibrium $P < 1 × 10^{-5}$, and

performed a differential missingness test between the patients with neurodevelopmental conditions and controls and removed variants with $P < 1 \times 10^{-5}$. We repeated the GWAS comparing DDD patients with neurodevelopmental conditions on the CoreExome array ($N = 6,397$) to UKHLS controls ($N = 9,270$) using PLINK v.1.9, after excluding DDD patients recruited from Scottish centres.

We used METAL[80] to conduct an inverse-variance-weighted GWAS meta-analysis between the DDD-UKHLS and GEL GWASs. We removed palindromic SNPs with MAF > 0.4 as the strand could not be easily inferred using MAF. We also excluded SNPs with discordant allele frequency (difference > 0.05) between the two cohorts. This left 5,451,801 overlapping SNPs in the meta-analysis.

## Heritability

We used several methods to estimate the SNP heritability (the fraction of phenotypic variance explained by genome-wide common variants) on the liability scale assuming a cumulative population prevalence of 1% for rare neurodevelopmental conditions[2]. First, we applied two methods to estimate SNP heritability using individual-level data in DDD and GEL separately. We performed GREML-LDMS[81] stratified by linkage disequilibrium (LD; two bins of equal size) and MAF (three bins: 1–5%, 5–10%, >10%). We also ran phenotype correlation–genotype correlation (PCGC) regression[82], using the LDAK-Thin Model to compute the kinship matrix using the direct method. We corrected for sex, and ten genetic principal components as covariates in both methods. We then meta-analysed the SNP heritability estimates from DDD and GEL using an inverse-variance-weighted method. We also used linkage disequilibrium score regression (LDSC)[83] to estimate SNP heritability using summary statistics from the GWAS of neurodevelopmental conditions in DDD, in GEL, and a meta-analysis of the two cohorts. We used roughly 1 million common SNPs from HapMap3 with precomputed LD scores. We used the effective sample size ($4/(1/N_{cases} + 1/N_{controls})$) or the sum of two effective sample sizes for the meta-analysis and a sample prevalence of 50% in LDSC, as recommended previously[84]. We presented the GREML-LDMS estimate in the results, because the estimates were similar to PCGC, and LDSC estimates are known to be under-estimated, especially at low sample size. All estimates are reported in Supplementary Table 3.

## Genetic correlations

We used LDSC to estimate genetic correlations between the DDD GWAS or the meta-analysed GWAS for neurodevelopmental conditions and various brain-related traits and conditions listed in Supplementary Table 17. We did not use the GEL GWAS to calculate genetic correlations as the SNP heritability was not significantly different from zero according to LDSC.

To estimate the genetic correlations between neurodevelopmental conditions and various brain-related traits or conditions independent of cognitive performance or educational attainment signals, we used genomic structural equation modelling (GenomicSEM)[35,85]. We estimated the genetic correlation between the target trait and a latent variable representing the non-cognitive component of neurodevelopmental conditions, which was genetic influences on neurodevelopmental conditions that were not captured in the GWAS for cognitive performance[31]. We applied the GenomicSEM model without SNP effects. We also estimated genetic correlation with the 'non-educational attainment' latent variable, which represented genetic influences on neurodevelopmental conditions that were not accounted for by the educational attainment latent variable. We also used GenomicSEM to estimate the percentage of the genetic correlation between neurodevelopmental conditions and the target trait that was explained by latent variables, namely the cognitive and non-cognitive components of neurodevelopmental conditions when conditioning on the cognitive performance GWAS, or EA and non-EA components of neurodevelopmental conditions when conditioning

on the educational attainment GWAS (Supplementary Fig. 1 and Extended Data Fig. 9bc). The GenomicSEM model and formulae used to estimate these percentages can be found in Supplementary Fig. 17 and Supplementary Methods.

## Calculating polygenic scores

For calculating polygenic scores, we used the set of SNPs that were well imputed in all array cohorts (Minimac4 $R^2 > 0.8$), passed QC in GEL aggV2 samples, and had MAF > 1% in all cohorts. We used LDPred[86] to estimate weights for calculating polygenic scores and an LD reference panel composed of HapMap3 (ref. 87) common variants based on 5,000 unrelated individuals of genetically inferred White British ancestry from the UK Biobank[88] (Supplementary Methods). GWAS summary statistics for years of schooling (a measure for EA)[31], the non-cognitive component of educational attainment (NonCogEA)[35], cognitive performance (CP)[31], schizophrenia (SCZ)[32] and neurodevelopmental conditions[2] were matched with the list of overlapping SNPs (Supplementary Table 17). $PGS_{NDC,DDD}$ was evaluated in the DDD OmniChip samples and the GEL samples that were not in the DDD GWAS. To make polygenic scores comparable across cohorts (DDD, GEL, UKHLS, MCS and ALSPAC), we performed a joint PCA across all cohorts and adjusted the raw scores for 20 principal components. For most analyses and unless noted otherwise, residuals were scaled so that the combined set of unrelated control samples from GEL and UKHLS (or GEL controls only for $PGS_{NDC,DDD}$) had mean of 0 and s.d. of 1, and the resultant scores were used for all analyses unless otherwise indicated. In Fig. 3b and Extended Data Fig. 5, we instead show principal component-adjusted polygenic scores that were standardized using weighted MCS average polygenic scores that should represent an unbiased estimate representative of the background population (Supplementary Methods). We also constructed composite polygenic scores combining individual polygenic scores (Supplementary Methods).

## Analyses of polygenic scores

**Evaluating variance explained by polygenic score.** We evaluated how much variance in risk of neurodevelopmental conditions was explained by the polygenic score on the liability scale[82,89,90]. We compared 6,397 probands with neurodevelopmental conditions from DDD to 9,270 controls from UKHLS, and 3,618 probands with neurodevelopmental conditions from GEL to 13,667 GEL controls defined as described above. We assumed the population prevalence of neurodevelopmental conditions to be 1% (ref. 2).

**Comparing polygenic scores between different subsets.** We used two-sided $t$-tests to compare polygenic scores between different groups of probands, parents and controls seen in Figs. 2a and 3b, Extended Data Figs. 5 and 6 and Supplementary Tables 5–7. We report the mean difference in principal component-corrected polygenic scores between groups. Groups who were compared with each other include:
- Combined set of controls from GEL and UKHLS
- Control individuals from UK birth cohorts, ALSPAC and MCS
- Undiagnosed neurodevelopmental condition (NDC) probands regardless of trio status
- Diagnosed NDC probands regardless of trio status
- Undiagnosed NDC probands for whom both parents are unaffected
- Unaffected parents of undiagnosed NDC probands
- Undiagnosed NDC probands with one or both parents affected
- Affected parents of undiagnosed NDC probands
- Diagnosed NDC probands for whom both parents are unaffected
- Unaffected parents of diagnosed NDC probands
- NDC probands with de novo diagnoses for whom both parents are unaffected
- Unaffected parents of NDC probands with de novo diagnoses
- Diagnosed NDC probands with one or both parents affected
- Affected parents of diagnosed NDC probands.

Note that 'undiagnosed' and 'diagnosed' here indicate whether the patient has a monogenic diagnosis. The sample size of each subset is listed in Supplementary Table 1. We excluded controls from UKHLS as well as DDD CoreExome and GSA probands when testing the DDD-derived polygenic score for neurodevelopmental conditions (as these had been included in the original GWAS, whereas the individuals genotyped on the OmniChip had not). All the $t$-tests involving probands with a neurodevelopmental condition or their parents were performed in samples from DDD and GEL combined.

We also compared female probands versus male probands without a monogenic diagnosis regardless of trio status (2,427 and 1,574 male probands from DDD and GEL, and 1,426 and 918 female probands from DDD and GEL), and unaffected mothers versus unaffected fathers (1,523 trios from DDD and 1,343 trios from GEL) using two-sided $t$-tests (Extended Data Fig. 8ab).

**Polygenic score and diagnostic status.** We compared average polygenic scores in probands with a neurodevelopmental condition with and without a monogenic diagnosis using two-sided $t$-tests, combining probands from DDD and GEL regardless of whether they were in a trio or not. We compared subgroups from families affected by neurodevelopmental conditions to the combined control set from UKHLS and GEL, as well as to unrelated children from the MCS cohort who were reweighted using available sociodemographic data to make them more representative of the general UK population (Supplementary Note 4).

Within DDD ($N = 7,549$ without excluding Scottish samples or samples who were related to GEL participants), we tested whether the proband's $PGS_{EA}$ was associated with factors affecting getting a diagnosis in linear regression models:

$$PGS \sim factor$$

Note that we use the tilde symbol to indicate that the variable before the tilde was regressed on the variable(s) after the tilde. We investigated the following binary factors: trio status ($N = 5,507$ with both parents exome sequenced but not necessarily genotyped), proband sex ($N = 4,421$ male probands), whether the proband had any affected first-degree relatives ($N = 1,623$), whether the proband was born preterm ($N = 1,098$ with gestation <37 weeks), whether the mother had diabetes ($N = 242$) and whether the proband had severe intellectual disability or developmental delay (ID/DD; $N = 941$) versus mild or moderate ID/DD ($N = 1,887$). We compared probands with the above-mentioned characteristics to all other probands, except when comparing probands with severe versus mild or moderate ID/DD for which we excluded probands without ID/DD or with ID/DD of unknown severity. We also investigated a continuous factor, the degree of consanguinity, quantified by the fraction of the genome in runs of homozygosity ($F_{ROH}$) divided by 0.0625, which is the expected fraction given a first-cousin marriage.

We also tested whether the mother's or father's $PGS_{EA}$ was associated with the above factors, in a total of 2,497 samples; we did not test for association with trio status as parental genotype data were only available for full trios anyway.

To assess how the association between the above-mentioned factors and diagnostic status changed after correcting for proband's $PGS_{EA}$, as well as how the association between proband's $PGS_{EA}$ and diagnostic status changed after controlling for these factors, we fitted the following logistic regression models:

$$Diagnostic\ status \sim factor$$

$$Diagnostic\ status \sim PGS$$

$$Diagnostic\ status \sim PGS + factor$$

We also fitted a joint model to assess the effect of $PGS_{EA}$ on diagnostic status controlling for both trio status and prematurity, which showed significant associations with both $PGS_{EA}$ and diagnostic status. We excluded from this joint model factors that were not associated with $PGS_{EA}$ or diagnostic status within the DDD samples with European ancestry (sex, maternal diabetes and $F_{ROH}$), and factors that are likely to be the consequence of having or not having a monogenic diagnosis, rather than a cause of getting a diagnosis (severity of ID/DD and having affected family members).

See the Supplementary Methods for a description of estimation of the odds ratio of diagnosis for different configurations of affected relatives shown in Extended Data Fig. 7a.

**Evaluating over-transmission of polygenic scores.** We conducted polygenic transmission disequilibrium tests (pTDTs) in undiagnosed and diagnosed probands from DDD ($N = 1,523$ undiagnosed, 443 diagnosed) and GEL ($N = 1,343$ undiagnosed, 507 diagnosed) combined. We also conducted pTDTs in these trios excluding autistic probands.

The pTDT is a two-sided one-sample $t$-test of the probands' polygenic score deviation from expectation, which is their parents' mean polygenic score. The pTDT deviation is defined as:

$$pTDT\ deviation = PGS_{child} - (PGS_{mother} + PGS_{father})/2$$

To evaluate whether the pTDT deviation is significantly different from 0, the pTDT test statistic ($t_{pTDT}$) is defined as:

$$t_{pTDT} = \frac{mean(pTDT\ deviation)}{\frac{s.d.(pTDT\ deviation)}{\sqrt{n}}}$$

**Association with non-transmitted alleles.** Alleles in parents that are not transmitted to the child can still influence the child's phenotype by affecting the parents' behaviour. This phenomenon is called genetic nurture or indirect genetic effects[4,26,30]. Alleles that are transmitted to the child can influence the child's phenotype both directly (direct genetic effects) and indirectly through other relatives who carry the same alleles (indirect genetic effects) and whose behaviour is influenced by those alleles. Kong et al. proposed to estimate the direct genetic effect as $\delta = \theta_T - \theta_{NT}$, where $\theta_T$ indicates the effect of parental transmitted alleles and $\theta_{NT}$ indicates the effect of parental non-transmitted alleles, which capture both the indirect genetic effects and potential confounding factors[4,91]. We can estimate $\theta_T$ and $\theta_{NT}$ of a given polygenic score in the following regression model:

$$child's\ phenotype \sim \hat{\theta}_T \times PGS_T + \hat{\theta}_{NT} \times PGS_{NT}$$

where $PGS_T$ is a polygenic score calculated using transmitted alleles (which is the child's polygenic score), and $PGS_{NT}$ is a polygenic score calculated using parental non-transmitted alleles, which is equivalent to the difference between the sum of parents' polygenic scores and the child's polygenic score. This model can also be rewritten as:

$$child's\ phenotype \sim (\hat{\theta}_T - \hat{\theta}_{NT}) \times PGS_{child} + \hat{\theta}_{NT} \times (PGS_{mother} + PGS_{father})$$

Therefore, in a regression model in which the child's polygenic score and parents' polygenic scores are both fitted, the coefficient on the child's polygenic score captures the direct genetic effect, and the coefficient on parents' polygenic scores captures the association between non-transmitted alleles and the child's phenotype. The latter may reflect true indirect genetic effects as well as confounding effects such as uncorrected population stratification and parental assortment[29]. Thus, we refer to the coefficients on parents' polygenic scores in this

model as 'non-transmitted coefficients' rather than simply 'indirect genetic effects', following Young et al.[24], as they are mathematically equivalent to the coefficients on the polygenic score constructed from the non-transmitted alleles in a joint regression with the proband's polygenic score.

We evaluated direct genetic effects ($\hat{\delta}$) and effects of maternal and paternal non-transmitted common alleles ($\hat{\theta}_{m,\mathrm{NT}}$ and $\hat{\theta}_{f,\mathrm{NT}}$) on case status in the following trio model using logistic regression on polygenic scores:

$$1_{\mathrm{NDC\,status}} \sim \hat{\delta} \times \mathrm{PGS}_{\mathrm{child}} + \hat{\theta}_{m,\mathrm{NT}} \times \mathrm{PGS}_{\mathrm{mother}} + \hat{\theta}_{f,\mathrm{NT}} \times \mathrm{PGS}_{\mathrm{father}}$$

where $1_{\mathrm{NDC\,status}}$ is an indicator variable for whether the individual is a case with a neurodevelopmental condition (1) or control (0). We also ran the regression without correcting for parents' polygenic scores (proband-only model) in the same samples for comparison:

$$1_{\mathrm{NDC\,status}} \sim \hat{\theta}_{\mathrm{T}} \times \mathrm{PGS}_{\mathrm{child}}$$

Probands with a neurodevelopmental condition were from DDD and GEL trios where the proband was undiagnosed and both parents were unaffected ($N = 2{,}866$ trios). Control samples were trios from the two birth cohorts (ALSPAC and MCS, $N = 1{,}434$ and $N = 2{,}498$, respectively) as well as trios from GEL where the proband did not have DDD-like developmental disorders or neurodevelopmental conditions ($N = 872$).

We verified that the polygenic scores in the trio model did not show excessive collinearity (Supplementary Methods).

We performed various sensitivity analyses in the following subsets (Supplementary Fig. 4): patients versus controls from GEL trios only, and patients from GEL and DDD versus each of the three control cohorts separately (GEL, MCS or ALSPAC). We also conducted the analysis while controlling for the rare variant burden score (RVBS) in GEL trios (Extended Data Fig. 10b; section below on 'Analyses of polygenic scores and rare coding variants').

$$1_{\mathrm{NDC\,status}} \sim \mathrm{PGS}_{\mathrm{child}} + \mathrm{RVBS}_{\mathrm{child}} + \mathrm{PGS}_{\mathrm{mother}} \\ + \mathrm{RVBS}_{\mathrm{mother}} + \mathrm{PGS}_{\mathrm{father}} + \mathrm{RVBS}_{\mathrm{father}}$$

We restricted this latter analysis to GEL trios to minimize artefactual differences in rare variant calling and QC between cases and controls, which could otherwise create spurious associations.

See the Supplementary Methods for a description of how we modified the running of this trio model to investigate the hypothesis that the effects of non-transmitted alleles associated with educational attainment and cognition might be mediated by prematurity.

**Analyses of polygenic scores and rare coding variants.** Sequence data from DDD, GEL and MCS were annotated with the Ensembl Variant Effect Predictor (VEP)[92]. We kept the 'worst consequence' annotation across transcripts. From parents and probands, we extracted autosomal heterozygous PTVs (transcript ablation, frameshift, splice acceptor, splice donor and stop gained) annotated as high-confidence by LOFTEE[93] (HC PTVs), as well as variants in the following classes that we grouped as 'missense': missense, stop lost, start lost, inframe insertion, inframe deletion and loss-of-function variants annotated as low-confidence by LOFTEE[93]. We retained rare variants with MAF $< 1 \times 10^{-5}$ in each gnomAD super-population and MAF $< 1 \times 10^{-4}$ in the respective cohorts.

We considered four (non-mutually exclusive) groups of damaging rare variants:
1. HC PTVs in constrained genes (pLI $> 0.9$)[94]
2. HC PTVs and missense variants (MPC $\geq 2$)[95] in constrained genes (pLI $> 0.9$)
3. HC PTVs in monoallelic DDG2P genes with a loss-of-function mechanism (that is, 'absent gene product')

4. HC PTVs and missense variants (MPC $\geq 2$) in monoallelic DDG2P genes with a loss-of-function mechanism.

We retained probands and parents who were heterozygous for these variants. We required the variants in the children to have been inherited from a parent.

To investigate whether parental assortment leads to correlated rare and common variant burden, we calculated rare variant burden scores as the number of rare variants in the classes described above, then calculated the Pearson's correlation coefficients between rare variant burden scores and polygenic scores using the 'cor' function in R. We used trios in which both parents were unaffected in this analysis. Rare variant burden scores were corrected for 20 genetic principal components using linear regression models. We then calculated the correlation coefficients between the principal component-adjusted rare variant burden scores in parents and the principal component-adjusted polygenic scores in their partners. We also assessed the correlation within the same person among either children or parents. We repeated the analysis in subsets of trios in which the proband was undiagnosed as well as in trios in which the proband had a monogenic de novo diagnosis (Supplementary Fig. 6). The main analysis in Fig. 5 and the sensitivity analysis in Extended Data Fig. 10b is based on group 2 above, whereas Supplementary Figs. 6–8 show the results for all four groups of variants. To investigate whether the results were affected by uncorrected population structure, we also calculated rare variant burden scores using rare synonymous variants in either monoallelic DDG2P genes with a loss-of-function mechanism or constrained genes, and assessed their correlation with polygenic score.

To assess whether polygenic scores modify penetrance of rare inherited variants, we conducted one-sided paired $t$-tests comparing the polygenic score between unaffected parents transmitting a damaging variant to their affected offspring who inherited the variant (Supplementary Fig. 8). We hypothesized that the unaffected parents would have a more protective polygenic background than their affected offspring (indicated by higher $\mathrm{PGS}_{\mathrm{EA}}$, $\mathrm{PGS}_{\mathrm{CP}}$, $\mathrm{PGS}_{\mathrm{NonCogEA}}$ and lower $\mathrm{PGS}_{\mathrm{SCZ}}$, $\mathrm{PGS}_{\mathrm{NDC,DDD}}$). If more than one parent transmitted a variant to a proband, one parent–child pair was chosen at random from the trio. We used trios in which the proband was undiagnosed and both parents were unaffected in this analysis.

### Construction and use of weights for MCS

We were concerned that control cohorts might not be random samples of the population with respect to educational attainment, and that this might bias our effect sizes for the difference in polygenic scores between cases and controls (Supplementary Note 4). We decided to use MCS, for which extensive sociodemographic data are available, to calculate a mean polygenic score that would be representative of the general population, using inverse-probability weighting. MCS deliberately oversampled minority ethnic and disadvantaged individuals in the United Kingdom[96] (sampling bias), and they provide sampling weights to account for this. Furthermore, missingness in each wave of data collection, including the collection of DNA for genotyping, was non-random (non-response bias). To correct for non-response bias, we produced non-response weights per individual using the inverse of the probability of being genotyped estimated from a logistic regression, considering covariates collected at the first study sweep, as previously described[96,97] (Supplementary Methods). We fitted the model to predict who was in the sample of unrelated children of GBR ancestry with genotype data ($N = 5{,}884$ of 6,036 children who had complete data for these covariates), and separately to predict who was in the subset of these that also had genotype data on both parents ($N = 2{,}445$ of 2,498 trio children who had no missingness). To produce weights that account for both sampling bias and non-response bias, we multiplied the non-response weight from regression models by the sampling weights provided by MCS. These weights were then used to calculate adjusted polygenic

scores shown in Fig. 3b and Extended Data Figs. 5 and 6c and adjusted correlation between polygenic score and rare variant burden score shown in Supplementary Fig. 7.

## Reporting summary

Further information on research design is available in the Nature Portfolio Reporting Summary linked to this article.

## Data availability

The raw and post-QC genotype array data and exome sequence data from DDD are available through European Genome-phenome Archive, under EGAS00001000775. WGS data and phenotypic data from the 100,000 Genomes project can be accessed by application to Genomics England (https://www.genomicsengland.co.uk/research/academic/join-gecip). GWAS summary statistics of neurodevelopmental conditions generated in this study are available at Figshare (https://doi.org/10.6084/m9.figshare.27060895)[98]. Researchers can apply to access genotype array data from UKHLS (https://www.understandingsociety.ac.uk/documentation/access-data/), ALSPAC (https://www.bristol.ac.uk/alspac/researchers/access/) and MCS (https://cls.ucl.ac.uk/data-access-training/data-access/). Publicly available GWAS summary statistics can be accessed at various resources: http://www.thessgac.org/data, https://pgc.unc.edu/for-researchers/download-results/ and https://egg-consortium.org/Gestational-duration-2023.html. DDG2P genes can be downloaded at https://www.deciphergenomics.org/ddd/ddgenes.

## Code availability

We used publicly available software: LDpred (https://github.com/bvilhjal/ldpred), LDSC (https://github.com/bulik/ldsc), GCTA-LDMS (https://yanglab.westlake.edu.cn/software/gcta/#GREMLinWGSorimputeddata), PCGC regression (https://dougspeed.com/pcgc-regression/) and GenomicSEM (https://github.com/PerlineDemange/non-cognitive/blob/master/GenomicSEM/Genetic%20correlations/Without%20using%20SNP%20effects/function_rG_woSNP.R). Custom code is available on GitHub (https://github.com/QinqinHuang/NDC_polygenic).

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

**Acknowledgements** We are grateful to families for their participation and engagement in the DDD study and 100,000 Genomes projects; without them, this research would not be possible. We also thank their clinicians and our colleagues (including E. Delage and the Sanger Human Genetics Informatics team, particularly I. Popov and R. Eberhardt) who assisted in the generation and processing of data. We are grateful to J. Hastings-Ward, H. Podd and H. Humphrey from the Participant Panel for the 100,000 Genomes project, A. L. Taylor Tavares from Genomics England and S. Wynn from the patient organization Unique, for their assistance with writing the frequently asked questions. We thank A. Kousathanas and L. Moutsianas from Genomics England Bioinformatics Research Services for help with data QC, H. Wong for useful discussions on prematurity, M. Nivard for advice on use of GenomicSEM and A. Ronald, N. Wray, N. Martin and O. Wootton for helpful discussions. DDD: the DDD study presents independent research commissioned by the Health Innovation Challenge Fund (grant no. HICF-1009-003). The full acknowledgements can be found at www.ddduk.org/access.html. This study makes use of DECIPHER, which is funded by the Wellcome Trust. GEL: this research was made possible through access to data in the National Genomic Research Library, which is managed by Genomics England Limited (a wholly owned company of the Department of Health and Social Care). The National Genomic Research Library holds data provided by patients and collected by the NHS as part of their care and data collected as part of their participation in research. The National Genomic Research Library is funded by the National Institute for Health Research and NHS England. The Wellcome Trust, Cancer Research UK and the Medical Research Council have also funded research infrastructure. UKHLS: we used data from 'Understanding Society: The UK Household Longitudinal Study', which is led by the Institute for Social and Economic Research at the University of Essex and funded by the Economic and Social Research Council (grant number ES/M008592/1). The data were collected by NatCen and the genome-wide scan data were analysed by the Wellcome Trust Sanger Institute. Data governance was provided by the METADAC data access committee, funded by the ESRC, Wellcome and MRC (grant number MR/N01104X/1). ALSPAC: we are extremely grateful to all the families who took part in ALSPAC, the midwives for their help in recruiting them and the whole ALSPAC team, which includes interviewers, computer and laboratory technicians, clerical workers, research scientists, volunteers, managers, receptionists and nurses. The UK Medical Research Council and Wellcome (grant no. 217065/Z/19/Z) and the University of Bristol provide core support for ALSPAC. This publication is the work of the authors and H.C.M. will serve as a guarantor for the contents of this paper. Genome-wide genotyping data was generated by Sample Logistics and Genotyping

Facilities at the Wellcome Sanger Institute and LabCorp (Laboratory Corporation of America) using support from 23andMe. MCS: we are grateful to the Centre for Longitudinal Studies (CLS), UCL Social Research Institute, for the use of these data and to the UK Data Service for making them available. However, neither CLS nor the UK Data Service bear any responsibility for the analysis or interpretation of these data. This research was funded in part by Wellcome (grant no. 220540/Z/20/A, 'Wellcome Sanger Institute Quinquennial Review 2021–2026'). For the purpose of open access, the authors have applied a CC-BY public copyright license to any author accepted manuscript version arising from this submission. D.B. thanks the University of Cambridge Amgen Scholar Program for support. H.V.F. was supported by the NIHR Cambridge Biomedical Research Centre (grant no. NIHR203312). The views expressed are those of the authors and not necessarily those of the NIHR or the Department of Health and Social Care.

**Author contributions** Q.Q.H. and E.M.W. conducted most of the analyses, with the remainder being conducted by P.C. and D.S.M. Q.Q.H. and E.M.W. carried out data preparation and QC, with assistance from K.E.S., V.K.C., P.D., S.L., T.M., M.K., S.A. and D.B. E.S., C.F.W. and H.V.F. helped supervise the DDD study, together with M.E.H. Q.Q.H., E.M.W., D.S.M., E.J.R., V.W., A.S.Y. and M.E.H. provided key intellectual input. H.C.M. supervised the analyses and directed the study. Q.Q.H., E.M.W. and H.C.M. wrote the first draft of the manuscript, with input from P.C., D.S.M., J.C.B., V.W., A.S.Y. and M.E.H. All authors read and commented on the final manuscript.

**Competing interests** M.E.H. is a cofounder of, consultant to and holds shares in Congenica, a genetics diagnostic company and is also a consultant to AstraZeneca Centre for Genomics Research.

**Additional information**
**Correspondence and requests for materials** should be addressed to Hilary C. Martin.

**Aim 1**: to better understand the nature of common variant risk for NDCs and its genetic overlap with brain-related conditions and traits

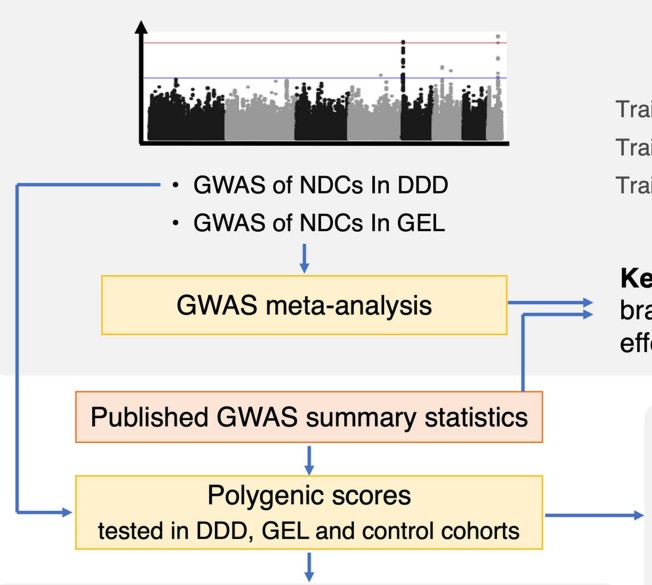

- GWAS of NDCs In DDD
- GWAS of NDCs In GEL

GWAS meta-analysis

Published GWAS summary statistics

Polygenic scores
tested in DDD, GEL and control cohorts

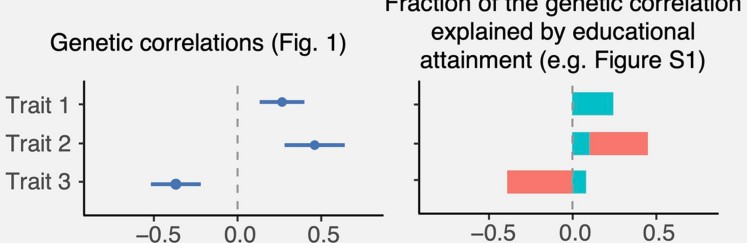

Genetic correlations (Fig. 1)

Fraction of the genetic correlation explained by educational attainment (e.g. Figure S1)

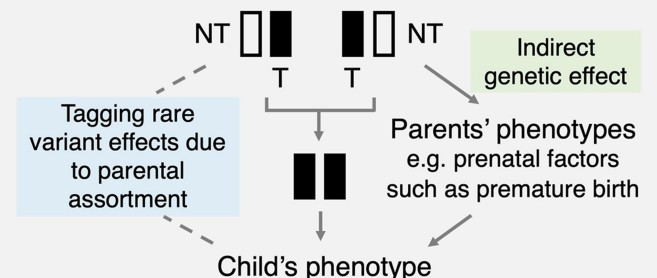

**Key finding**: NDCs are genetically correlated with multiple brain-related traits, partially but not entirely driven by genetic effects shared with educational attainment.

**Aim 2**: to elucidate the interplay between common and rare variant risk

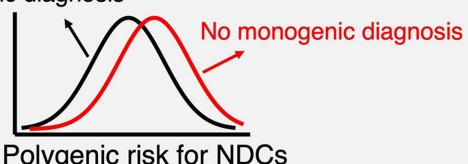

monogenic diagnosis

No monogenic diagnosis

Polygenic risk for NDCs

**Key finding**: Patients with a monogenic diagnosis have less polygenic risk than undiagnosed patients, consistent with the liability threshold model (Fig. 2).

**Aim 3**: to explore the contribution of common variants in the parents to their child's risk

NT  T  T  NT

Indirect genetic effect

Tagging rare variant effects due to parental assortment

Parents' phenotypes e.g. prenatal factors such as premature birth

Child's phenotype

**Key finding**: Non-transmitted common alleles in the parents associated with educational attainment and cognitive performance are associated with their child's risk in dependent of child's genetics; transmitted common alleles ascertained for their association with NDCs directly influence child's risk (Fig. 4).

**Extended Data Fig. 1 | Outline of main questions and analyses in this paper, and the key findings from these.** We conducted a GWAS of neurodevelopmental conditions in GEL, and meta-analysed the results with the DDD-derived GWAS. We calculated genetic correlations between neurodevelopmental conditions and various brain-related conditions and traits using published GWAS summary statistics (Fig. 1), then estimated the fraction of each genetic correlation that was explained by genetic effects shared with educational attainment (Supplementary Fig. 1). Next we constructed polygenic scores for neurodevelopmental conditions and relevant traits using the DDD-derived GWAS and external GWASs. We tested for differences in average polygenic scores between patients with versus without a monogenic diagnosis (Fig. 2). Given that clinically unaffected parents and probands showed similar polygenic background (Fig. 3), we tested whether non-transmitted common alleles in the parents were correlated with their child's risk of neurodevelopmental conditions (Fig. 4), and explored two potential explanations. In the figure in the bottom right, T and NT indicate transmitted and non-transmitted alleles in the parents, respectively. We indicate two possible reasons (left and right) that parental non-transmitted alleles may associate with the child's phenotype, both of which can pertain to either maternal and/or paternal non-transmitted alleles. The first is that prenatal risk factors, specifically prematurity, might mediate the correlation between parental non-transmitted alleles and child's risk (Extended Data Fig. 9) (a type of indirect genetic effect, which has a causal interpretation, hence the arrow); we did not find significant evidence for this (Supplementary Fig. 5). A second possible explanation we explored (blue box) is that the non-transmitted common alleles may simply tag rare variant effects due to parental assortment (hence, the association may simply reflect correlation with the causal factor, as indicated by the dotted line). We show a correlation between common and rare variant components of risk for neurodevelopmental conditions in Fig. 5.

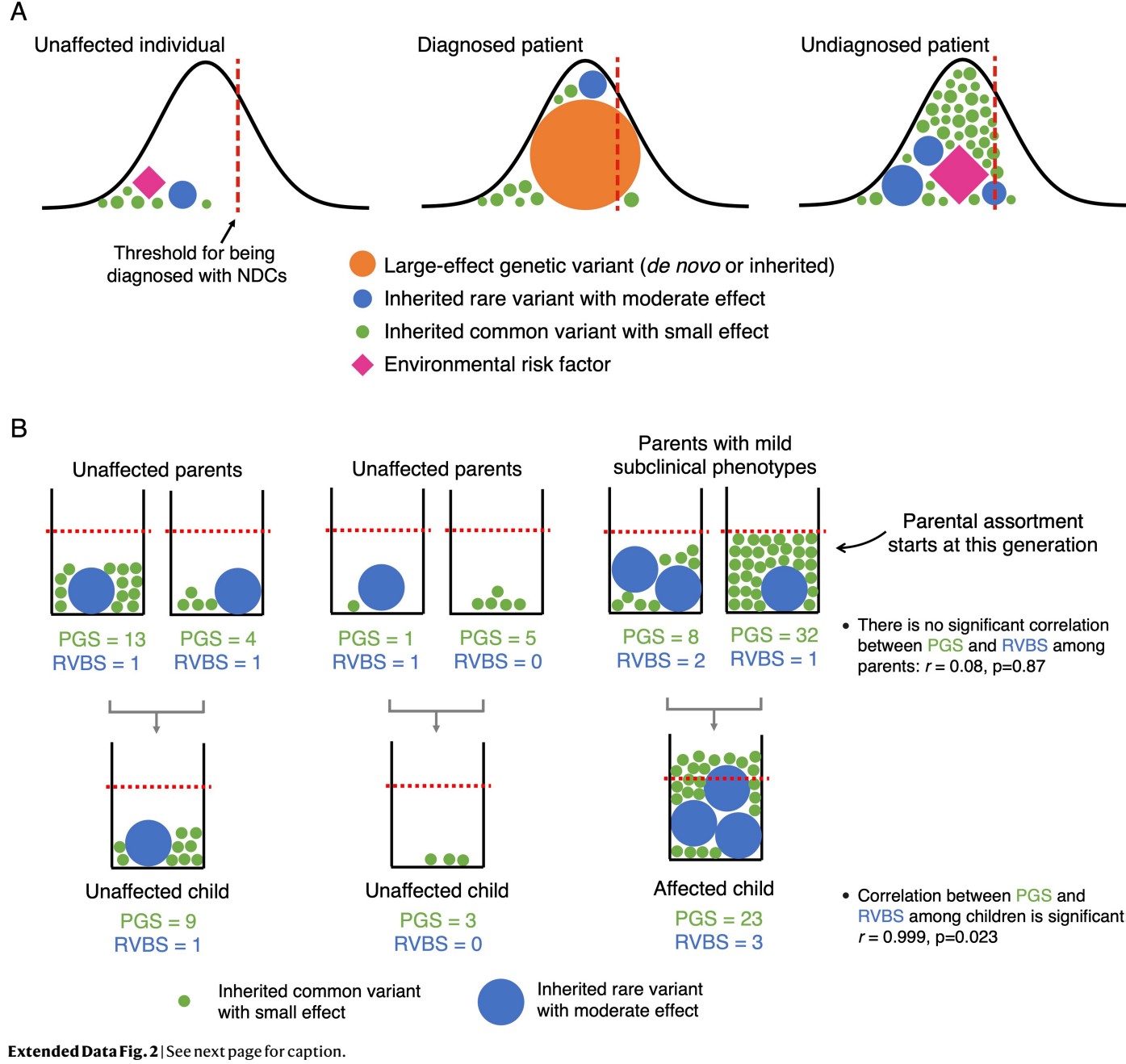

A

Unaffected individual

Diagnosed patient

Undiagnosed patient

Threshold for being diagnosed with NDCs

Large-effect genetic variant (*de novo* or inherited)

Inherited rare variant with moderate effect

Inherited common variant with small effect

Environmental risk factor

B

Unaffected parents

Unaffected parents

Parents with mild subclinical phenotypes

Parental assortment starts at this generation

PGS = 13
RVBS = 1

PGS = 4
RVBS = 1

PGS = 1
RVBS = 1

PGS = 5
RVBS = 0

PGS = 8
RVBS = 2

PGS = 32
RVBS = 1

- There is no significant correlation between PGS and RVBS among parents: *r* = 0.08, p=0.87

Unaffected child

PGS = 9
RVBS = 1

Unaffected child

PGS = 3
RVBS = 0

Affected child

PGS = 23
RVBS = 3

- Correlation between PGS and RVBS among children is significant: *r* = 0.999, p=0.023

Inherited common variant with small effect

Inherited rare variant with moderate effect

**Extended Data Fig. 2** | See next page for caption.

**Extended Data Fig. 2 | Schematic illustrating key concepts in the paper.**
(**A**) Illustration of the liability threshold model for rare neurodevelopmental conditions. The figure shows why one might expect patients with a monogenic diagnosis to have less polygenic (common variant) risk than those without a monogenic diagnosis. The normal distribution represents the underlying distribution of liability in the population, which is assumed to be Gaussian. Both genetic and environmental factors of different effects contribute to this total liability. Each panel represents a hypothetical example of one individual, either unaffected, affected and diagnosed with a monogenic cause, or affected and without a monogenic diagnosis. The red line indicates a threshold for being diagnosed with neurodevelopmental conditions. Circles represent different genetic factors, and diamonds represent environmental factors. The size of circles and diamonds represents their impact on disease risk. The second patient, who has a monogenic diagnosis, has fewer green circles (fewer NDC risk-increasing common variants) than the undiagnosed patient on the right, since the orange circle (diagnostic large-effect variant) is sufficient on its own to push the diagnosed patient over the diagnostic threshold. (**B**) Illustration of how parental assortment leads to correlation between the common and rare variant components of risk for neurodevelopmental conditions. The figure shows three hypothetical families in which the mother in each pair has a similar level of cognitive ability/educational attainment to the father (a phenomenon called parental assortment). Mother and father from the same family also have similar genetic predispositions towards these traits and hence also towards

risk of NDCs. Numbers on the bottom of each jar represents the simulated count of risk alleles from NDC-associated common variants represented by green circles (PGS) and that from NDC-associated rare variants represented by blue circles (RVBS). In the lefthand two families, both parents have a low risk for NDCs, as shown by the total height of the blue and green circles being well below the liability threshold indicated by the red line. Children in these two families have inherited about the expected number of parental common and rare variant risk alleles (the average of their parents) and also have low risk for developing NDCs. In the third family, both parents are not clinically affected by NDCs but both have subclinical phenotypes (for example, mild learning difficulties) due to having more risk alleles at rare (lefthand parent) or common (righthand parent) variants which contribute to reduced cognitive performance. Their child's risk is above the diagnostic threshold indicated by the red line. In the parents' generation, when parental assortment starts, there is no significant correlation between PGS and RVBS (two-sided $P = 0.87$, Pearson correlation $r = 0.08$ using the simulated counts). In their children, those who have more polygenic risk also tend to have more rare variant risk (correlation between PGS and RVBS is significant with $P = 0.023$, $r = 0.999$). Note that the values for PGS and RVBS have been chosen deliberately to emphasize the point for illustrative purposes, but the correlation in the child is much weaker than this in reality (Fig. 5). Also note that when analysing the real data, we regressed out principal components from PGS and RVBS before calculating the correlations.

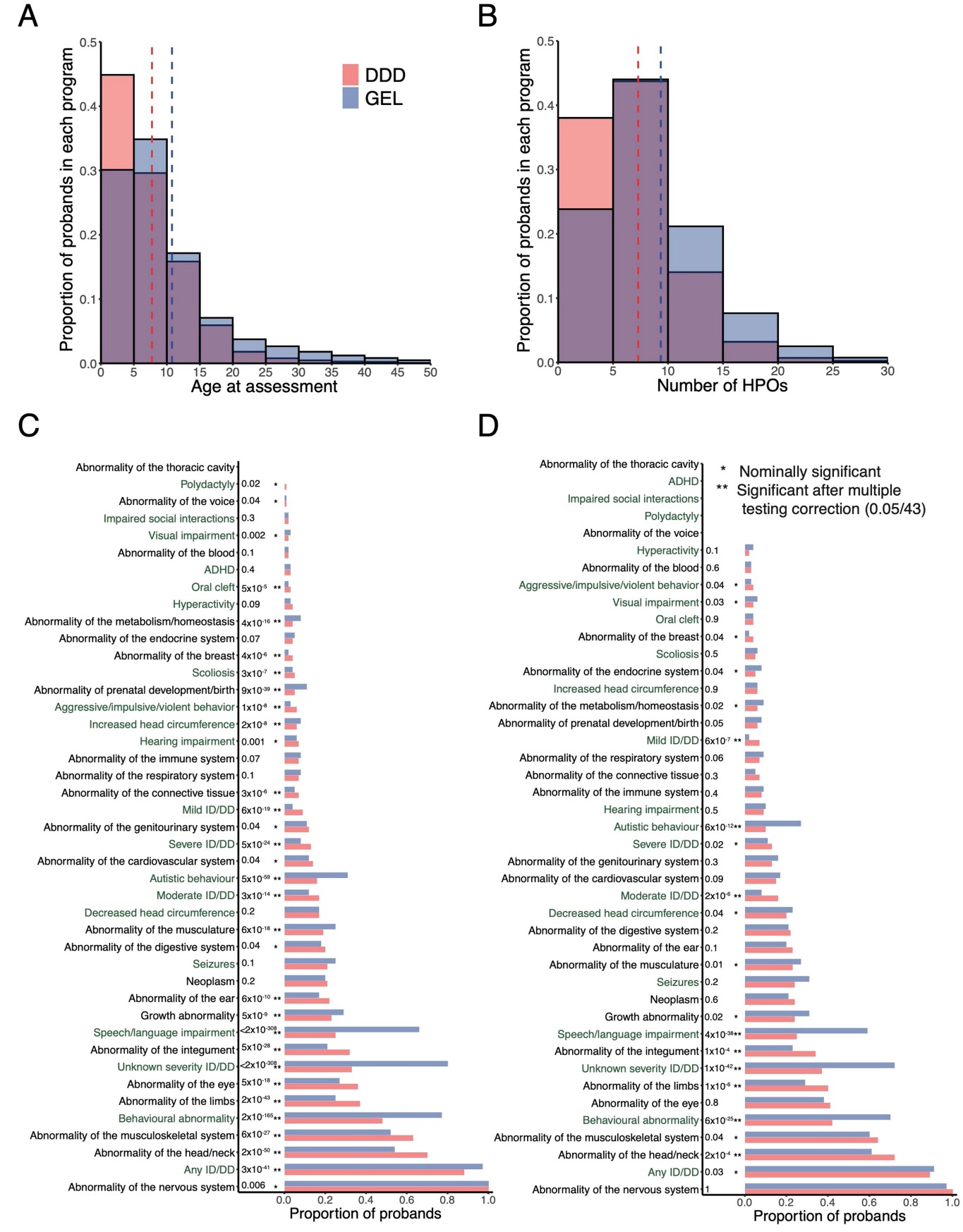

**Extended Data Fig. 3** | See next page for caption.

**Extended Data Fig. 3 | Phenotypic comparisons between DDD and GEL.** Distribution of age at assessment (**A**) and number of HPO terms (**B**) in both DDD and GEL probands with neurodevelopmental conditions who have GBR ancestry. The vertical lines indicate the means. A small number of probands in each program were aged over 50 and had more than 30 HPOs, and these have been omitted from the plot due to data sharing restrictions. (**C**) Proportion of probands from each cohort with at least one HPO term within the indicated chapter (black text) or specific phenotype (green text), ordered by the prevalence in DDD. The asterisks indicate results from a logistic regression testing whether there was a significant difference in phenotype prevalence between cohorts after controlling for sex and age (** indicates two-sided $P < 0.05/43$; * indicates two-sided $P < 0.05$; exact $P$ values are annotated beside the asterisks). (**D**) Proportion of probands recruited to both DDD and GEL ($N = 789$) with at least one HPO term within the indicated chapter (black) or specific phenotype (green text) from the phenotype information from each program, ordered by the prevalence in DDD. The same logistic regression was used as in (**C**).

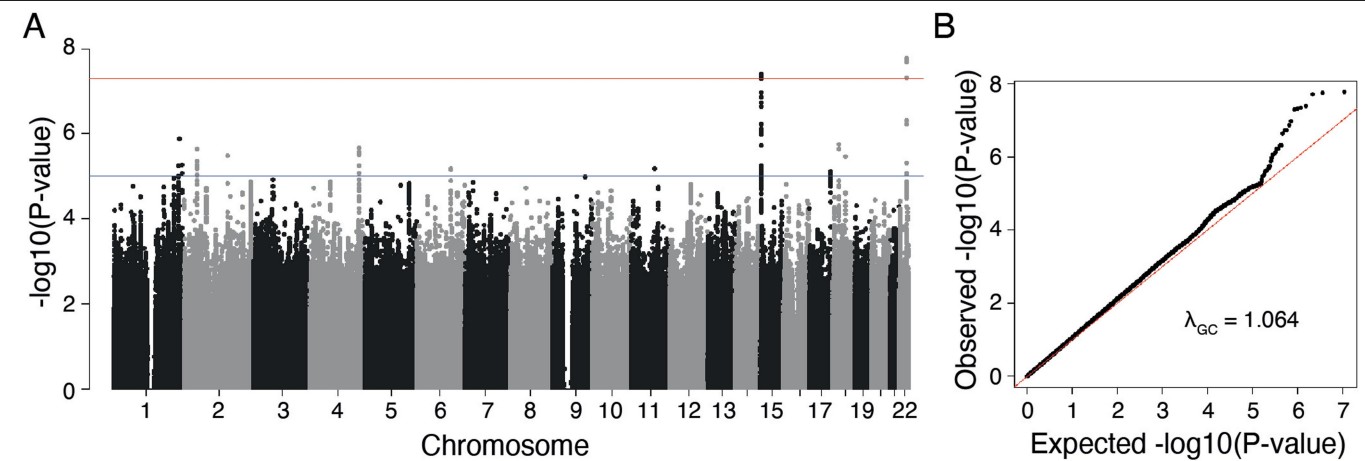

**Extended Data Fig. 4 | GWAS meta-analysis of neurodevelopmental conditions.** We meta-analyzed the GWASs derived from DDD-UKHLS (6,397 cases with neurodevelopmental conditions and 9,270 controls from UKHLS) and GEL (3,618 cases and 13,667 controls). We used overlapping SNPs with MAF > 1% in both cohorts. (**A**) Manhattan plot. The red line indicates the genome-wide significance threshold ($5×10^{-8}$). (**B**) Quantile-quantile plot. GWAS summary statistics including exact *P* values are available in Supplementary Data 3.

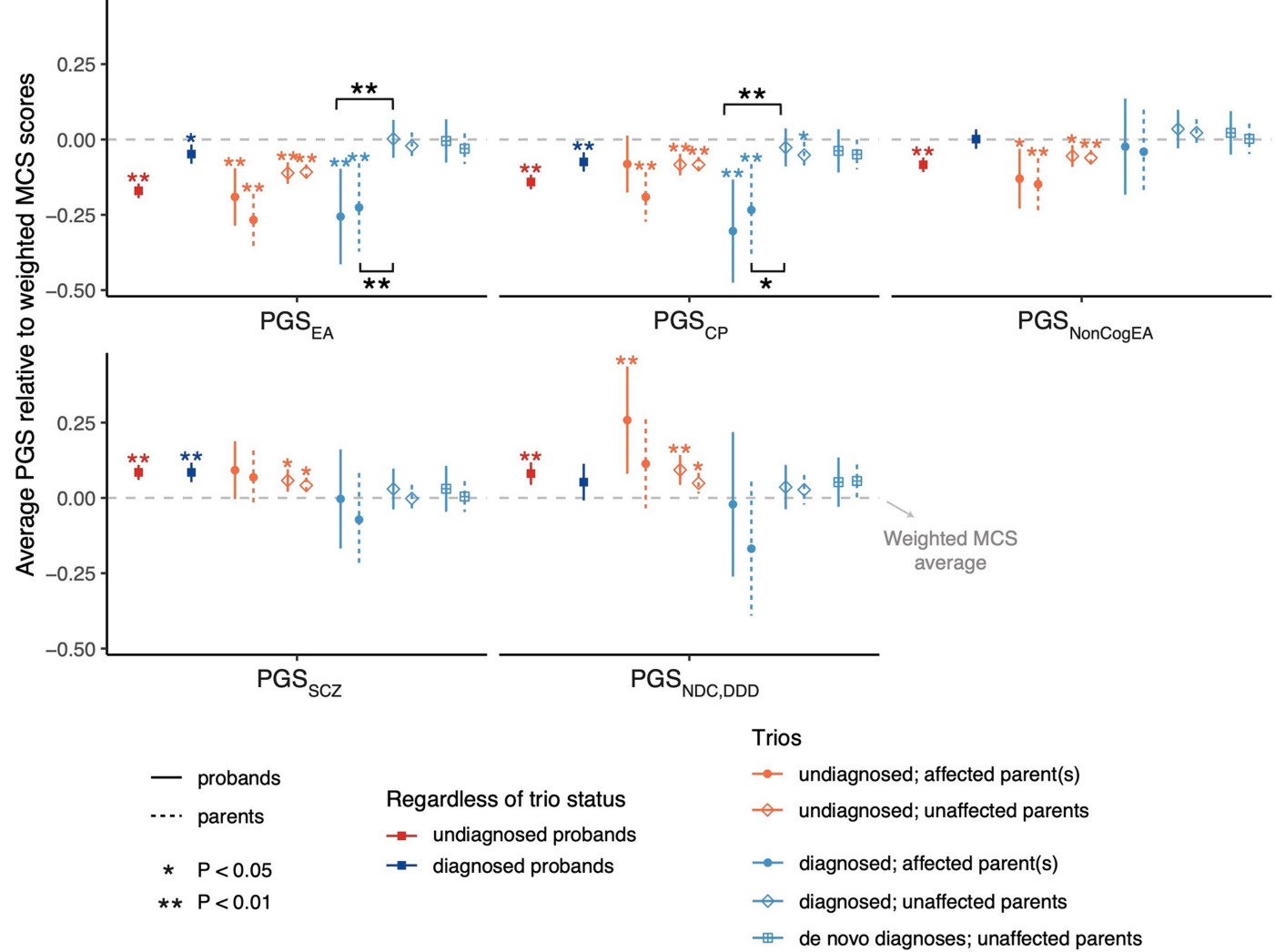

**Extended Data Fig. 5 | Average polygenic scores in undiagnosed (red) and diagnosed (blue) probands with neurodevelopmental conditions from DDD and GEL combined.** PGSs were standardized so that, after reweighing to adjust for sampling and non-response bias, MCS children had mean of 0 and s.d. of 1 (see Methods and Supplementary Note 4). Subsets of probands with neurodevelopmental conditions and their parents from trios are shown in light red (undiagnosed subsets) and light blue (diagnosed subsets). PGS$_{NDC,DDD}$ was tested in a held-out set of patients in DDD that were not included in the original GWAS as well as in GEL. Error bars show 95% confidence intervals. Asterisks in blue or red indicate subgroups that showed significantly different PGS compared to weighted MCS control children indicated by the horizontal line. Black asterisks indicate significant differences in average PGS between two subgroups highlighted by brackets which are specifically mentioned in the main text. One asterisk indicates nominally significant differences ($P < 0.05$) and a double asterisk indicates significant differences that passed Bonferroni correction for five PGSs ($P < 0.01$). See also Supplementary Table 7 for results of two-sided $t$-tests comparing groups.

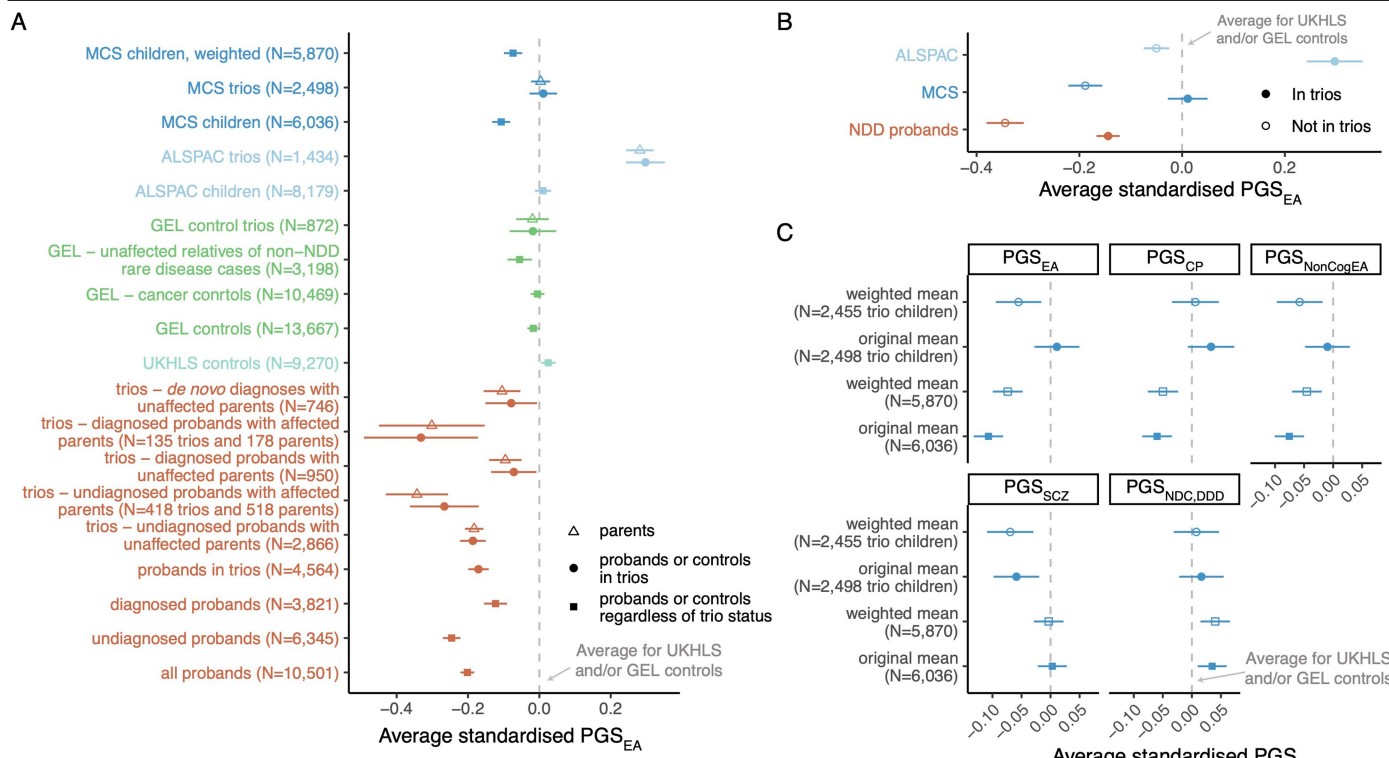

**Extended Data Fig. 6 | Average polygenic scores in various subgroups.**
**A**) Average polygenic score for educational attainment (PGS$_{EA}$) in different control cohorts and subsets thereof, subsets of probands with neurodevelopmental conditions, and their unaffected parents. **B**) Comparing average PGS$_{EA}$ in trio probands and probands who did not have genetic data on both parents in ALSPAC, MCS, and affected patients from DDD and GEL. Note that in the case of DDD, "in trios" refers to those who had *exome sequence* data on both parents (only a subset of which also had genotype array data, since we prioritized genotyping full trios for which the child was undiagnosed), whereas

in the rest of the manuscript (except for Fig. 2b which uses the same definition as here), "trio proband" refers to those who had *genotype* data on both parents. **C**) Average polygenic scores for all five traits in MCS before and after reweighting to adjust for sampling bias and attrition. Note that the PGS are corrected for 20 PCs and then normalized so that a combined set of unrelated controls from UKHLS and GEL have mean of 0 and s.d. of 1. Error bars show 95% confidence intervals. See Supplementary Table 6 for results of two-sided *t*-tests comparing the various groups.

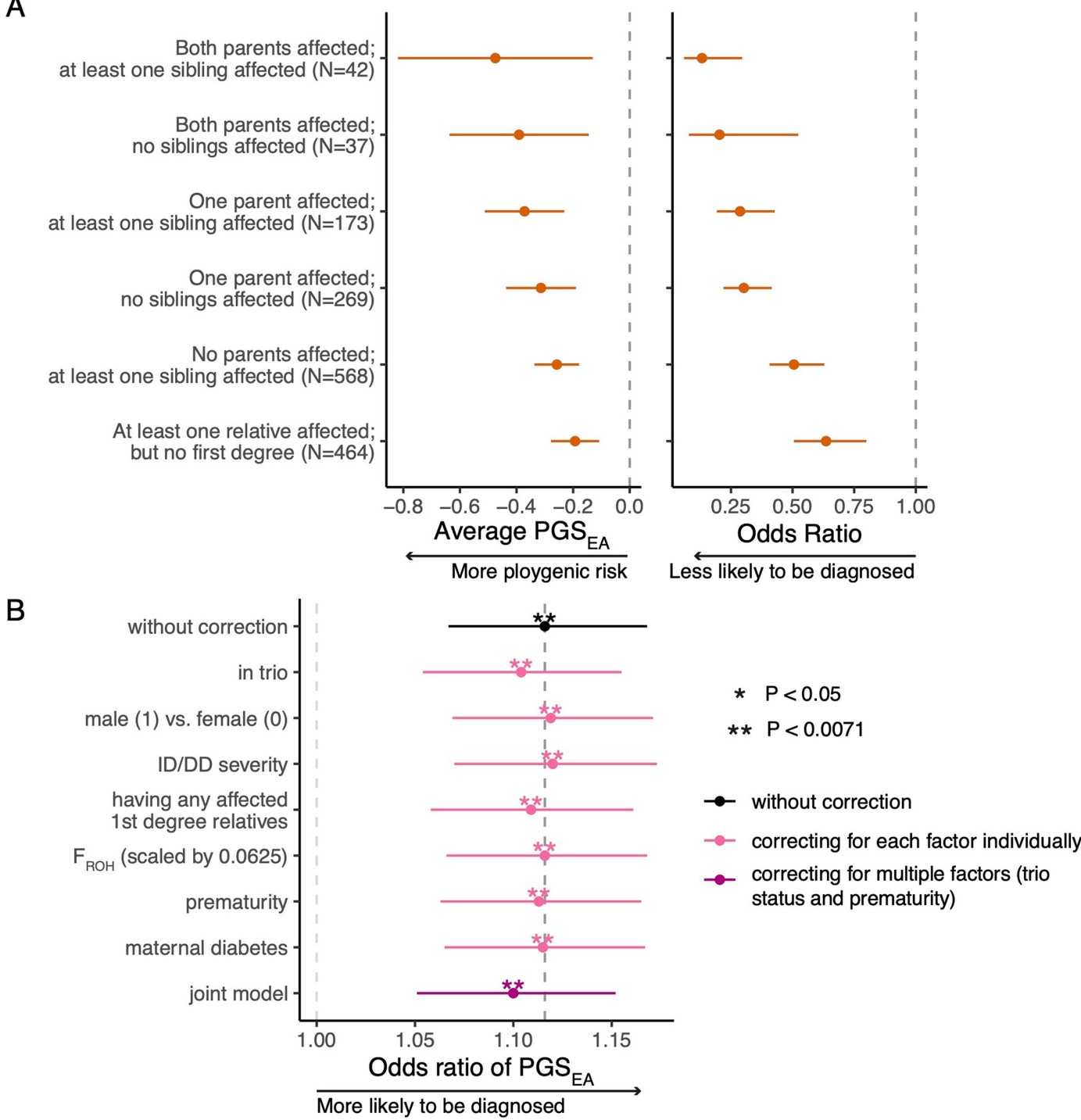

**Extended Data Fig. 7 | Factors associated with having a monogenic diagnosis in DDD.** (**A**) Association between different configurations of affected relatives and the child's $PGS_{EA}$ (left) or the odds of getting a monogenic diagnosis (right). Left: Average proband $PGS_{EA}$ in subgroups with different configurations of affected relatives based on the number of affected parents, siblings, and more distant relatives. Right: Odds ratio for having a monogenic diagnosis, compared to probands with no affected relatives, estimated from logistic regression. See Supplementary Methods for a description of how this was calculated. (**B**) Association between proband's $PGS_{EA}$ and diagnostic status, with or without correcting for technical, clinical and prenatal factors that are associated with receiving a monogenic diagnosis in DDD, assessed via logistic regression.

We corrected for each factor individually (light purple), and also corrected both trio status and prematurity in a joint model (dark purple). In the joint model, we did not include factors that were not associated with $PGS_{EA}$ (sex and maternal diabetes) or diagnostic status ($F_{ROH}$) (Fig. 2), nor factors that are likely the consequence of having or not having a monogenic diagnosis, rather than a cause of getting a diagnosis (severity of ID/DD or having any affected family members). One asterisk indicates nominally significant results ($P < 0.05$) and double asterisk indicates significant results that passed Bonferroni correction for seven factors. See Supplementary Table 8 for exact estimates and two-sided $P$ values. Error bars show 95% confidence intervals in both panels.

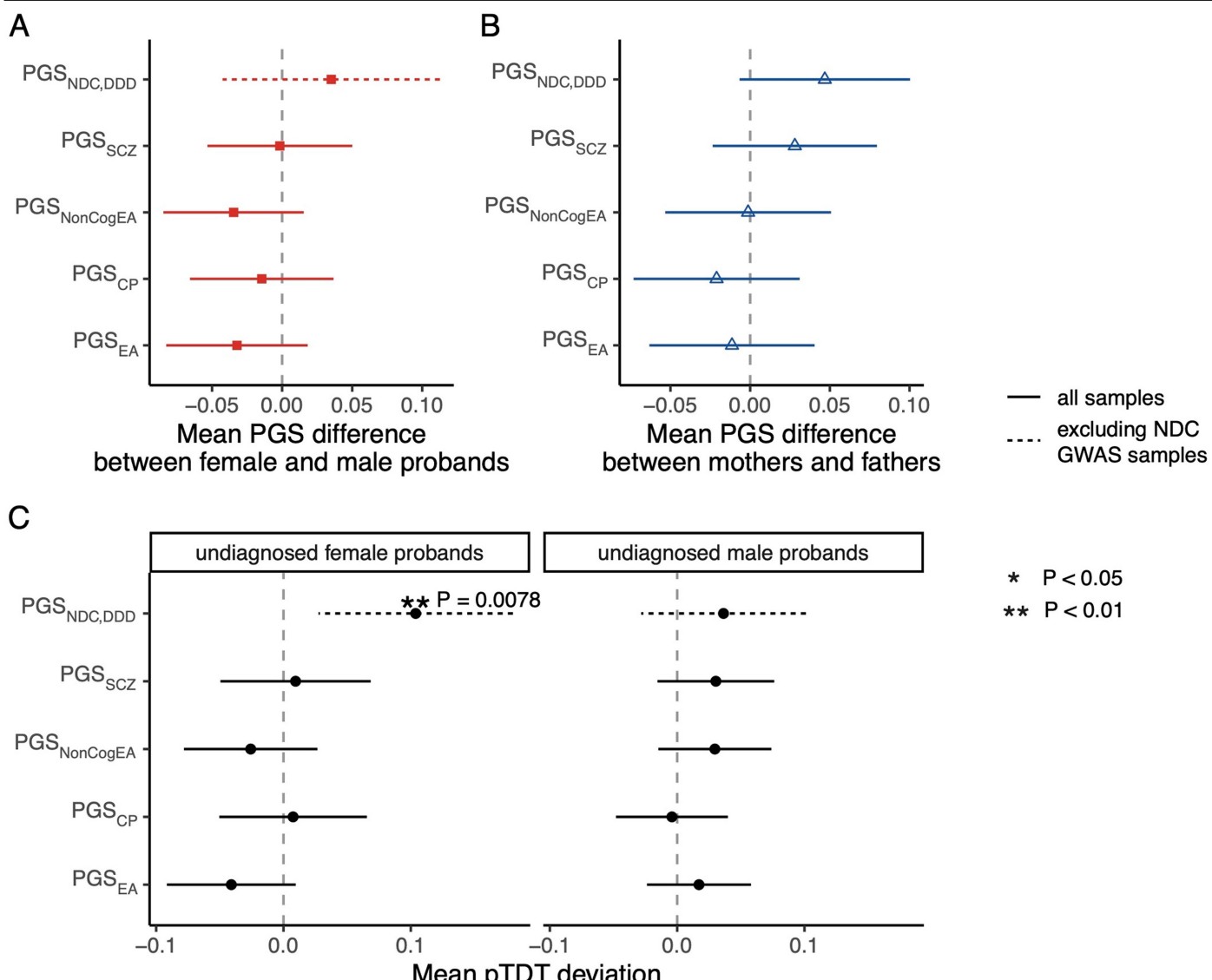

**Extended Data Fig. 8 | Exploring sex differences in polygenic risk.**
**A**) Comparison of polygenic scores between undiagnosed male and female probands in DDD and GEL combined. We used all undiagnosed probands with neurodevelopmental conditions regardless of trio status in this analysis ($N = 1,426$ females and $N = 2,427$ males in DDD; $N = 112$ females and $N = 146$ males in DDD excluding GWAS samples; $N = 918$ females and $N = 1,574$ males in GEL). Square points show the differences in average polygenic scores between female and male probands. A positive difference indicates that female probands have higher PGS than male probands. **B**) Comparison of polygenic scores between unaffected mothers and fathers of undiagnosed probands from a combined sample of 1,523 trios and 1,343 trios from DDD and GEL, respectively. Triangles show the differences in average polygenic scores

between mothers and fathers. A positive difference indicates that mothers have higher PGS than fathers. Two-sided $t$-tests were used to compare average PGSs in **A**) and **B**). **C**) pTDT results in undiagnosed female and male probands with unaffected parents ($N = 586$ females and $N = 937$ males in DDD; $N = 99$ females and $N = 125$ males in DDD excluding GWAS samples; $N = 490$ females and $N = 853$ males in GEL). We tested if probands' polygenic scores deviated from the mean parental polygenic scores using two-sided one-sample $t$-tests. Points show the mean pTDT deviation (difference between the child's polygenic score and the mean parental polygenic score, in units of the s.d. of the latter). Error bars show 95% confidence intervals. The significant result that passes Bonferroni correction of five tests is highlighted by a double asterisk. See Supplementary Table 9 for results of pTDT.

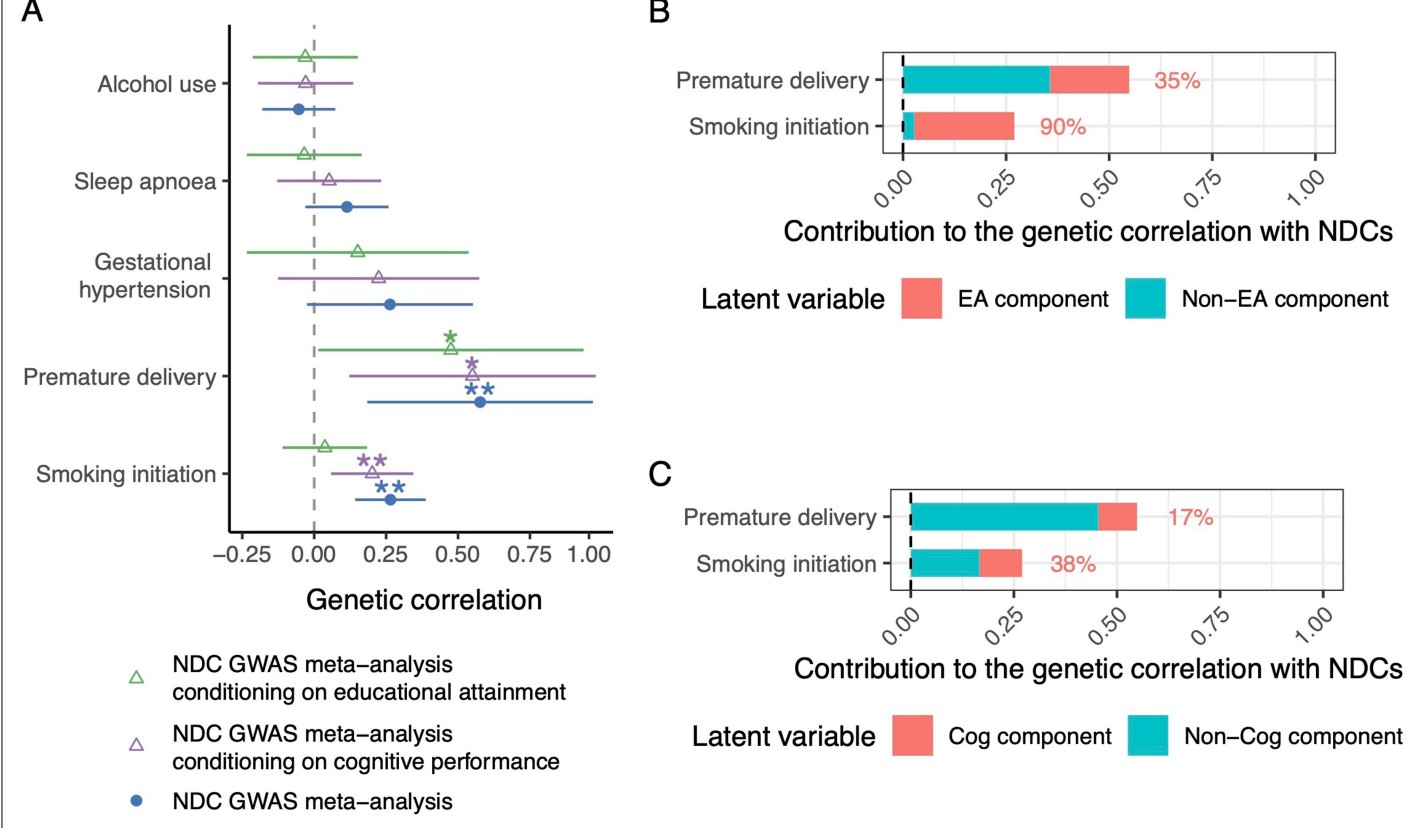

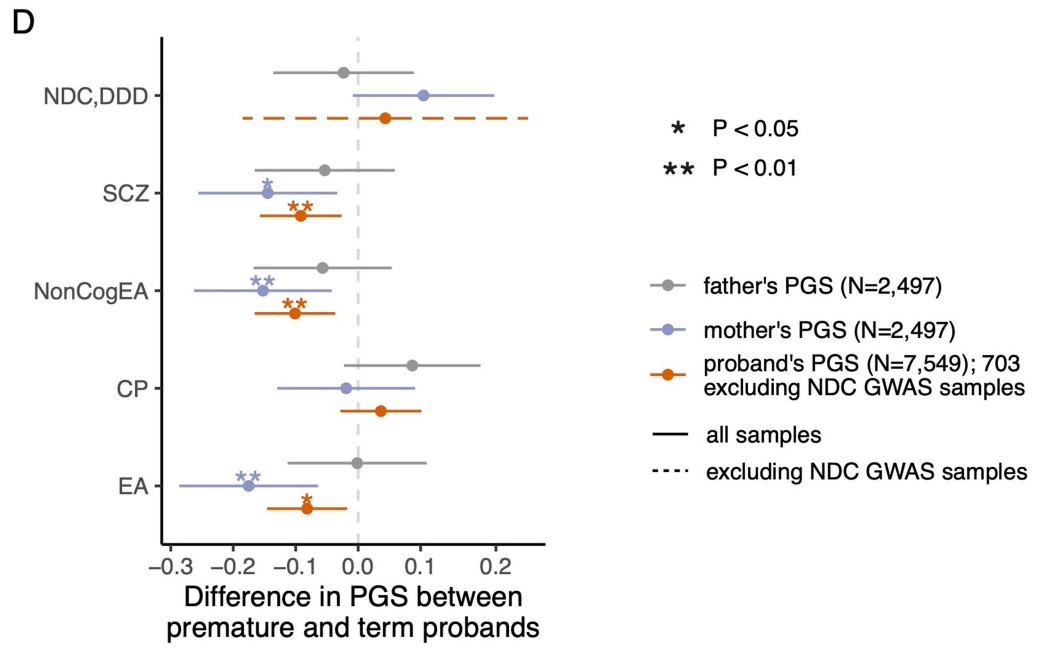

**Extended Data Fig. 9** | See next page for caption.

**Extended Data Fig. 9 | Exploring prenatal factors that may influence risk of neurodevelopmental conditions.** (**A**) Points show genetic correlations between neurodevelopmental conditions and prenatal risk factors, before and after conditioning on educational attainment or cognitive performance. Genetic correlations with our GWAS meta-analysis for neurodevelopmental conditions were estimated using Linkage Disequilibrium Score Regression. Those conditioned on the GWAS summary statistics for educational attainment or cognitive performance were estimated using GenomicSEM. See Supplementary Table 11 for exact estimates of genetic correlations and two-sided $P$ values. (**B**) Percentage of the genetic correlation between neurodevelopmental conditions and prenatal risk factors that is explained by the latent educational attainment (EA) variable estimated using GenomicSEM (red bars and percentage written in text). Green bars indicate the contribution from the non-EA latent variable. The estimates are standardized so that the total height represents the genetic correlation between neurodevelopmental conditions and prenatal risk factors. (**C**) Percentage of the genetic correlation between neurodevelopmental conditions and prenatal risk factors that is explained by the latent cognitive (Cog) variable (red bars and percentage written in text). Green bars indicate the contribution from the non-cognitive (Non-Cog) variable. In (B) and (C), we focused on prenatal factors that showed significant genetic correlations with neurodevelopmental conditions. (**D**) Association between PGSs and prematurity, a risk factor for neurodevelopmental conditions. Points show the differences in PGSs between premature and term probands, estimated in DDD using linear regression models. See Supplementary Table 8 for exact two-sided $P$ values and sample sizes. Note that for $PGS_{NDC,DDD}$, probands who were included in the GWAS were not tested, which left 703 probands, of which 83 were born prematurely. A negative estimate indicates that probands who were born prematurely had a lower polygenic score than term probands, or their parents had a lower polygenic score than the parents of term probands. Associations that pass Bonferroni correction for five traits in (A) or five polygenic scores in (B) are indicated by a double asterisk and nominally significant ($P < 0.05$) results by one asterisk. Error bars show 95% confidence intervals.

**A**

| Family | | Control trios | | | NDC trios | | |
|---|---|---|---|---|---|---|---|
| | | No. 1 | No. 2 | No. 3 | No. 4 | No. 5 | No. 6 |
| Mother (T+NT) | PGS | 3 | 5 | 6 | 18 | 20 | 21 |
| | RVBS | 0 | 2 | 1 | 6 | 7 | 10 |
| | Total risk | 3 | 7 | 7 | 24 | 27 | 31 |
| Father (T+NT) | PGS | 5 | 2 | 4 | 20 | 26 | 20 |
| | RVBS | 1 | 0 | 0 | 6 | 7 | 7 |
| | Total risk | 6 | 2 | 4 | 26 | 33 | 27 |
| Mother (NT) | PGS | 1 | 3 | 3 | 8 | 10 | 11 |
| | RVBS | 0 | 1 | 1 | 2 | 3 | 4 |
| Father (NT) | PGS | 3 | 1 | 2 | 10 | 14 | 11 |
| | RVBS | 0 | 0 | 0 | 3 | 3 | 4 |
| Child | PGS (T) | 4 | 3 | 5 | 20 | 22 | 19 |
| | RVBS (T) | 1 | 1 | 0 | 7 | 8 | 9 |
| | Total risk | 5 | 4 | 5 | 27 | 30 | 28 |
| | PGS deviation | 0 | -0.5 | 0 | 1 | -1 | -1.5 |
| | RVBS deviation | 0.5 | 0 | -0.5 | 1 | 1 | 0.5 |

Mother (T+NT): $r = 0.96$, $p = 0.002$
Father (T+NT): $r = 0.98$, $p = 0.0006$
Mother (NT): $r = 0.97$, $p = 0.001$
Father (NT): $r = 0.93$, $p = 0.007$
Child PGS(T)/RVBS(T): $r = 0.96$, $p = 0.002$
Child deviation: $r = 0.03$, $p = 0.85$

$$1_{NDC\,status} \sim \delta \times PGS_{child} + \theta_{m,NT} \times PGS_{mother} + \theta_{f,NT} \times PGS_{father}$$

$$1_{NDC\,status} \sim \theta_T \times PGS_{child}$$

**B**

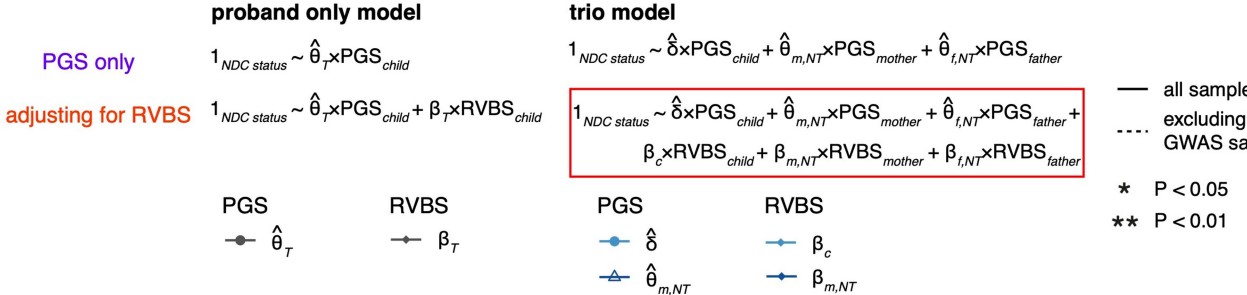

Panels: $PGS_{EA}$, $PGS_{CP}$, $PGS_{NonCogEA}$, $PGS_{SCZ}$, $PGS_{NDC,DDD}$

y-axis: effect size

x-axis categories: proband only model, proband only model + RVBS, trio model, trio model + RVBS

**proband only model**

PGS only: $1_{NDC\,status} \sim \hat{\theta}_T \times PGS_{child}$

adjusting for RVBS: $1_{NDC\,status} \sim \hat{\theta}_T \times PGS_{child} + \beta_T \times RVBS_{child}$

PGS: $\hat{\theta}_T$  RVBS: $\beta_T$

**trio model**

$$1_{NDC\,status} \sim \hat{\delta} \times PGS_{child} + \hat{\theta}_{m,NT} \times PGS_{mother} + \hat{\theta}_{f,NT} \times PGS_{father}$$

$$1_{NDC\,status} \sim \hat{\delta} \times PGS_{child} + \hat{\theta}_{m,NT} \times PGS_{mother} + \hat{\theta}_{f,NT} \times PGS_{father} + \beta_c \times RVBS_{child} + \beta_{m,NT} \times RVBS_{mother} + \beta_{f,NT} \times RVBS_{father}$$

PGS: $\hat{\delta}$, $\hat{\theta}_{m,NT}$, $\hat{\theta}_{f,NT}$  RVBS: $\beta_c$, $\beta_{m,NT}$, $\beta_{f,NT}$

— all samples
--- excluding NDC GWAS samples

\* $P < 0.05$
\*\* $P < 0.01$

**Extended Data Fig. 10** | See next page for caption.

**Extended Data Fig. 10 | Exploring how the correlation between rare and common variant components of risk for NDCs affects estimates from the trio model.** (**A**) Illustration of how assortment-induced correlation between common and rare components of risk for neurodevelopmental conditions affects the non-transmitted coefficients but not the estimate of the direct genetic effect in the trio model. We simulated three NDC trios and three control trios. Each individual has a polygenic score (PGS) and a rare variant burden score (RVBS), representing the measured common and rare variant risk for NDCs, respectively. The child in each trio family has inherited about the expected number of risk alleles (the average of their parents) - the transmitted alleles (T). In these simulated hypothetical families, the child does not show significant deviation from parental average, which is what we observe for $PGS_{EA}$ (Fig. 3). We also show the PGS and RVBS derived from the parental non-transmitted risk alleles (NT). An individual's PGS is correlated with their RVBS (black double arrows) due to parental assortment which started in previous generations (Extended Data Fig. 2b). However, in these hypothetical families, the child's PGS deviation from their parental average is not significantly correlated with their RVBS deviation (grey double arrows). In the 'proband-only model', $\theta_T$ captures both the association between child's PGS and NDC risk and the association between child's RVBS and NDC risk (blue solid arrow) due to the correlation between child's PGS and RVBS. In the 'trio model', the parental non-transmitted coefficients ($\theta_{m,NT}$, $\theta_{f,NT}$) capture the effects of both the parental PGS and RVBS (purple solid arrows) for the same reason. However, the coefficient on the child's PGS (the estimate of the direct genetic effect, $\delta$) captures the association of the deviation from parental average PGS due to Mendelian segregation (orange solid arrow), which is uncorrelated with the rare variant effects. Note that the values for PGS and RVBS have been chosen deliberately to emphasize the point for illustrative purposes, but real correlations between the measured scores are much weaker (Fig. 5). We used simulated counts to calculate Pearson correlation coefficients and reported two-sided $P$ values. (**B**) Effect sizes of PGS and RVBS on case/control status within GEL estimated from the 'proband-only' and 'trio' models. Two-sided $P$ values and effect sizes (reported in Supplementary Table 10) were estimated from logistic regression models fitted to 1,343 trios in which the proband with a neurodevelopmental condition is undiagnosed and parents are unaffected, and 872 trios without neurodevelopmental conditions. Case/control status was regressed on either the child's PGS (proband-only model), the child's PGS and child's RVBS (proband-only model + RVBS), all three trio members' PGSs (trio model), or all three trio members' PGSs and RVBSs (trio model+RVBS). We have indicated results from the latter with a red box, since they are the main focus of this figure. One asterisk indicates nominally significant results ($P < 0.05$) and a double asterisk indicates significant results that passed Bonferroni correction for five PGSs. Note that the 'proband-only' model and 'trio' model were also shown in Fig. 4 using additional cases and controls, rather than just GEL. The RVBS was defined as the number of rare damaging PTVs and missense variants in constrained genes (excluding de novo mutations in the child), corrected for 20 genetic principal components.

# Reporting Summary

## Statistics

For all statistical analyses, confirm that the following items are present in the figure legend, table legend, main text, or Methods section.

| n/a | Confirmed | |
|---|---|---|
| ☐ | ☒ | The exact sample size (*n*) for each experimental group/condition, given as a discrete number and unit of measurement |
| ☒ | ☐ | A statement on whether measurements were taken from distinct samples or whether the same sample was measured repeatedly |
| ☐ | ☒ | The statistical test(s) used AND whether they are one- or two-sided<br>*Only common tests should be described solely by name; describe more complex techniques in the Methods section.* |
| ☐ | ☒ | A description of all covariates tested |
| ☐ | ☒ | A description of any assumptions or corrections, such as tests of normality and adjustment for multiple comparisons |
| ☐ | ☒ | A full description of the statistical parameters including central tendency (e.g. means) or other basic estimates (e.g. regression coefficient) AND variation (e.g. standard deviation) or associated estimates of uncertainty (e.g. confidence intervals) |
| ☐ | ☒ | For null hypothesis testing, the test statistic (e.g. *F*, *t*, *r*) with confidence intervals, effect sizes, degrees of freedom and *P* value noted<br>*Give P values as exact values whenever suitable.* |
| ☒ | ☐ | For Bayesian analysis, information on the choice of priors and Markov chain Monte Carlo settings |
| ☒ | ☐ | For hierarchical and complex designs, identification of the appropriate level for tests and full reporting of outcomes |
| ☐ | ☒ | Estimates of effect sizes (e.g. Cohen's *d*, Pearson's *r*), indicating how they were calculated |

*Our web collection on statistics for biologists contains articles on many of the points above.*

## Software and code

Policy information about availability of computer code

| Data collection | No software was used for Data Collection. |
|---|---|
| Data analysis | Plink (v1.9) were used to process genotype array data and perform PCA and GWAS. GCTA (v1.94.1) was used to perform projection PCA. KING (v2.2.4) was used to estimate kinship relationships in genotype array samples. The "umap" R package (v0.2.4.1) was used to assign individuals to genetically inferred ancestry groups. Bcftools (v1.16) were used to process sequence data. Plink (v2.0) was used by GEL team to estimate pairwise kinship relationships using the KING robust algorithm. Python (v3.7.0) was used to extract rare variants from sequence data. Hail v0.2.105 was used to perform QC of exome sequence data in birth cohorts. Metal (the version released on 2011-03-25) was used to perform the GWAS meta-analysis. LDpred (v1.0.11) was used to generate SNP weights in PGS. SNP heritability was estimated using LDSC (v1.0.1), GCTA (v1.94.1), and PCGC regression implemented in LDAK (v5.2). LDSC (v1.0.1) and GenomicSEM (0.0.5c) were used to estimate genetic correlations. Remaining analyses were performed in R (4.0.2). |

For manuscripts utilizing custom algorithms or software that are central to the research but not yet described in published literature, software must be made available to editors and reviewers. We strongly encourage code deposition in a community repository (e.g. GitHub). See the Nature Portfolio guidelines for submitting code & software for further information.

# Data

Policy information about <u>availability of data</u>

All manuscripts must include a <u>data availability statement</u>. This statement should provide the following information, where applicable:

- Accession codes, unique identifiers, or web links for publicly available datasets
- A description of any restrictions on data availability
- For clinical datasets or third party data, please ensure that the statement adheres to our <u>policy</u>

The raw and post-quality control genotype array data and exome sequence data from DDD are available through European Genome-phenome Archive, under EGAS00001000775. Whole-genome sequence data and phenotypic data from the 100,000 Genomes project can be accessed by application to Genomics England (https://www.genomicsengland.co.uk/research/academic/join-gecip). GWAS summary statistics of neurodevelopmental conditions generated in this study are available in Supplementary Data. Researchers can apply to access genotype array data from UKHLS (https://www.understandingsociety.ac.uk/documentation/access-data/), ALSPAC (https://www.bristol.ac.uk/alspac/researchers/access/), and MCS (https://cls.ucl.ac.uk/data-access-training/data-access/). Publicly available GWAS summary statistics can be accessed at various resources: http://www.thessgac.org/data, https://pgc.unc.edu/for-researchers/download-results/, and https://egg-consortium.org/Gestational-duration-2023.html. DDG2P genes can be downloaded at https://www.deciphergenomics.org/ddd/ddgenes.

# Research involving human participants, their data, or biological material

Policy information about studies with <u>human participants or human data</u>. See also policy information about <u>sex, gender (identity/presentation), and sexual orientation</u> and <u>race, ethnicity and racism</u>.

| | |
|---|---|
| Reporting on sex and gender | We used biological sex reported by clinicians, participants, or parents of patients. We removed participants when their sex inferred by genetic data is not consistent with the reported sex.<br>Majority of the analyses were performed in both sex combined, and sex was corrected as a covariate when appropriate (e.g. GWAS). We also compared polygenic scores between sexes and performed the pTDT analysis in a sex-specific manner.<br>Sample sizes can be found in the Methods section. |
| Reporting on race, ethnicity, or other socially relevant groupings | We focused on individuals of white British ancestry, which was defined by genetic similarity to British individuals from the 1,000 Genomes Project. Self-reported ethnicity was also available in the MCS cohort, and we further restricted to individuals who self-reported as being of White ethnicity. We use 20 genetic principal components to adjust for remaining fine-scale population structure. |
| Population characteristics | We restricted our analyses to participants with genome-wide genotype or whole-genome sequence data available. Patients have been diagnosed with neurodevelopmental conditions, and about 40% of them had a monogenic diagnosis. Age of onset is <16 years old. Genetic data of both parents of 35% of the DDD patients and 60% of GEL patients were available. Unaffected parent-offspring trios were from two UK birth cohorts: ALSPAC where the children were born between 1991 and 1992, and MCS where the children were born between 2000 and 2001.<br>A more detailed description of each cohort can be found in the "Cohort Descriptions and phenotypes" section in the paper. |
| Recruitment | DDD patients affected by developmental conditions and parents were recruited by clinical geneticists across the UK, between 2011 and 2015. The 100,000 Genomes project recruited rare disease families and cancer patients through NHS. The UKHLS cohort aimed to capture a representative sample of people living in the UK and to collect longitudinal socioeconomic and other data on them. The ALSPAC cohort recruited families in the Avon region of southwest England. The MCS cohort recruited families all over the UK, and children living in disadvantaged areas were intentionally over-sampled.<br>A detailed description of recruitment of cohorts and potential biases inherent in these can be found in Supplementary Note 4. In summary, the estimation of the effect size of polygenic scores (particularly for educational attainment) is sensitive to the choice of controls. For example, the differences in PGS between patients and unaffected controls would be larger if the control cohort is biased towards individuals with higher socio-economic status (SES). Volunteer-based cohorts such as ALSPAC and UKHLS show on average higher SES than the general UK population, while control individuals from GEL, recruited through the National Health Service, have lower SES than ALSPAC and UKHLS. In MCS, individuals from disadvantaged areas were over-sampled. However, the weights we developed to correct for recruitment bias and non-response bias should help to mitigate the biases. |
| Ethics oversight | The DDD study has UK Research Ethics Committee approval (10/H0305/83, granted by the Cambridge South Research Ethics Committee and GEN/284/12, granted by the Republic of Ireland Research Ethics Committee). The 100,000 Genomes project was approved by the East of England—Cambridge Central Research Ethics Committee (REF 20/EE/0035). Ethical approval for ALSPAC was obtained from the ALSPAC Ethics and Law Committee and the Local Research Ethics Committees. Ethical approval for each sweep of MCS was obtained from NHS Research Ethics Committees (MREC). Ethical approval for the sixth MCS sweep - which included the collection of saliva samples from children and biological resident parents - was obtained from London-Central REC (MREC; 13/LO/1786). |

Note that full information on the approval of the study protocol must also be provided in the manuscript.

# Field-specific reporting

Please select the one below that is the best fit for your research. If you are not sure, read the appropriate sections before making your selection.

☒ Life sciences          ☐ Behavioural & social sciences          ☐ Ecological, evolutionary & environmental sciences

For a reference copy of the document with all sections, see <u>nature.com/documents/nr-reporting-summary-flat.pdf</u>

# Life sciences study design

All studies must disclose on these points even when the disclosure is negative.

| | |
|---|---|
| Sample size | The sample size was determined by the maximum subset of unrelated individuals who had both post-quality-control genotype and phenotype data in each cohort. We did not perform a sample size calculation; instead, we used as many samples as were available in the existing rare disease and control cohorts. The sample sizes in our GWAS exceed the minimum sample size recommended by authors of software that we used to estimate SNP heritability: 5000 for LD score regression, 3160 for GCTA-LDMS, and 7000 for PCGC. For some analyses, we are uncertain if our sample size is sufficient to detect smaller effects, as we do not have relevant estimates from previous literature. |
| Data exclusions | We excluded participants who were not identified as having white British ancestry using genetic data. To get unbiased estimates, we excluded one individual from each pair of related individuals (up to third-degree relatives). |
| Replication | We replicated the findings of polygenic signals (more specifically, the association between neurodevelopmental conditions and polygenic score for relevant traits) observed in the DDD cohort in the 100,000 Genomes project. For downstream analysis, we thus combined the two cohorts, or performed a meta-analysis. We did not seek replication in a third rare disease cohort, due to lack of similar cohorts in the UK. However, we did use different control cohorts, such as UK birth cohorts in addition to UKHLS and GEL. Patients consistently show significantly lower polygenic scores for educational attainment compared to all control cohorts (Extended Data Figure 6). Additionally, the parental non-transmitted coefficients for the EA PGS in trio models are significant regardless of the control cohorts used (Supplementary Figure 4), indicating the robust role of common variants in rare neurodevelopmental conditions. |
| Randomization | No randomisation of participants was performed in this study. In association analyses, we controlled for genetic principal components and sex as covariates. |
| Blinding | Blinding was not possible because analysts needed to use the phenotype data in the analysis, or perform analysis in a subset of the participants with a certain characteristic (e.g patients with or without a monogenic diagnosis). |

# Reporting for specific materials, systems and methods

We require information from authors about some types of materials, experimental systems and methods used in many studies. Here, indicate whether each material, system or method listed is relevant to your study. If you are not sure if a list item applies to your research, read the appropriate section before selecting a response.

## Materials & experimental systems

| n/a | Involved in the study |
|---|---|
| ☒ | ☐ Antibodies |
| ☒ | ☐ Eukaryotic cell lines |
| ☒ | ☐ Palaeontology and archaeology |
| ☒ | ☐ Animals and other organisms |
| ☒ | ☐ Clinical data |
| ☒ | ☐ Dual use research of concern |
| ☒ | ☐ Plants |

## Methods

| n/a | Involved in the study |
|---|---|
| ☒ | ☐ ChIP-seq |
| ☒ | ☐ Flow cytometry |
| ☒ | ☐ MRI-based neuroimaging |

## Plants

| | |
|---|---|
| Seed stocks | *Report on the source of all seed stocks or other plant material used. If applicable, state the seed stock centre and catalogue number. If plant specimens were collected from the field, describe the collection location, date and sampling procedures.* |
| Novel plant genotypes | *Describe the methods by which all novel plant genotypes were produced. This includes those generated by transgenic approaches, gene editing, chemical/radiation-based mutagenesis and hybridization. For transgenic lines, describe the transformation method, the number of independent lines analyzed and the generation upon which experiments were performed. For gene-edited lines, describe the editor used, the endogenous sequence targeted for editing, the targeting guide RNA sequence (if applicable) and how the editor was applied.* |
| Authentication | *Describe any authentication procedures for each seed stock used or novel genotype generated. Describe any experiments used to assess the effect of a mutation and, where applicable, how potential secondary effects (e.g. second site T-DNA insertions, mosiacism, off-target gene editing) were examined.* |

