## [Peer Review File · Nature]

Examining the role of common variants in rare neurodevelopmental conditions

Corresponding Author: Dr Qin Qin Huang

Version 1:

Reviewer comments:

Referee #1

(Remarks to the Author)

In this manuscript, Huang, Wigdor, and colleagues set out to test the contributions of common variants to rare neurodevelopmental disorders. The authors embark on analyzing different themes associated with common variant associations, many of which were previously tested on other cohorts with reduced power.

The manuscript is well-written, outlining the rationale for the analysis and interpreting the observations while citing relevant previous work. It was very thoughtful of the authors to include an FAQ section.

The central hypothesis that common variants modulate the penetrance of neurodevelopmental disorders appears at a couple of locations in the manuscript, but the authors do not formally estimate the effect of common variants on penetrance.

The GWAS meta-analysis was successful with improved power and showed several genetic correlations of neurodevelopmental conditions with other related disorders. Analysis of patients with monogenic disease showed a reduced polygenic risk, in line with the liability threshold model. It is refreshing to see that the authors accounted for additional confounders (Fig. 2) in the analysis. It would be helpful if additional details regarding the mechanistic nature of this inverse correlation, i.e., what genes and pathways affected by rare and common variants are at interplay, are explored.

The authors test if probands without a monogenic diagnosis inherit a higher polygenic burden from unaffected parents but find only modest significance for overtransmission for neurodevelopmental conditions. It is not clear how the PGS was calculated for the broader neurodevelopmental condition (PGSND, DDD), and if clinical heterogeneity (number and severity of comorbid conditions) would affect the PGS calculations. Especially, grouping patients with intellectual disability with those with epilepsy, abnormal metabolism, and encephalopathy may not be ideal. This may cause difficulties in finding genotype-phenotype relevance for specific disease outcomes, especially in a clinical setting. It would be helpful if the analysis is performed based on developmental domains, such as IQ, SRS, etc.

The most interesting part of this paper comes from the analysis of non-transmitted alleles. The significant contribution of non-transmitted common alleles in parents on undiagnosed patients is interesting. The description of how the indirect effects were calculated was somewhat unclear. It would be helpful to expand on the description in the results and in the methods section. The authors investigate the effect of non-transmitted alleles on prenatal factors and unfortunately do not find significant evidence. This part begs for more analysis. Especially, it would be really interesting to know the indirect effects of non-transmitted alleles in terms of family-specific environment and other non-genetic factors. The authors did include assortative mating in their analysis, highlighting correlations between rare and common variants – aspects expected under assortative mating but not tested in previous studies.

Another comment lies with the use of the term “rare neurodevelopmental conditions,” which may imply “rare diseases” that are actually extremely rare (e.g., Prader-Willi, Noonan, Kleefstra syndromes). Unless families in the DDD cohort were ascertained for severe and very specific rare neurodevelopmental conditions (or for the presentation of idiopathic ID/DD and congenital anomalies), it would be fine to drop “rare” from describing the cohort. Some clarification (or correcting the reviewer’s view) would be helpful.

References 51 and 54 seem to be the same.

Referee #2

(Remarks to the Author)

This paper is a thorough study of the combined effects of rare and common variants for rare neurodevelopmental conditions (NDCs). The authors perform a GWAS of NDCs in a British ancestry (GBR) sample that is greatly expanded relative to their previous work, and they find 2 genome-wide significant loci. They use these summary statistics to estimate the heritability of NDCs and the genetic correlation between NDCs and a variety of other brain related traits and conditions. They assess whether these genetic correlations may be due to common pathways with educational attainment by using genomic SEM and find little differences between the raw genetic correlations and the genetic correlations based on genomic SEM.

In the central analysis of the paper, the authors find that the mean PGS for educational attainment (EA), cognitive performance (CP), and the noncognitive component of EA (NonCogEA) is lower among those without a monogenic diagnosis. They argue that is consistent with a liability threshold model. They evaluate whether this difference between diagnosed and undiagnosed individuals may be driven by other factors (e.g., being in a trio in the data or being born prematurely) but conducting a mediation analysis, and they find that each of the factors they consider each only explain a small fraction of the association between the PGS and diagnostic status. Relatedly, the authors also find that probands without a monogenic diagnosis have a higher PGS than the mean PGS of their parents. In a regression that includes the proband, mother, and father PGS, the child EA PGS is not statistically associated with NDCs but the parents' PGSs are. In contrast, the child NDC PGS is statistically associated with NDCs but not the parents' PGSs. They interpret this as suggesting that there are both direct and indirect effects of genes on NDCs.

Finally, they perform a series of follow-up analyses to understand what might be driving the results of the paper, including an MR analysis of the effect of premature birth on EA and vice versa (acknowledging that a positive result for the latter could not actually be a causal relationship) and tests for assortative mating.

Much about this paper is very interesting and it clearly advances the literature on the various genetic factors that influence genetic associations for NDCs. I have a number of concerns about the paper, however, that I describe below. Many of these concerns center around me having a hard time understanding what was done precisely or not understanding the motivation for the things that were done.

Major comments:

1) My biggest concern is that I struggle to understand the big picture contribution of the work. As far as I can tell, the big point is that common and rare variants both independently have an effect on risk of NDCs. They conduct many other analyses to verify the robustness and to understand the mechanisms driving this result, but most of those analyses are subject to substantial limitations (which the authors acknowledge), like assortative mating, residual population stratification, and core assumptions being implausible in the case of their MR studies. They bring up at several points that their results are consistent with a liability threshold model, which is true, but I'm not sure what models they are trying to rule out by their data. Are there other proposed models with different implications that are inconsistent with their data?

2) Relatedly, their primary designs to understanding the role of rare and common variants is to condition on the presence of rare variants and evaluate the mean of the PGS (and the mean relative to the parental mean in the case of their trio analyses). The reason for this is that collider bias would produce a negative correlation between common variants (and therefore PGSs) that are associated with NDCs and with NDCs. The magnitudes of this approach are a difficult to interpret, and the authors don't really attempt to interpret them. Is there a reason that the authors didn't just regress NDC on the PGS, an indicator for whether a person has a monogenic diagnosis, and the interaction between the two (as they do in the section that employs trios)? It seems like this approach is no worse than their collider approach, but it produces estimates that have a more straightforward interpretation. If they prefer to keep their specification, the authors should probably state what they would expect the estimates to be or provide some other way to interpret the magnitudes.

3) The authors' discussion of the genomicSEM analyses confused me a little, and part of the problem is that I'm not totally certain what the authors have done. After reading the paper and reading up on genomicSEM, I believe that the authors consider a model where there is some latent factor that is the residual of a project of the genetic factor for NDC onto the genetic factor for EA/CP. They estimate the genetic correlation between this residual and a variety of other traits. Because this genetic correlation is not statistically different than the raw genetic correlation between NDC and the corresponding trait, they argue that these correlations are not explained by their relationship with EA. But this seems like the wrong design for their question. Imagine the 99% of the variation in the genetic factor for NDC is explained by the genetic factor for EA and that this portion explained by EA was moderately correlated with ADHD, but the residual was also moderately correlated with ADHD. Then we would see that there would be little change in the rg in the genomicSEM estimate, but it would still be the case that much of the covariance between NDC and ADHD is actually operating through the EA pathway. This extreme case is obviously not actually true, but it hopefully highlights why I think the genomicSEM rg is not getting at the question you are asking. The more appropriate parameter would probably be to use the output of genomicSEM to obtain a parameter more similar to the mediation analysis parameter you estimate in your section on mediation.

4) The authors perform analyses to test whether the differences in PGS means might be due to technical, clinical, or prenatal factors. To do this, they say they perform a test of whether the association between the PGS and a person's diagnostic status mediated by these factors. I wonder if they were doing the analysis backwards though (e.g., whether the association between diagnostic status and trio status is mediated by the PGS). The reason for my confusion is that in the main text, in Figure 2B, and in Supplementary Table S8, they only report how the coefficient on the factor changes by including the PGS rather than how the coefficient on the PGS changes when they add the factor. (In the Table, they also appear to report an effect of the mediator on the PGS rather than the effect of the PGS on the mediator.) The way they describe what they did in Supplementary Note 4 seems more accurate, but it is still not totally clear. Can the authors confirm whether they are actually looking at the mediating effect of their factors on the relationship between the PGS and diagnostic status? If so, they should also probably include all of the relevant estimates to obtain their mediation estimates. It would also be of interest what fraction of the diagnostic status-PGS relationship is mediated by the full set of factors they consider rather than just one at a time.

Minor comments:

5) The authors could be more clear about what comparisons they are making in many places and why. I recognize that this is hard because there are several dimensions of comparisons (e.g., cases vs controls, monogenic diagnosis vs no diagnosis, trio vs non trio, etc.) but I was often confused about what was being done. For example, the authors state, "Despite this, we observed that for all polygenic scores except for PGSNonCogEA, the diagnosed probands still had significantly more polygenic risk than the controls" and they reference Figure 2A. In Figure 2A, I believe the diagnosed probands are the solid blue squares, and every solid blue square is statistically distinguishable from zero (which presumably is the mean of the controls), but some have lower risk (EA, CP, and NonCogEA) and some have higher risk (SCZ and NDC). My guess is that the authors use this language since a low PGS for EA or CP is associated with a higher risk of NDCs and that NonCogEA doesn't qualify because it doesn't pass some Bonferroni-corrected threshold, but the text doesn't clarify this for me and I'm left guessing. I found myself trying to sort this sort of thing out at several points while reading this manuscript.

6) It would be helpful if the authors were clear throughout whether their estimates of h^2 were on the observed scale or the liability scale.

7) In the section on nontransmitted alleles, the authors state, "We next explored one way in which familial polygenic background might affect children's risk of neurodevelopmental conditions, namely indirect genetic effects, i.e. effects of alleles in parents on parental phenotypes that affect their offspring's risk through the family environment." Why is this section framed as being about indirect effects when your data don't allow for the estimation of indirect effects? It seems like all you can actually say is that there are direct effects in some cases and you have a non-zero non-transmitted coefficient in others (which might be indirect effects but it could also be uncontrolled stratification, assortative mating, etc.).

Referee #3

(Remarks to the Author)

This is a remarkable project illuminating the polygenic contribution of common genetic variants to risk of rare neurodevelopmental disorders. The datasets used are some of the largest and best characterised that are available for these analyses. The authors have carried very careful and well thought analyses that convincingly demonstrate that common variants explain ~10% of the variance in risk.

The fact that diagnosed patients have lower polygenic risk compared to undiagnosed ones is intriguing, as is the evidence suggesting that non-transmitted alleles for educational attainment and cognitive ability from parents correlate with children's risk of developing neurodevelopmental conditions. Lower polygenic risk in the results are intriguing and provide further support for liability threshold models that include rare and common genetic effects, alongside potential environmental insults - though the postulated example of Zika infection is impossible to test from DNA sequencing, there might be ways to verify this once epigenetic data is present (if these index exposures).

I liked the FAQ document and thought it was an excellent addition to aid families and clinicians.

Minor points:

LDSC is a lower estimate of heritability, needed here for genetic correlations and as the engine for the SEM analyses. However, the h^2 SNP reported from it is always going to be the lowest of estimates as per what was found in the development of the LDMS approach. I would focus on the GREML-LDMS analyses and I would make secondary the result of heritability for LDSC and PCGC on summary stats. This is especially because you have access to all of the raw sequencing or GWAS data from both cohorts (unlike GWAS consortia). The power of PCGC surprised me in that it approximated GREML-LDMS but nevertheless I recommend focussing on the latter for variance explained.

The results stated in "Probands with monogenic diagnoses have less polygenic risk" should be stated as undiagnosed versus diagnosed to be consistent with the message and title of this section, and not diagnosed versus undiagnosed.

I would be cautious in calling "non-cognitive" the residual from GWAS by subtraction of EA from g as the other components of IQ (the remaining half of the variants) reflect both non-cognitive factors and the other non- g components of cognitive ability.

Point for consideration:

Construction of a multi-polygenic risk score, maximising the combined prediction across all of the selected traits would appear to be a logical extension of the findings and might clarify for the readers what % variance explained can be added by common variants, at present, for individual patients.

Version 2:

Reviewer comments:

Referee #1

(Remarks to the Author)

Thank you again for asking me to review this manuscript from Huang, Martin and colleagues. I thank the authors for their extensive response to my comments on the previous review. I had a few key aspects that I thought needed further attention, thought, and analysis, and I am very happy that the authors have done exactly that in the revision. I am particularly impressed with the thoroughness and due diligence paid to assessing the validity of using PGS for "neurodevelopmental condition", a catch-all term for a heterogeneous condition, and I am surprised (perhaps, as are the authors) that it works well, even when conditioned for different criteria. I am happy that this was done. The next point on a detailed analysis and description of non-transmitted alleles was also well done. The other points regarding the mechanistic relevance and penetrance of common variants were also handled satisfactorily. I feel that that revisions made to reflect these analyses are important and clear.

I also read through the responses to other reviewers' comments, and find that the authors have responded appropriately.

Referee #2

(Remarks to the Author)

The authors have thoroughly responded to each of my concerns from my previous review and made several substantial changes that improved the manuscript. Of note, the organization of the manuscript into aims (including a very clear schematic overview Figure 1) has made the contributions of the work much more clear and helped me as a reader to follow the reasoning behind each of the analyses carried out as part of this paper. I appreciate the time they have taken to do this.

Like the other reviewers, I also really appreciated the FAQ. It is carefully written and I expect that it will be useful to the broader public. If this work published, will the FAQ be provided online as Supplementary Materials, or do the authors plan to post it in some online repository? If possible, I recommend the former since that will likely increase its impact.

My remaining comments are all mostly-minor expositional issues.

1) The authors have made changes to the manuscript in several places to clarify that the association between the non-transmitted parental alleles and the child's outcome is not necessarily due to indirect genetic effects, but several places remain where it is not clear. For example, in the abstract, the authors say that "it is also unclear whether parents' polygenic background contributes to their children's risk beyond the direct effect of variants transmitted to the child." This implies to me that the authors will be able to assess when the parents' polygenic background contributes to their children's risk, but ultimately they cannot. Or in Figure 1, the arrow from the untransmitted alleles to the child's phenotype is labeled as an "indirect genetic effect" when it should be characterized as the non-transmitted coefficient. Also, in line 1145 of the Online Methods, the authors state that θ_{NT} captures the effect of non-transmitted alleles due to genetic nurture. (They subsequently clarify later in the paragraph that other factors could be included, but it seems like they should be more upfront about it.) There are several other places through the text where it's not clear that the authors aren't estimating indirect effects, so it would be good to go through and clean those up.

2) In the section on "Exploring the role of prenatal factors", the authors conduct a Mendelian randomization study. They state "In theory, the genetic correlation between educational attainment and premature delivery could reflect a causal effect of lower educational attainment on premature birth, and/or a causal effect of premature birth on 427 lower educational attainment." They fail to note that a significant MR result could reflect that the SNPs used in the analysis affect some third factor which affects both premature birth and educational attainment. Also both associations could be driven by confounds like population stratification. Indeed, the authors point out that the traditional MR interpretation that educational attainment is affecting premature birth doesn't make sense since the education would have to take place after birth or after diagnosis of a neurodevelopmental condition, concluding that it is "more likely to reflect a causal effect of parents' educational attainment on their children's risk, consistent with the presence of indirect genetic effects." While this is technically true, this evidence is no more evidence of indirect effects than it is evidence of population stratification or other confounds. Ultimately, given that the assumptions of MR are clearly not satisfied, I don't know what is gained from the MR analyses beyond what is learned from the analyses in Supplementary Figure 5 where they show the association between the EA PGS and neurodevelopmental conditions is unchanged when controlling for prematurity or excluding probands who were born prematurely. I'd recommend removing the MR results entirely, but if they authors would like to keep it, they need to argue what additional information is gained from the MR analyses above the more interpretable results that don't imply causality.

Referee #3

(Remarks to the Author)

I am satisfied that they authors have address my points well and thank them for their careful analyses. I believe this paper will be a substantial addition to the literature.

Response

We thank the reviewers for their constructive comments on the manuscript, to which we have responded below. Please note that all new figures mentioned in this document are referred to as Figure R* and are included at the bottom of the document. In the main manuscript file and supplement, new additions are indicated in blue text, and deleted text is indicated with ~~strikethroughs~~.

In addition to responding to the reviewers' comments, we have also added some new analyses to the manuscript using summary statistics from the within-sibship GWAS for educational attainment¹. This GWAS should capture only direct genetic effects on educational attainment that are unconfounded by population structure, assortative mating or indirect genetics effects, but it was much smaller ($N = \sim 55,000$) than the population-based GWAS for EA that we had already included² ($N = \sim 766,000$ excluding 23andMe), so analyses using these within-family GWAS summary statistics are less well powered. Nonetheless, the GWAS summary statistics for EA from this within-sibship study showed significant genetic correlation with NDCs ($r_g = -0.702$, $p = 0.0056$), which was reassuringly similar to that using population-based GWAS ($r_g = -0.654$, $p = 4.94 \times 10^{-12}$). A polygenic score derived from the within-sibship GWAS ($PGS_{EA_{Sib}}$) was significantly associated with NDC case-control status within GEL samples ($p = 0.0015$, variance explained $R^2 = 0.05\%$), but was not significant when comparing DDD patients with UKHLS control individuals ($p = 0.2$). For this reason, we decided not to add results from $PGS_{EA_{Sib}}$ to all our main figures, since it is relatively underpowered, but have instead described results from the key analyses using it in **Supplementary Note 7**. Most importantly, in the trio model, we still observed a significant non-transmitted coefficient in mothers using $PGS_{EA_{Sib}}$ ($p = 3.8 \times 10^{-5}$, $\beta = -0.122$) although not in fathers ($p = 0.14$, $\beta = -0.043$), and we still observed no significant direct genetic effect ($p = 0.77$, $\beta = 0.0099$). The fact that we were able to recapitulate the significant effect of non-transmitted alleles at least in the mother indicates that our original result with PGS_{EA} in the trio model was not due to uncontrolled stratification in the original population-based EA GWAS. We feel that this is an important finding so have included it in the third paragraph of the Discussion and referred to further details about the results obtained with this within-sibship EA GWAS in **Supplementary Note 7**.

Referee #1 (Remarks to the Author):

In this manuscript, Huang, Wigdor, and colleagues set out to test the contributions of common variants to rare neurodevelopmental disorders. The authors embark on analyzing different themes associated with common variant associations, many of which were previously tested on other cohorts with reduced power.

The manuscript is well-written, outlining the rationale for the analysis and interpreting the observations while citing relevant previous work. It was very thoughtful of the authors to include an FAQ section.

We thank the reviewer for these positive comments and are glad that the FAQ was appreciated. To our knowledge, the only analyses that were previously carried out in other

NDC cohorts with reduced power were the heritability and genetic correlation analyses (**Figure 2**) and the test of the liability threshold model (**Figure 3A**). These pertain respectively to the first and second of the three main aims we have now clearly articulated in the Introduction, which are:

1. to better understand the nature of common variant risk for rare neurodevelopmental conditions, particularly its overlap with common variant risk for other brain-related phenotypes
2. to elucidate the interplay between common and rare variants in the context of these conditions
3. to test whether there is an effect of common variants in the parents on their child's risk of these conditions, above and beyond the child's own genetics

However, we have extended upon the analyses carried out in our previous work³ in important ways. Firstly, we have used GenomicSEM to estimate the extent to which the genetic correlations with mental health traits are driven by genetic overlap with educational attainment or cognitive performance (**Supplementary Figure 1**). Secondly, as the reviewer noted below, we have extensively probed whether the association between diagnostic status and polygenic risk could be driven by confounders (**Figure 3B**, **Extended Data Figure 6B**). Furthermore, all the analyses pertaining to the third aim are entirely novel, to our knowledge. Thus, we still feel that there is substantial novelty to this manuscript.

The central hypothesis that common variants modulate the penetrance of neurodevelopmental disorders appears at a couple of locations in the manuscript, but the authors do not formally estimate the effect of common variants on penetrance.

We thank the reviewer for the comment and we would like to clarify our understanding of this query to make sure we understand what the reviewer means by 'the effect of common variants on penetrance of neurodevelopmental disorders'. Our assumption is that they mean, 'what fraction of people in a given quantile of the distribution for common variant risk for NDCs have an NDC?' In this case, we have not addressed this because we believe that doing so would require a large sample of the population which has been recruited in a way that is totally unbiased by cognitive ability/educational attainment, and to our knowledge, this does not exist. This analysis may become possible in future using datasets of genetic data from blood spots taken at birth from all babies in a population, such as is being mooted in Scotland. The iPSYCH data in Denmark is not suitable for this as the subset of individuals who have been genotyped to date includes only those with psychiatric conditions and a small sample of controls. Attempting to estimate penetrance from a sample such as UK Biobank, which is biased towards more highly educated people⁴, would undoubtedly lead to an under-estimate⁵. In contrast, estimates based on cohorts of families recruited for NDCs would over-estimate the penetrance⁶.

Alternatively, perhaps the reviewer is asking about the hypothesis that common variants modify the penetrance of rare variants associated with NDCs. We addressed this in Supplementary Note 6, using a within-family test. Our results were largely negative, which surprised us somewhat given recent conflicting results from UK Biobank, a population sample (albeit a biased one)⁷. However, as discussed in the manuscript, it may simply be that we have limited power due to several potential reasons, one of which is the correlation between rare and common variants associated with NDC predisposition due to parental assortment (see

last sentence of the second paragraph in the Results section entitled “Parental assortment obscures the true nature of common variant effects”).

The GWAS meta-analysis was successful with improved power and showed several genetic correlations of neurodevelopmental conditions with other related disorders. Analysis of patients with monogenic disease showed a reduced polygenic risk, in line with the liability threshold model. It is refreshing to see that the authors accounted for additional confounders (Fig. 2) in the analysis. It would be helpful if additional details regarding the mechanistic nature of this inverse correlation, i.e., what genes and pathways affected by rare and common variants are at interplay, are explored.

We carried out several analyses to explore potential mechanistic overlap between the rare and common variants contributing to NDC risk. We note that recent work has demonstrated that common SNPs associated with educational attainment are enriched around autosomal dominant DDG2P genes⁷, implying that the rare and common components converge on similar pathways. We started by taking a slightly different approach, taking 1,722 genes that had been prioritised with the DEPICT tool (at FDR < 5%) in the educational attainment GWAS we were using² (“EA genes”), and asking whether these showed a greater-than-expected overlap with 788 DDG2P genes responsible for at least one monogenic diagnosis amongst individuals with an NDC in DDD (“diagnostic DDG2P genes”; **Supplementary Methods**). We chose to focus on EA genes since the EA GWAS is strongly (negatively) genetically correlated with the NDC GWAS (**Figure 2**) but is so much more powerful than it; we do not have sufficient power to produce a plausible list of genes in which common variants predispose to NDCs from our own GWAS, given we have only two genome-wide significant loci. We found that the EA genes were significantly enriched amongst diagnostic DDG2P genes compared to all autosomal protein-coding genes (odds ratio = 3.41; p-value = 1.2×10^{-36} ; Fisher’s exact test).

We next examined a collection of gene sets defined based on expression in prenatal or postnatal brain⁸ and genes with cell type-specific expression in certain prenatal brain cell types⁸ (24 gene sets). We tested whether each of these gene sets was enriched for EA genes or for diagnostic DDG2P genes using Fisher’s exact test (**Figure R1AB**). Compared to all remaining genes, both EA genes and diagnostic DDG2P genes showed significant enrichment amongst 7,373 genes preferentially expressed in prenatal brain ($p=3.5 \times 10^{-47}$ and $p=2.9 \times 10^{-31}$ respectively) but only EA genes showed significant enrichment amongst the 6,567 genes preferentially expressed in postnatal brain ($p=1.0 \times 10^{-8}$), which was significantly stronger than the enrichment of diagnostic DDG2P genes (z-score test $p = 8.9 \times 10^{-4}$; **Figure R1A**). EA genes and/or diagnostic DDG2P genes were also significantly enriched amongst genes showing preferential expression in several of the particular cell types in the prenatal brain (**Figure R1B**), but there were some differences. For example, diagnostic DDG2P genes but not EA genes were significantly enriched in several types of neural progenitor and stem cells (NEPRGC2, NEPRGC4), and this difference was significant (z-score test $p < 2.7 \times 10^{-4}$). Conversely, EA genes showed a significantly stronger degree of enrichment in several subtypes of excitatory neurons (ExN1, ExN2, ExN3) than diagnostic DDG2P genes (z-score test $p < 1.3 \times 10^{-6}$).

As an orthogonal approach to explore mechanistic overlap between the common and rare variant components, we applied stratified LD score regression⁹ to the GWAS summary statistics for both NDCs and EA, to test whether the common variant signals were enriched around particular gene sets. For the EA GWAS, SNPs in or near diagnostic DDG2P genes

(8.3% of SNPs) explain 13.0% (95% CI: 11.3-14.8%) of the total common SNP heritability, which is a significant enrichment (1.6-fold, $p=9\times 10^{-8}$), in line with findings from Kingdom et al.⁷. For the NDC GWAS, the SNPs near diagnostic DDG2P genes explain 18.6% (95% CI: 3.4-33.8%) of the SNP heritability, but this is not a significant enrichment (2.3-fold, $p=0.14$), likely due to the low power of this GWAS. We also examined enrichment of SNP heritability in the aforementioned gene sets enriched in particular cell types in the prenatal brain. We focused on the EA GWAS since the NDC GWAS proved underpowered. The pattern of enrichment of SNP heritability across these cell types mimicked what we observed for enrichment of the genes defined by the EA GWAS, as expected, with significant enrichment in excitatory neurons and glial cells relative to other prenatal brain cell types but significant depletion in nascent neurons (NasN) relative to other prenatal brain cell types (**Figure R1CD**).

Taken together, these results suggest that although rare and common variants predisposing to NDCs converge on similar sets of genes and are affecting many of the same cell types, the highly penetrant rare variants (affecting the diagnostic DDG2P genes) may be preferentially affecting very early brain development, whereas common variants may be preferentially affecting brain function slightly later in prenatal development and postnatally. In future, better-powered studies of NDCs will allow us to repeat these analyses using genes/SNPs implicated specifically in NDC risk rather than those ascertained for their associations with EA, as well as genes implicated through less penetrant rare variants, which may reveal different patterns.

We have added these results in detail in **Supplementary Note 8** and **Supplementary Figure 18** and referred to them briefly in the final paragraph of the Discussion: '*Larger GWASs for NDCs will also give us more power to explore the extent to which the common variants impacting these conditions are targeting different pathways and cell types from the rare variants (Supplementary Note 8).*' We did not add them to the main results since the paper is already very long, and we felt that they were a bit of a distraction from the key messages of the paper. The methods are all described in the Supplementary Methods section "Enrichment of gene sets and pathways".

The authors test if probands without a monogenic diagnosis inherit a higher polygenic burden from unaffected parents but find only modest significance for overtransmission for neurodevelopmental conditions. It is not clear how the PGS was calculated for the broader neurodevelopmental condition ($PGS_{NDC, DDD}$), and if clinical heterogeneity (number and severity of comorbid conditions) would affect the PGS calculations. Especially, grouping patients with intellectual disability with those with epilepsy, abnormal metabolism, and encephalopathy may not be ideal. This may cause difficulties in finding genotype-phenotype relevance for specific disease outcomes, especially in a clinical setting. It would be helpful if the analysis is performed based on developmental domains, such as IQ, SRS, etc.

We thank the reviewer for this comment and we agree that phenotype heterogeneity is one of the limitations that might reduce the power for all analyses. To clarify, $PGS_{NDC, DDD}$ was calculated based on the GWAS of all NDC patients in DDD versus UKHLS controls, as mentioned in the Results ("*polygenic scores for neurodevelopmental conditions from the DDD-derived GWAS⁶ ($PGS_{NDC, DDD}$)*") and described in the Methods section "Calculating polygenic scores". As always when trying to maximise power in a GWAS (and hence the accuracy of polygenic scores built from it), there is a tradeoff between the specificity of the phenotype and the sample size.

Unfortunately these cohorts do not have any quantitative scores for metrics such as IQ or SRS that we could incorporate. Additionally, the cohorts are so phenotypically heterogeneous (as shown in Extended Data Figure 2) that it is not feasible to group patients into what anyone would reasonably consider a 'clinical homogenous' subgroups that are sufficiently large to enable powered analyses. However, to address the reviewer's comment, we have rerun the main analyses in the two largest subgroups of patients, to try to reduce the phenotypic heterogeneity somewhat: patients with intellectual disability or developmental delay (ID/DD; 88% of DDD patients and 97% of GEL patients), and patients with seizures (21% of DDD patients and 25% of GEL patients). As we will show below, most of the analyses were underpowered when restricting to patients with seizures, suggesting that further stratification by phenotype is not likely to be powered.

We firstly reran GWASs in DDD and GEL comparing ID/DD patients or epileptic patients to the same control individuals used in the main text, and performed a meta-analysis between the cohorts. As expected given that they were largely overlapping samples, the GWAS meta-analysis of NDCs restricting to ID/DD showed similar SNP heritability estimates (**Table R1**) and genetic correlations (**Figure R2**) to those obtained with all NDC patients, except for the genetic correlation with anorexia nervosa which was no longer nominally significant. However, when restricting to patients with seizures, LDSC SNP heritability estimates were not significant, so we did not calculate genetic correlations for this GWAS meta-analysis. Furthermore, using more powerful methods that utilise individual-level data, PCGC showed a significant heritability estimate in DDD but not in GEL, and GREML-LDMS did not show a significant heritability estimate in either cohort, which is probably due to the small number of cases and limited power (**Table R1**).

To investigate whether and how phenotype heterogeneity in the GWAS cases affects the performance of PGS in predicting case/control status, we constructed a new PGS for NDCs using the DDD-derived GWAS restricting to ID/DD patients ($PGS_{NDC-ID/DD,DDD}$) and tested it in GEL, comparing all NDC cases to controls. $PGS_{NDC-ID/DD,DDD}$ showed significant association with NDC risk within GEL (logistic regression $p=2.5 \times 10^{-6}$), with the explained variance on the liability scale (0.107%) similar to the original $PGS_{NDC,DDD}$ derived from the GWAS of all NDC patients (0.114%, $p=1.1 \times 10^{-6}$). All PGSs tested were significantly associated with NDC risk after restricting cases to patients with ID/DD (**Figure R3**), and the variance explained slightly increased compared to that observed when using all NDC patients, for all comparisons except for PGS_{SCZ} in DDD. After restricting the NDC patients to those with seizures (21% of DDD patients and 25% of GEL patients), $PGS_{NDC,DDD}$ and $PGS_{NDC-ID/DD,DDD}$ were no longer significantly associated with NDC+seizure risk in GEL samples. In addition, PGS_{CP} was no longer significantly different between DDD patients with seizures and UKHLS controls. All PGSs showed expected directions of associations, including the non-significant ones.

We then repeated the analyses from the original Figure 3A, comparing PGSs between patients with and without a monogenic diagnosis. After restricting to patients with ID/DD, results were very similar to those reported in the main text using all NDC patients (**Figure R4**). The differences in average PGSs between diagnosed and undiagnosed patients increased slightly after restricting to ID/DD patients. No differences were significant after restricting to patients with seizures, likely due to reduced sample size. The new $PGS_{NDC-ID/DD,DDD}$ derived from a

GWAS in DDD restricting to ID/DD patients did not show a significant difference between undiagnosed and diagnosed patients, similar to the original $PGS_{NDC,DDD}$.

We then repeated the polygenic transmission disequilibrium tests (pTDT) shown in the original Figure 4A using these subsets of affected individuals. In the subset of NDC trios with ID/DD, results were very similar to the original results obtained using all NDC trios (**Figure R5**). The result with $PGS_{NDC,DDD}$ became slightly stronger (pTDT deviation = 0.069, $p=0.0076$, versus pTDT deviation = 0.062, $p=0.014$ in the original analysis). Similarly, $PGS_{NDC-ID/DD,DDD}$ was significantly over-transmitted from unaffected parents to undiagnosed probands before (pTDT deviation = 0.073, $p=0.0039$) and after restricting to individuals with ID/DD (pTDT deviation = 0.081, $p=0.0018$). We did not observe any significant pTDT deviation in the subset of patients with seizures.

We then repeated the “trio model” in the two subgroups of NDC trios with ID/DD or seizure to assess the effects of proband’s PGS after controlling for parents’ PGS (the direct genetic effect), and whether parental non-transmitted alleles were associated with NDC risk. After restricting cases to NDC trio probands affected by ID/DD, the results largely recapitulated those obtained using all NDC probands (**Figure R6** - compare middle to top row). The new $PGS_{NDC-ID/DD,DDD}$ showed significant evidence for direct genetic effects before ($p=0.0082$, $\beta = 0.11$) and after restricting to subgroups of ID/DD trios ($p=0.0041$, $\beta = 0.12$). Power was limited when comparing NDC trios affected by seizures to control trios, but we still observed some evidence for a significant non-transmitted coefficient of the maternal PGS for traits that were related to NDCs (bottom plot; **Figure R6**).

Finally, we repeated the analyses of correlations between PGSs and rare variant burden in constrained genes shown in the original Figure 6. We observed similar significant correlations in the subgroup of patients affected by ID/DD to those observed in the full set of NDC probands (compare middle to left-hand panel; **Figure R7A**). In the smaller subgroup of patients affected by seizures (946 trios), we observed only two nominally significant correlations for PGS_{EA} and $PGS_{NonCogEA}$ within the parents (right-hand panel of **Figure R7A**). As for $PGS_{NDC,DDD}$, we did not observe significant correlations between rare variants and $PGS_{NDC-ID/DD,DDD}$. Using rare synonymous variants in constrained genes as the negative control, we did not see significant correlations with any PGSs in subgroups of NDC patients (**Figure R7B**).

In conclusion, all our major findings remain significant in patients with ID/DD, which is the largest subgroup of patients. Some results are even stronger in this subset (e.g. slightly higher variance explained by PGSs, slightly bigger difference in average PGSs between patients with and without a monogenic diagnosis, and slightly more significant pTDT deviation for $PGS_{NDC,DDD}$). The results remain robust regardless of whether the PGS for NDCs was constructed using a GWAS including all available NDC patients, or a GWAS comparing the subset of patients with ID/DD to controls. We do not feel that these changes are sufficiently notable to merit replacing all results in the manuscript using those obtained using the subset of patients with ID/DD, and we feel that the analysis of patients with seizures is underpowered and probably not worth including. However, if the reviewer or editor feels strongly, we would happily incorporate them into the manuscript. It would be interesting to assess further by developmental domains in future studies with appropriate phenotype data. We have added the following sentence to the second-last paragraph of the Discussion to that effect: “*In future,*

larger cohorts with quantitative phenotype data (e.g. on IQ or social responsiveness) may allow us to revisit these questions while subsetting to reduce phenotypic heterogeneity.”

The most interesting part of this paper comes from the analysis of non-transmitted alleles. The significant contribution of non-transmitted common alleles in parents on undiagnosed patients is interesting. The description of how the indirect effects were calculated was somewhat unclear. It would be helpful to expand on the description in the results and in the methods section. The authors investigate the effect of non-transmitted alleles on prenatal factors and unfortunately do not find significant evidence. This part begs for more analysis. Especially, it would be really interesting to know the indirect effects of non-transmitted alleles in terms of family-specific environment and other non-genetic factors. The authors did include assortative mating in their analysis, highlighting correlations between rare and common variants – aspects expected under assortative mating but not tested in previous studies.

We have substantially modified the Methods section on “Trio model to assess effects of transmitted versus non-transmitted alleles” to better explain the methodology used, which was first developed by others, whom we cite. In the last sentence of the first paragraph of the Results section on “Non-transmitted common alleles in unaffected parents are associated with their children’s risk”, we have now clearly referred to this specific Methods section, as well to Figure 5, in which we now show the ‘proband only model’ and ‘trio models’ in mathematical notation, with parameters defined in the legend. We have also made some changes to this Results paragraph to emphasise that what we are really estimating is the association with non-transmitted alleles rather than the indirect genetic effects per se, since the non-transmitted coefficients don’t only reflect indirect genetic effects but also confounders, as noted by Reviewer 2 in their final comment. We expand on this below.

We agree with the reviewer that this is the most interesting part of the paper, and that understanding the cause of these associations between the non-transmitted alleles and NDCs is an important and fascinating question. However, as far as we can see, it is not possible to do this effectively with the available data (or any data that is likely to become available any time soon), as explained below.

As discussed in the paper, the associations between non-transmitted alleles and NDC case/control status could be driven by several factors (not mutually exclusive). These include the following:

1. Genetic nurture (effects of genetically-influenced phenotypes in the parents or other family members on the child’s phenotype). One could divide this into a) genetic nurture that is unique to the nuclear family, versus b) genetic nurture effects shared with the extended family, which could include the effects of socioeconomic status.
2. Confounding due to parental assortment, which generates an association between non-transmitted common alleles in the parents and common and rare variants transmitted to the child.
3. Confounding due to population stratification i.e. correlations between the genetic component captured by the PGS and environmental factors due to broader scale structure in the population (e.g. correlations between the genetic component of educational attainment and regional socio-economic status). Population stratification confounding may be exacerbated when the PGS weights are derived from GWAS with imperfectly controlled population stratification.

In order to understand the nature of the associations with non-transmitted alleles, one would need to disentangle these various factors.

To partially address population stratification as a potential explanation, we have now repeated the trio analysis using the polygenic score derived from a within-family GWAS for educational attainment¹, as described on the first page of this response document. Reassuringly, this recapitulated the significant effect of non-transmitted alleles (at least in the mother), suggesting that uncontrolled population stratification in the original GWAS is not driving this finding. This result has been added to the manuscript in the third paragraph of the Discussion and in Supplementary Note 7. We took care to retain only case and control samples who cluster together and with white British samples (GBR) from the 1000 Genomes project (Supplementary Figures 10-15), and indeed we see only very subtle differences between the various case and control cohorts on the first twenty common variant-based principle components (Figure R8), which we have, in any case, corrected for. However, we cannot totally rule out that subtle population stratification remains uncorrected within our sample and contributes to the non-transmitted coefficients.

Recent work by Michel Nivard and colleagues has shown that, if one has genetic data on the grandparents or aunts/uncles of the index children, one can use this extended family design to remove the contribution of population stratification to non-transmitted coefficients¹⁰. Thus, if we had a sufficiently large sample size of NDC cases and controls with genetic data on parents as well as aunts, uncles or grandparents, we might be able to use this approach to rule out this particular confounder. However, within the current datasets (DDD and GEL) there are only a few dozen probands who fulfil this criterion, so we would be hugely underpowered. The only available dataset with large numbers (i.e. 1000s) of children, parents and aunts/uncles genotyped is the Norwegian MoBa birth cohort. However, having consulted those familiar with the cohort, we anticipate that there would be no more than ~700 probands with NDCs (defined as intellectual disability, developmental delay, seizures or autism) within the cohort who have parents plus an aunt/uncle genotyped. We anticipate that this sample size would be insufficiently well powered to address this question, particularly since MoBa does not yet have any sequence data to allow us to determine which NDC cases have a likely monogenic diagnosis, who should be removed from this analysis to reduce noise. Additionally, it is questionable the extent to which conclusions could be extended to clinically recruited cohorts such as DDD/GEL given the differential ascertainment of NDC cases to those cohorts versus MoBa, a population-based birth cohort. In MoBa, NDC cases would need to be identified through linked electronic health record data which seem to be less reliable for the identification of NDC cases than clinician-reported HPO terms (from our experience in the GEL 100,000 Genomes Project). Furthermore, children with NDCs in MoBa may well be biased towards milder phenotypes than DDD/GEL, so may have a different genetic architecture.

A second possibility we have considered to address the reviewer's question is whether we could incorporate phenotype data on the parents of NDC cases and controls to explore traits that might be mediating the effect of non-transmitted coefficients (i.e. to explore the indirect genetic effects/'genetic nurture' hypothesis). Ideally these would include data on parental cognitive ability, education level, parenting behaviours (e.g. measures of cognitively-stimulating parenting), and possible prenatal exposures (e.g. alcohol use). Many such measures are available on the parents from the birth cohorts we are using as controls.

However, unfortunately only extremely limited phenotype data are available on parents from DDD and GEL. In DDD, we have only HPO terms considered relevant to the child's condition plus a small number of questions asked by the clinicians about prenatal exposures (e.g. maternal diabetes, anti-epileptic drugs, and premature birth). In GEL, the only parental phenotype data available are the HPO terms relevant to the child's condition plus electronic health records. We did explore whether we could use the Index of Multiple Deprivation (IMD) in GEL as a measure of socioeconomic status that might at least be correlated with environments or parental phenotypes that could mediate the effects of the non-transmitted alleles. Socioeconomic status is known to be correlated with genetic propensity for educational attainment (e.g. ¹¹). We found that rerunning the trio model in GEL controlling for IMD slightly attenuated the non-transmitted coefficients by ~20% for PGS_{EA} , PGS_{CP} and $PGS_{NonCogEA}$ (see **Figure R9** below). This could be interpreted as suggesting that some aspect/s of the family-specific environment which are correlated with IMD are partly mediating the effects of non-transmitted alleles. However, this analysis has multiple limitations and caveats, including the following:

- The attenuation of the non-transmitted coefficients was not significant, and the analysis is underpowered. We estimate that at this sample size, we would need to observe parental coefficients for PGS_{EA} that are greater than 0 to see a significant change upon adding the IMD term, which is implausible.
- It does not tell us which parental behaviours or aspects of the family-specific environment are actually mediating the effect captured by IMD.
- It has been shown that this approach of controlling for parental phenotypes (or other factors) thought to be mediating indirect effects in this trio model can produce biased estimates of residual genetic nurturing effects ¹². This is particularly true when the polygenic scores used explain only a small amount of variance in the phenotype, as is the case here, or if there is parental assortment. Thus, if suitable cohorts become available in future (i.e. of NDC families and controls with genetic data plus relevant environmental measures and phenotypic data on the parents), we anticipate that new methodological development will be required to robustly examine which factors are mediating genetic nurture effects. We believe this is beyond the scope of the current paper.

Thus, in summary, although we agree that understanding the role of the non-transmitted alleles in risk is really interesting and important, to our knowledge, no methodology or datasets currently exist that would allow us to fully separate out true indirect genetic effects from confounders and, if indirect effects are indeed present, to understand what is driving them.

Another comment lies with the use of the term "rare neurodevelopmental conditions," which may imply "rare diseases" that are actually extremely rare (e.g., Prader-Willi, Noonan, Kleeftstra syndromes). Unless families in the DDD cohort were ascertained for severe and very specific rare neurodevelopmental conditions (or for the presentation of idiopathic ID/DD and congenital anomalies), it would be fine to drop "rare" from describing the cohort. Some clarification (or correcting the reviewer's view) would be helpful.

We feel that it is important to retain the term "rare neurodevelopmental conditions" for two main reasons.

Firstly, the DDD and GEL cohorts were indeed ascertained for rare conditions that were at least suspected to have a genetic (likely monogenic) cause. Some of the recruited patients

had a prior clinical diagnosis of a very rare syndrome (although not a genetic diagnosis), but many did not, and were simply suspected to have a very rare/novel monogenic syndrome. We have listed the DDD recruitment criteria below for more information. The GEL patients included in our study were recruited under a variety of phenotypic categories with different criteria including “neurology and neurodevelopmental disorders” and “ultra-rare disorders” (listed at <https://files.genomicsengland.co.uk/forms/Rare-Disease-Eligibility-Criteria.pdf>). Broadly, these recruitment criteria are likely to have resulted in a similar pool of NDC patients in DDD versus GEL, although the differing approaches to recording HPO terms make this difficult to assess robustly (as noted in Supplementary Note 1). Certainly our unpublished analyses suggest that DDD and GEL NDC patients have a similar burden of damaging *de novo* mutations, which account for the majority of monogenic diagnoses.

Secondly, we think it’s important to distinguish the phenotypes in these cohorts from other neurodevelopmental conditions that are relatively common (i.e. more common than the kinds of monogenic syndromes that contribute to DDD and GEL), such as schizophrenia and autism. (We note that many individuals in DDD and GEL do have autism, but they are unlikely to have been recruited to either cohort if they were autistic in the absence of other phenotypic abnormalities - note DDD recruitment criterion #5 listed below; the autistic individuals included in these cohorts typically have autism with ID.) More common neurodevelopmental conditions such as autism and schizophrenia are known to have a different genetic architecture from the kind of rare neurodevelopmental disorders on which we are focusing, with a much smaller contribution of monogenic causes and a larger polygenic component. Thus, we think it’s important to make this distinction by referring to the subject of our paper as “*rare neurodevelopmental conditions*”.

For further context, the DDD recruitment criteria were as follows:

- 1. Neurodevelopmental disorder – for example developmental delay and/or learning disability (of a level requiring or likely to require a statement of special educational needs), epileptic encephalopathy or severe cerebral palsy*
- 2. Congenital anomalies – multiple congenital anomalies (two or more major anomalies) or a single major anomaly together with a neurodevelopmental disorder, aberrant growth, dysmorphic features or unusual behaviour*
- 3. Abnormal growth parameters (height, weight, OFC) – two or more parameters >3SD above or below the mean or a single parameter >4SD above or below the mean (except for obesity where the threshold for isolated obesity is >4.5SD together with a strong suspicion of a genetic aetiology)*
- 4. Dysmorphic features*
- 5. Unusual behavioural phenotype in conjunction with one or more of the above features or extreme behavioural phenotype strongly suspected to have a genetic basis (including classical autism)*
- 6. Genetic disorder of significant impact for which the molecular basis is currently unknown with: i. several affected family members or ii. one other affected family member with a rare, consistent and distinctive phenotype or iii. a single case that is associated with a severe phenotype.*

References 51 and 54 seem to be the same.

We thank the reviewer for pointing this out. We have now removed the duplicate and updated to the published version of this paper.

Referee #2 (Remarks to the Author):

This paper is a thorough study of the combined effects of rare and common variants for rare neurodevelopmental conditions (NDCs). The authors perform a GWAS of NDCs in a British ancestry (GBR) sample that is greatly expanded relative to their previous work, and they find 2 genome-wide significant loci. They use these summary statistics to estimate the heritability of NDCs and the genetic correlation between NDCs and a variety of other brain related traits and conditions. They assess whether these genetic correlations may be due to common pathways with educational attainment by using genomic SEM and find little differences between the raw genetic correlations and the genetic correlations based on genomic SEM.

In the central analysis of the paper, the authors find that the mean PGS for educational attainment (EA), cognitive performance (CP), and the noncognitive component of EA (NonCogEA) is lower among those without a monogenic diagnosis. They argue that is consistent with a liability threshold model. They evaluate whether this difference between diagnosed and undiagnosed individuals may be driven by other factors (e.g., being in a trio in the data or being born prematurely) but conducting a mediation analysis, and they find that each of the factors they consider each only explain a small fraction of the association between the PGS and diagnostic status. Relatedly, the authors also find that probands without a monogenic diagnosis have a higher PGS than the mean PGS of their parents. In a regression that includes the proband, mother, and father PGS, the child EA PGS is not statistically associated with NDCs but the parents' PGSs are. In contrast, the child NDC PGS is statistically associated with NDCs but not the parents' PGSs. They interpret this as suggesting that there are both direct and indirect effects of genes on NDCs.

Finally, they perform a series of follow-up analyses to understand what might be driving the results of the paper, including an MR analysis of the effect of premature birth on EA and vice versa (acknowledging that a positive result for the latter could not actually be a causal relationship) and tests for assortative mating.

Much about this paper is very interesting and it clearly advances the literature on the various genetic factors that influence genetic associations for NDCs. I have a number of concerns about the paper, however, that I describe below. Many of these concerns center around me having a hard time understanding what was done precisely or not understanding the motivation for the things that were done.

We thank the reviewer for their positive comments, and hope that our responses below help address their concerns.

Major comments:

1) My biggest concern is that I struggle to understand the big picture contribution of the work. As far as I can tell, the big point is that common and rare variants both independently have an effect on risk of NDCs. They conduct many other analyses to verify the robustness and to understand the mechanisms driving this result, but most of those analyses are subject to substantial limitations (which the authors acknowledge), like assortative mating, residual population stratification, and core assumptions being implausible in the case of their MR studies. They bring up at several points that their results are consistent with a liability threshold

model, which is true, but I'm not sure what models they are trying to rule out by their data. Are there other proposed models with different implications that are inconsistent with their data?

We apologise that the big picture contribution of this work did not come through more clearly to the reviewer. It is not quite correct that “the big point is that common and rare variants both independently have an effect on risk of NDCs”. It is true that our demonstration of a negative correlation between monogenic diagnoses and polygenic risk within NDC cases supports a liability threshold model, and that this was one of the important findings in the paper. However, several other findings were equally important in our opinion (listed below). Additionally, our results do not show that common and rare variants have independent effects on NDCs; in fact, our results in Figure 6 show that common variants conferring risk for NDCs are actually positively correlated with *inherited* rare variants conferring risk (which are typically not considered diagnostic, as explained in response to comment #2 below).

To try to emphasise the main points of our work better, we have clearly articulated our three main aims in the first paragraph of the Introduction: “*Firstly, we aim to better understand the nature of common variant risk for rare neurodevelopmental conditions, particularly its overlap with common variant risk for mental health and cognitive phenotypes. Secondly, we aim to elucidate the interplay between common and rare variants in the context of these conditions. Thirdly, we aim to test whether there is an effect of common variants in the parents on their child’s risk of these conditions, above and beyond the child’s own genetics.*” We have added a schematic overview figure (Figure 1) structured around these aims, which states the key finding under each aim. We have then flagged in the Results which sections are addressing each aim. To emphasise, the most important conclusions under each aim are as follows:

- **Aim 1.** The common variants associated with rare NDCs are also associated with various other mental health conditions and this is partly, but not entirely, mediated by a latent trait that is genetically correlated with cognitive ability/educational attainment.
- **Aim 2.** Patients with a monogenic diagnosis involving a highly penetrant rare variant have significantly less polygenic risk than those without. This is not driven by confounders, and supports a liability threshold model. (For added nuance, see response to the next reviewer comment).
- **Aim 3.** Parents’ polygenic scores for educational attainment and cognitive performance are associated with their children’s risk of NDCs even after controlling for their children’s polygenic scores, which show no evidence for direct genetic effects, in contrast to the polygenic score derived specifically for NDCs. We propose two potential mechanisms underlying this observation, which are not mutually exclusive:
 - a) Polygenic scores for educational attainment and cognitive performance may affect the children’s risk through the prenatal or family environment. Supporting this, we show some evidence that alleles associated with educational attainment may exert their effects on NDC risk via premature delivery.
 - b) The observation is confounded by parental assortment in the present and previous generations, which has caused the (inherited) rare and common variant components contributing to NDC risk to become positively correlated. This implies that the contributions of rare and common variation

to cognition-related traits are inherently compounded and suggests that the typical approach of investigating either class of variation in isolation may overestimate its contribution.

Points #2, #3, #3a and #3b are made in the Abstract, and all points are highlighted in the Discussion (point #1 in paragraph 1, point #2 in paragraph 2, point #3 in paragraph 3, point #3a in paragraph 5, point #3b in paragraph 4).

There are several models we can rule out given our data, including:

- That liability for NDCs is conferred only by fully penetrant monogenic causes and environmental factors. If this were the case, and in the absence of confounders, we would not expect to see a difference in common variant risk between patients with a monogenic diagnosis and those without (**Figure 3**). We have emphasised this by adding a sentence to the second paragraph of the Discussion: “*Our finding suggests we can rule out a model whereby liability for NDCs is conferred only by fully penetrant monogenic causes and environmental factors.*” In theory, being in a trio could drive this difference in polygenic risk between diagnosed and undiagnosed patients, since it is easier to find genetic diagnoses for patients in trios, who tend to have lower NDC-associated polygenic risk (specifically, lower PGS_{EA}); however, we show that this does not fully explain the difference in polygenic risk between diagnosed and undiagnosed patients (PGS_{EA} is still significantly associated with diagnostic status after controlling for trio status; new **Extended Data Figure 6B**).
- That the association between NDCs and common variants is entirely due to *direct* genetic effects on risk (i.e. these common variants function purely by directly affecting the phenotype of the individual who carries them). If this were the case, we would not have observed significant coefficients on any parental PGSs in the trio model (**Figure 5**). We have inserted the following sentence into the third paragraph of the Discussion to emphasise this: “*Thus, a key conclusion from this work is that the association between NDCs and common variants is not entirely due to their having direct genetic effects on risk.*”
- That common variants exert no direct genetic effects on NDC risk. If this were the case, we would not have observed any significant transmission disequilibrium in the pTDT (**Figure 4A**), and similarly, no PGS would have remained significantly associated with NDC risk in the child after controlling for parents' PGSs (**Figure 5**). In fact, we saw evidence that common variants ascertained for their association with NDC risk (i.e. $PGC_{NDC,DDD}$) did show significant direct genetic effects.

2) Relatedly, their primary designs to understanding the role of rare and common variants is to condition on the presence of rare variants and evaluate the mean of the PGS (and the mean relative to the parental mean in the case of their trio analyses). The reason for this is that collider bias would produce a negative correlation between common variants (and therefore PGSs) that are associated with NDCs and with NDCs. The magnitudes of this approach are difficult to interpret, and the authors don't really attempt to interpret them. Is there a reason that the authors didn't just regress NDC on the PGS, an indicator for whether a person has a monogenic diagnosis, and the interaction between the two (as they do in the section that employs trios)? It seems like this approach is no worse than their collider approach, but it produces estimates that have a more straightforward interpretation. If they prefer to keep their

specification, the authors should probably state what they would expect the estimates to be or provide some other way to interpret the magnitudes.

Before responding to the reviewer's comment, we wish to further clarify that our initial manuscript included two types of analyses involving rare variants, and we have added some minor changes to the third paragraph of the Introduction to emphasise the difference here. Firstly, a key analysis focused on comparing patients with a monogenic diagnosis versus those without (**Figure 3**) - we believe this is the main analysis the reviewer is referring to. These monogenic diagnoses are due to rare variants that mostly occur *de novo* in families in which the parents are unaffected, although some are inherited (biallelic variants in autosomal recessive genes, variants in X-linked recessive variants, and variants in dominant genes inherited from affected parents)¹³. Secondly, in several other analyses, we considered rare variants that are inherited, in many cases from unaffected parents. These have been shown to impact risk of NDCs but are rarely considered diagnostic if inherited from unaffected parents since they are incompletely penetrant. We tested three different hypotheses regarding these inherited rare variants, namely, whether their penetrance was impacted by PGSs within families (**Supplementary Figure 8**), whether they were correlated with PGSs due to assortative mating (**Figure 6, Supplementary Figures 6/7**), and whether this correlation mediated the significant effects of non-transmitted common alleles (**Extended Data Figure 10B**). These hypotheses were fundamentally different from the hypothesis we were testing in **Figure 3**, and involved different predictions. In **Figure 3**, we expected to see a *negative* correlation between polygenic risk for NDCs and the presence of a diagnostic rare variant due to the collider bias effect the reviewer mentions; our results confirmed this. In **Supplementary Figure 8**, we expected to see that the child with the inherited rare variant had *more* polygenic risk for NDCs than the unaffected parent who had passed on this rare variant, but we did not observe this very convincingly. In **Figure 6**, we expected to see a *positive* correlation between polygenic risk for NDCs and rare inherited variants due to parental assortment, and indeed we did observe this. (Note below our point about different PGSs having different directions of effect and hence the use of the umbrella term "more polygenic risk for NDCs".)

In hindsight, we can see why one might expect to obtain a more interpretable result by fitting the model suggested by the reviewer, namely:

$$\text{NDC case/control status} \sim \text{PGS} + (\text{has diagnosis}) + \text{PGS}^* (\text{has diagnosis})$$

The reason we did not do this is largely that the controls used (UKHLS, ALSPAC and MCS) are not strictly 'controls' in the sense that they have not been screened for having NDCs or highly penetrant NDC-causing variants - rather, they are population samples. It is possible that some small fraction of them (probably <1%) do have monogenic NDCs. However, identifying these cases robustly would require sequence data for all individuals and their parents (to be able to identify *de novo* mutations, which are much more likely to be highly penetrant for NDCs than inherited variants), as well as detailed phenotype data, which are not available. Nonetheless, since the vast majority of diagnostic variants identified in the DDD and GEL NDC patients are considered fully penetrant (since most parents are unaffected and most diagnostic variants are *de novo* or recessive), it is probably fair to assume that none of the 'controls' have such variants. We thus followed the reviewer's suggestion and fitted the logistic regression model above using PGS_{EA}, assuming none of the 'controls' have a diagnosis (using the same controls from UKHLS and GEL as were used for the GWASs). As expected, higher PGS_{EA} was associated with lower risk of NDCs (odds ratio=0.78, p=5x10⁻⁶⁶). However, diagnostic status was not significantly associated with NDC risk (p=0.85) and the coefficient and standard

error were unrealistic ($\beta=19.9$, $se=106.3$). We presume this is due to the fact that all individuals with a monogenic diagnosis are cases; such quasi-complete separation leads to unstable maximum likelihood estimates in logistic regression.

To address this issue, we applied Firth correction which introduces a correction factor into the likelihood function. In this analysis, diagnostic status was significantly associated with higher NDC risk ($\beta = 9.70$, 95% CI 8.08–1001.7), as was PGS_{EA} ($\beta = -0.247$, 95% CI -0.275 – -0.219). We did not observe a significant interaction between PGS_{EA} and diagnostic status ($p=0.86$), but it is difficult to interpret this given that all of the controls have diagnostic status=0. We also fitted a model without controlling for diagnostic status, applying the Firth correction:

$$\text{NDC case/control status} \sim PGS_{EA}$$

In this simpler model, the effect of PGS_{EA} on NDC risk was estimated at $\beta = -0.199$ ($se = 0.012$), which was significantly smaller in magnitude (z score test $p\text{-value} = 0.0092$) than the effect size estimated when controlling for diagnostic status and its interaction with PGS ($\beta = -0.247$, $se = 0.014$). This is consistent with our observation that the difference in PGS_{EA} between undiagnosed patients and controls was larger than that between diagnosed patients and controls (**Figure 3A**).

Given all this, we do not think that reporting the results from the logistic regression that the reviewer suggested (even with the Firth correction) is going to be more interpretable than simply emphasising the average difference in PGS between undiagnosed patients and controls *versus* between diagnosed patients and controls. We have substantially altered the first paragraph of the Results section on “Probands with monogenic diagnoses have less polygenic risk for NDCs than undiagnosed patients” to focus on differences compared to the weighted MCS controls, since these should be representative of the background population (**Supplementary Note 3**). We hope that the following addition provides sufficiently interpretable estimates for the reviewer: “*After correcting all polygenic scores for principal components and standardizing them such that the weighted MCS controls have mean 0 and variance 1, we observe that the undiagnosed probands have an average PGS_{EA} that is 0.25 standard deviations (95% CI: 0.22-0.27) lower than these controls, whereas the diagnosed probands have an average PGS_{EA} that is 0.12 standard deviations (95% CI: 0.09-0.16) lower than controls ($p=3.0 \times 10^{-9}$ for difference between diagnosed and undiagnosed).*”

3) The authors' discussion of the genomicSEM analyses confused me a little, and part of the problem is that I'm not totally certain what the authors have done. After reading the paper and reading up on genomicSEM, I believe that the authors consider a model where there is some latent factor that is the residual of a project of the genetic factor for NDC onto the genetic factor for EA/CP. They estimate the genetic correlation between this residual and a variety of other traits. Because this genetic correlation is not statistically different than the raw genetic correlation between NDC and the corresponding trait, they argue that these correlations are not explained by their relationship with EA. But this seems like the wrong design for their question. Imagine the 99% of the variation in the genetic factor for NDC is explained by the genetic factor for EA and that this portion explained by EA was moderately correlated with ADHD, but the residual was also moderately correlated with ADHD. Then we would see that there would be little change in the rg in the genomicSEM estimate, but it would still be the case that much of the covariance between NDC and ADHD is actually operating through the EA pathway. This extreme case is obviously not actually true, but it hopefully highlights why I think the genomicSEM rg is not getting at the question you are asking. The more appropriate

parameter would probably be to use the output of genomicSEM to obtain a parameter more similar to the mediation analysis parameter you estimate in your section on mediation.

We thank the reviewer for this astute comment. In hindsight, we see that it is not straightforward to interpret the comparison between the raw genetic correlation and the genetic correlation of the residual after subtracting the EA signal using GenomicSEM. We have now removed the comparison of the genetic correlations using the z-test from the manuscript, and instead added results from a new analysis of the decomposition of the genetic correlations using GenomicSEM, which we think is what the reviewer was suggesting (**new Supplementary Figure 1**). We performed this analysis on brain-related traits that showed a significant genetic correlation with NDCs. Specifically, we calculated the percentage of the genetic correlation between NDCs and the target trait that was explained by the latent EA variable when the contribution from the EA and non-EA components showed the same direction, following Demange *et al.*¹⁴. This is now explained in the “Genetic correlations” section of the main Methods and the “Decomposition of genetic correlation using GenomicSEM” section in **Supplementary Methods**. We also performed the same analysis using cognitive performance and the latent cognitive variable rather than EA. Among all the traits that were assessed in the mediation analysis, the genetic correlation with ADHD showed the highest contribution from the latent EA variable (77%), which is consistent with our finding that the non-EA component of NDCs no longer showed a significant genetic correlation with ADHD. We revised the fourth paragraph of the Results and first paragraph of the Discussion to reflect these new results.

Similarly, we also updated the GenomicSEM analysis assessing the genetic correlation with prenatal risk factors (**Extended Data Figure 8BC**) in **Supplementary Note 5**. The latent variable representing the EA signal explained 90% of the genetic correlation between NDCs and smoking initiation, and only 35% of that with premature delivery, suggesting that the genetic correlation with smoking initiation is primarily driven by EA while there is considerable residual genetic covariance between premature delivery and NDCs that is independent of EA.

4) The authors perform analyses to test whether the differences in PGS means might be due to technical, clinical, or prenatal factors. To do this, they say they perform a test of whether the association between the PGS and a person's diagnostic status mediated by these factors. I wonder if they were doing the analysis backwards though (e.g., whether the association between diagnostic status and trio status is mediated by the PGS). The reason for my confusion is that in the main text, in Figure 2B, and in Supplementary Table S8, they only report how the coefficient on the factor changes by including the PGS rather than how the coefficient on the PGS changes when they add the factor. (In the Table, they also appear to report a effect of the mediator on the PGS rather than the effect of the PGS on the mediator.) The way they describe what they did in Supplementary Note 4 seems more accurate, but it is still not totally clear. Can the authors confirm whether they are actually looking at the mediating effect of their factors on the relationship between the PGS and diagnostic status? If so, they should also probably include all of the relevant estimates to obtain their mediation estimates. It would also be of interest what fraction of the diagnostic status-PGS relationship is mediated by the full set of factors they consider rather than just one at a time.

The reviewer raises a valid point. In the original manuscript, we treated PGS as the exposure, and trio status and prematurity as the mediators, and estimated the proportion of the

association between diagnostic status and PGS that was mediated by each of these factors. The mediation analysis does not tell us whether a causal model in which the PGS is a mediator fits better than a model in which one of these factors is mediator; it relies on assumption of the causal pathways that we provide. We do not need mediation analysis to reach the conclusion that the difference in PGS between diagnosed and undiagnosed individuals is largely independent of these factors tested in the paper. Given these reasons, we have now removed the mediation analysis from the manuscript. Instead, we now simply show the association between diagnostic status and probands' PGS_{EA} before and after correcting for one or more of the factors shown in **Figure 3B** (new **Extended Data Figure 6B** - similar to **Figure R10** below; **Supplementary Table 8**), as suggested by the reviewer. The association between PGS_{EA} and diagnostic status remained significant after controlling for any or all of these factors, and the change in effect size was negligible.

After controlling for all the factors in **Figure 3B**, the probands' PGS_{EA} was still significantly associated with diagnostic status, with only a small decrease in odds ratio (1.102 (95% CI: 1.051–1.155) versus 1.116 (95% CI: 1.067–1.168) without correction) (**Figure R10**). However, we do not believe that this model including all those factors really makes sense, for the following three reasons. (1) Proband sex and maternal diabetes are not likely to confound the association between PGS_{EA} and diagnostic status because they are not significantly associated with PGS_{EA} (**Figure 3C**). (2) F_{ROH} is not significantly associated with diagnostic status within this subgroup of probands with white British ancestry (**Figure 3B**), although it was within the full DDD study¹³. (3) Severity of ID/DD and whether the proband has any affected first degree relatives are most likely to be consequences of having or not having a monogenic cause, rather than a cause of getting a diagnosis. On the other hand, trio status affects ability to discover the diagnostic variant, and prematurity is a causal risk factor of NDCs (and thus probably influences whether or not the proband actually has a diagnostic variant), which is why we only considered these two factors in our original mediation analyses. Thus, in **Extended Data Figure 6B** we show the results of a joint model conditioning on only trio status and prematurity, but not all the other factors. This showed a similar odds ratio for the association of PGS_{EA} with diagnostic status (1.100; 95% CI: 1.051–1.152) to that observed without conditioning.

We have removed the sentence about mediation analysis in the main text and replaced it with the following sentence: “*Similarly, after controlling for these factors, the association between PGS_{EA} and diagnostic status remained significant and the change in effect size was negligible (Extended Data Figure 6B).*” We have also updated the section on “Associations between PGS and diagnostic status” in the Methods to reflect these new analyses.

We note that, when restricting to trios (thus reducing the sample size considerably), diagnostic status was no longer significantly associated with proband's PGS_{EA}, mother's PGS_{EA}, or father's PGS_{EA}, so we did not include the analysis of parents' PGS.

Minor comments:

5) The authors could be more clear about what comparisons they are making in many places and why. I recognize that this is hard because there are several dimensions of comparisons (e.g., cases vs controls, monogenic diagnosis vs no diagnosis, trio vs non trio, etc.) but I was often confused about what was being done. For example, the authors state, "Despite this, we

observed that for all polygenic scores except for PGSNonCogEA, the diagnosed probands still had significantly more polygenic risk than the controls" and they reference Figure 2A. In Figure 2A, I believe the diagnosed probands are the solid blue squares, and every solid blue square is statistically distinguishable from zero (which presumably is the mean of the controls), but some have lower risk (EA, CP, and NonCogEA) and some have higher risk (SCZ and NDC). My guess is that the authors use this language since a low PGS for EA or CP is associated with a higher risk of NDCs and that NonCogEA doesn't qualify because it doesn't pass some Bonferroni-corrected threshold, but the text doesn't clarify this for me and I'm left guessing. I found myself trying to sort this sort of thing out at several points while reading this manuscript.

We apologise for this lack of clarity. Throughout the manuscript, we were trying to strike the right balance between summarising the key findings without weighing down the text by mentioning every significant or non-significant result (which would greatly inflate the word count and make the manuscript fairly unreadable), but see that this may have sometimes created confusion. The reviewer is right that we are using the term 'more polygenic risk' to mean a lower polygenic score for the traits that are negatively correlated with NDCs (EA, CP, and NonCogEA) or a higher polygenic score for NDCs or schizophrenia (positively correlated with NDCs). We have added the following sentence to the end of the first section of the Results to clarify this: "*Since polygenic scores for traits showing genetic correlations with NDCs in the same direction usually showed similar results, in the text below, we often use the term "more polygenic risk" (for NDCs) as a shorthand for having higher $PGS_{NDC,DDD}$ and/or PGS_{SCZ} , and/or lower PGS_{EA} , PGS_{CP} and/or $PGS_{NonCogEA}$.*"

In the case of the reviewer's particular comment regarding **Figure 3**, the reviewer is correct in their interpretation, and we see why this may have been unclear without digging into **Supplementary Table 5**. We have now added asterisks to **Figure 3A** to indicate the significance of the comparison between each subgroup of patients and control individuals. As the reviewer guessed correctly, the difference in $PGS_{NonCogEA}$ between diagnosed patients and controls did not pass multiple testing correction. We also annotated p-values for significant differences between diagnosed and undiagnosed patients. We have added the following text to the legend of **Figure 3A** to clarify: "*Subgroups that have significantly different average PGS from controls (dashed line) are indicated by an asterisk ($p < 0.05$) or double asterisk ($p < 0.01$ after Bonferroni correction for five PGSs). Significant differences between diagnosed (dark blue) and undiagnosed (red) patients are annotated with P-values.*"

We have also made changes in the following places to try to improve clarity about which groups were being compared and/or why conclusions were being drawn, with new additions indicated by blue text and old deleted text indicated with ~~strikethroughs~~:

- Title of the Results section "Probands with monogenic diagnoses have less polygenic risk for NDCs than undiagnosed patients", and in the first and second paragraphs of this section. We added brackets and asterisks to **Figure 3** to highlight the comparisons we reported in the text. We also updated the legend to **Figure 3** and x axes of **Extended Data Figure 6** to indicate the directions of effect sizes (e.g. lower PGS_{EA} indicates more polygenic risk and $OR > 1$ indicates more likely to get a monogenic diagnosis).
- First and second paragraph of the Results section on "Limited evidence for over-transmission of polygenic risk from unaffected parents to probands"
- Legend to **Figure 4** and asterisks in **Figure 4B**

- Legend to **Figure 5**, and first and second paragraphs of the section “Non-transmitted common alleles in unaffected parents are associated with their children’s risk”
- The paragraph on “Exploring the role of prenatal factors” in the Results. We have also added brackets and asterisks to **Supplementary Figure 5** to make clear which coefficients are being compared to back up the final sentence in this section.
- In the Results section “Parental assortment obscures the true nature of common variant effects”, we have indicated more clearly which specific figure panels back up each statement in the second paragraph, and made a small addition to the third paragraph.
- Second paragraph of the Discussion: “we find probands with more affected first-degree relatives had both a lower PGS_{EA} and a lower chance of getting a monogenic diagnosis in DDD *than probands with no affected relatives.*” We also made some minor changes to the final sentence of this paragraph and to **Extended Data Figure 4** to clarify which comparisons are being made here.

6) It would be helpful if the authors were clear throughout whether their estimates of h^2 were on the observed scale or the liability scale.

All the SNP heritability estimates are on the liability scale assuming a population prevalence of 1%, and we have added this clarification in the paper when heritability estimates are mentioned.

7) In the section on nontransmitted alleles, the authors state, “We next explored one way in which familial polygenic background might affect children’s risk of neurodevelopmental conditions, namely indirect genetic effects, i.e. effects of alleles in parents on parental phenotypes that affect their offspring’s risk through the family environment.” Why is this section framed as being about indirect effects when your data don’t allow for the estimation of indirect effects? It seems like all you can actually say is that there are direct effects in some cases and you have a non-zero non-transmitted coefficient in others (which might be indirect effects but it could also be uncontrolled stratification, assortative mating, etc.).

The reviewer raises a very fair point. We have replaced that sentence with the following: “*Given these findings, and to address our third aim, we next tested whether non-transmitted alleles in the parents were correlated with their children’s risk of neurodevelopmental conditions. This could potentially be indicative of indirect genetic effects i.e. effects of alleles in parents on parental phenotypes that affect their offspring’s risk through the family environment (otherwise known as “genetic nurture”), as opposed to the direct genetic effects of alleles transmitted to the child.*” We have also made it clear in this paragraph that we assessed “*the association with non-transmitted parental alleles*”, rather than “*indirect genetic effects*” using the trio model.

Referee #3 (Remarks to the Author):

This is a remarkable project illuminating the polygenic contribution of common genetic variants to risk of rare neurodevelopmental disorders. The datasets used are some of the largest and best characterised that are available for these analyses. The authors have carried very careful

and well thought analyses that convincingly demonstrate that common variants explain ~10% of the variance in risk.

The fact that diagnosed patients have lower polygenic risk compared to undiagnosed ones is intriguing, as is the evidence suggesting that non-transmitted alleles for edu attainment and cognitive ability from parents correlate with children's risk of developing neurodevelopmental conditions. Lower polygenic risk in the results are intriguing and provide further support for liability threshold models that include rare and common genetic effects, alongside potential environmental insults - though the postulated example of zika infection is impossible to test from DNA sequencing, there might be ways to verify this once epigenetic data is present (if these index exposures).

I liked the FAQ document and thought it was an excellent addition to aid families and clinicians.

We thank the reviewer for these very positive comments.

Minor points:

LDSC is a lower estimate of heritability, needed here for genetic correlations and as the engine for the SEM analyses. However, the h2SNP reported from it is always going to be the lowest of estimates as per what was found in the development of the LDMS approach. I would focus on the GREML-LDMS analyses and I would make secondary the result of heritability for LDSC and PCGC on summary stats. This is especially because you have access to all of the raw sequencing or GWAS data from both cohorts (unlike GWAS consortia). The power of PCGC surprised me in that it approximated GREML-LDMS but nevertheless I recommend focussing on the latter for variance explained.

We thank the reviewer for the comment. We agree that SNP heritability estimates obtained from LDSC are likely downward-biased given the sample size, and methods using individual-level data give more accurate estimates. For PCGC, we also ran it with individual-level data, which is probably why it showed similar estimates to GCTA-LDMS. Following the reviewer's suggestion, we have now only mentioned the GCTA-LDMS estimate in the Results. We have rearranged the first paragraph of the Methods section on "Heritability" to reflect this new emphasis, and also explained our reasoning for it: *"We focused on the GREML-LDMS estimate in Results, since the estimates were similar to PCGC, and LDSC estimates are known to be under-estimated, especially at low sample size. All estimates are reported in Supplementary Table 3."*

The results stated in "Probands with monogenic diagnoses have less polygenic risk" should be stated as undiagnosed versus diagnosed to be consistent with the message and title of this section, and not diagnosed versus undiagnosed.

We thank the reviewer for trying to improve the consistency of the manuscript. The section title previously read "Probands with monogenic diagnoses have less polygenic risk", but we have changed this to be more explicit, namely "Probands with monogenic diagnoses have less polygenic risk for NDCs than undiagnosed patients". We believe we have consistently presented the results as diagnosed versus undiagnosed in this section, but please do correct us if we have missed something.

Does the reviewer's comment stem from confusion about the fact that different PGSs have different directions of effect (i.e. NDC risk is associated with *higher* PGS_{NDC,DDD} and PGS_{SCZ} but *lower* PGS_{EA}, PGS_{CP} and PGS_{NonCogEA})? This is similar to reviewer 2's comment #5. To try to clarify this, we have added this sentence to the bottom of the first section of the Results: "*Since polygenic scores for traits showing genetic correlations with NDCs in the same direction usually showed similar results, in the text below, we often use the term "more polygenic risk" (for NDCs) as a shorthand for having higher PGS_{NDC,DDD} and/or PGS_{SCZ}, and/or lower PGS_{EA}, PGS_{CP} and/or PGS_{NonCogEA}.*"

Alternatively, is the reviewer referring to the fact that, in **Supplementary Table 5**, we previously presented the undiagnosed patients as group 1 and diagnosed patients as group 2 (the reference group), which was inconsistent with the way this was presented in the main text? We have now changed **Supplementary Table 5** to correct this. We have also altered the legend to **Extended Data Figure 1** to ensure that the interpretation of the figure is expressed in the same way as the results are presented in the main text, namely: "*The second patient, who has a monogenic diagnosis, has fewer green circles (i.e. fewer NDC risk-increasing common variants) than the undiagnosed patient on the right, since the orange circle (i.e. diagnostic large-effect variant) is sufficient on its own to push the diagnosed patient over the diagnostic threshold.*"

I would be cautious in calling "non-cognitive" the residual from GWAS by subtraction of EA from g as the other components of IQ (the remaining half of the variants) reflect both non-cognitive factors and the other non-g components of cognitive ability.

This is a valid point, but we would like to retain the term "non-cognitive" throughout firstly for the sake of succinctness, and secondly to be consistent with the literature. However, we have now referred to this as the '*so-called "non-cognitive component"*' when this is first mentioned in the Results, and also added the following caveat in the first paragraph of the Discussion: "*Furthermore, although we observe a significant negative genetic correlation with what has been termed the "non-cognitive" component of educational attainment, we note that this could also contain elements of cognitive ability not captured in the GWAS for cognitive performance that was used in the paper that derived it.*" We wish to avoid mentioning "g" (general cognitive ability) since we suspect many readers will not be familiar with this concept.

Point for consideration:

Construction of a multi-polygenic risk score, maximising the combined prediction across all of the selected traits would appear to be a logical extension of the findings and might clarify for the readers what % variance explained can be added by common variants, at present, for individual patients.

We agree with the reviewer that combining PGSs tested in the paper would improve the prediction power. We constructed PGSs combining individual scores except for PGS_{NonCogEA}, since it was derived from a GWAS-by-subtraction analysis using GWAS summary statistics of educational attainment and cognitive performance, and thus does not provide additional information. We constructed two PGSs, with and without incorporating PGS_{NDC,DDD}:

$$\begin{aligned} \text{PGS}_{\text{EA+CP+SCZ}} &= w_1 \times \text{PGS}_{\text{EA}} + w_2 \times \text{PGS}_{\text{CP}} + w_3 \times \text{PGS}_{\text{SCZ}} \\ \text{PGS}_{\text{EA+CP+SCZ+NDC}} &= w_1 \times \text{PGS}_{\text{EA}} + w_2 \times \text{PGS}_{\text{CP}} + w_3 \times \text{PGS}_{\text{SCZ}} + w_4 \times \text{PGS}_{\text{NDC,DDD}} \end{aligned}$$

where w_i indicates the weight assigned to the PGS.

We included the composite PGS without $\text{PGS}_{\text{NDC,DDD}}$ because the latter showed different results in some analyses from other PGS, and combining all four scores could make it challenging to interpret the results. (Notably, $\text{PGS}_{\text{NDC,DDD}}$ was the only PGS that showed evidence of significant over-transmission and direct genetic effects in our original submission.) Therefore, we used $\text{PGS}_{\text{EA+CP+SCZ+NDC}}$ to try to maximise the power to distinguish patients and unaffected controls but also tested $\text{PGS}_{\text{EA+CP+SCZ}}$ in other major analyses.

To avoid overfitting when training the weights to combine individual PGSs, we trained the weights in one cohort (DDD or GEL) using a logistic regression, and calculated the composite $\text{PGS}_{\text{EA+CP+SCZ}}$ using these weights in the other cohort, and *vice versa*. PGSs were scaled so that control individuals for the testing cohort had mean of 0 and variance of 1 (i.e. GEL controls when testing GEL NDC cases, and UKHLS when testing DDD cases). We then performed the inverse-variance based meta-analysis of the two cohorts. When calculating the variance explained by $\text{PGS}_{\text{EA+CP+SCZ+NDC}}$ in GEL, we used five-fold cross-validation, since $\text{PGS}_{\text{NDC,DDD}}$ was derived from the GWAS in DDD patients and we could not use them to train the weights. More specifically, we randomly split the GEL case-control samples into five equal-sized sets, keeping the same sample prevalence for each. We trained the weights in four sets and estimated the variance explained in the remaining set, then repeated this procedure four times to estimate the average variance explained across all five sets.

As expected, the composite PGS explained higher variance than each individual score in both DDD and GEL (**Figure R11**). $\text{PGS}_{\text{EA+CP+SCZ}}$ explained 0.769% (logistic regression $p=9.81 \times 10^{-47}$) of variance in risk in DDD versus UKHLS controls, and 0.890% ($p=2.98 \times 10^{-41}$) of variance in risk in GEL. Incorporating $\text{PGS}_{\text{NDC,DDD}}$ further increased the variance explained to 0.994% in GEL. In comparison, the maximum variance explained by any of the individual PGSs was 0.55% by PGS_{EA} in DDD and 0.83% by PGS_{EA} in GEL (**Supplementary Table 2**).

We next repeated the major analyses to assess whether the composite PGSs showed better performance than using individual PGSs. We focused on $\text{PGS}_{\text{EA+CP+SCZ}}$ in these analyses, since $\text{PGS}_{\text{EA+CP+SCZ+NDC}}$ could not be tested in DDD due to over-fitting. We flipped the sign of $\text{PGS}_{\text{EA+CP+SCZ}}$ such that a lower score indicates higher risk of NDCs, for easier comparison with PGS_{EA} , which was the most predictive individual PGS and had the highest weight in the composite PGS. Similar to PGS_{EA} , patients who had a monogenic diagnosis had significantly higher $\text{PGS}_{\text{EA+CP+SCZ}}$ (lower risk of NDCs) than undiagnosed patients ($\Delta=0.11$ SD, two-sided t -test $p=5.6 \times 10^{-7}$; **Figure R12**), a very similar result to that seen for PGS_{EA} . In the polygenic transmission disequilibrium test (pTDT), the combined score did not show significant transmission disequilibrium ($\Delta=0.012$ SD of mean parental PGS, $p=0.44$; **Figure R13**), similar to the constituent PGSs. In the trio model controlling for both the children's and parents' PGSs, $\text{PGS}_{\text{EA+CP+SCZ}}$ showed significant non-transmitted coefficients ($p<5.5 \times 10^{-9}$; **Figure R14**) but no significant direct genetic effects ($p=0.15$), also consistent with the results seen for the constituent PGSs. In the analyses of correlation between PGSs and rare variants (**Figure R15**), $\text{PGS}_{\text{EA+CP+SCZ}}$ showed similar results to PGS_{EA} .

In summary, the composite PGSs showed improved power to distinguish between patients and unaffected control individuals and in major analyses, the composite $\text{PGS}_{\text{EA+CP+SCZ}}$ showed consistent results with PGS_{EA} . However, given the different observations for $\text{PGS}_{\text{NDC,DDD}}$ versus the other PGSs, we are inclined to retain the results of the individual PGSs in the

manuscript, and not to include the results from the composite PGSs (other than the variance explained in Supplementary Table 2) since it simply adds to the multiple-testing burden while actually being very correlated with PGS_{EA} and invariably showing similar results to it. We have added the following text to the second-last paragraph in the Discussion: *“We explored combining the different PGSs into a composite PGS to try to improve power; although this explained slightly more variance on the liability scale than PGSEA (Supplementary Table 2), results from the main analyses were very concordant between this composite PGS and PGSEA (which had the highest weight), so we decided not to include them in the manuscript.”*

Figures and tables pertaining to the responses

Table R1

Table R1. SNP heritability estimates using different methods before and after restricting to patients with ID/DD or seizures. All SNP heritability estimates are on the liability scale assuming a population prevalence of 1%. LD score regression (LDSC) was run on summary statistics from the GEL-derived GWAS, DDD-derived GWAS, and the meta-analysed GWAS. LD- and MAF-stratified GREML (LDMS) and phenotype-correlation genotype-correlation (PCGC) regression were run in DDD and GEL GWAS samples separately, then the SNP heritability estimates were meta-analysed. CI: confidence interval; h2: heritability estimate; LRT: likelihood ratio test

GWAS	N_cases	N_controls	LDSC h2 (95% CI)	LDMS h2 (95% CI)	LDMS LRT p-value	PCGC h2 (95% CI)	PCGC LRT p-value
GEL cases versus GEL controls	3618	13667	1.9% (-2.2%–5.9%)	8.7% (3.2%–14.1%)	0.28	9.0% (3.3%–14.8%)	8.1E-06
GEL cases with ID/DD versus GEL controls	3509	13667	2.4% (-2.0%–6.6%)	9.3% (3.6–14.9%)	0.27	10.3% (4.3%–16.3%)	6.8E-07
GEL cases with seizures versus GEL controls	899	13667	4.7% (-8.5%–17.8%)	18.4% (-0.1%–37.0%)	0.15	9.8% (-9.6%–29.3%)	0.15
DDD versus UKHLS	6397	9270	5.9% (2.4%–9.5%)	12.0% (8.9%–15.0%)	9.6E-03	11.5% (7.5%–15.4%)	3.3E-19
DDD cases with ID/DD versus UKHLS	5679	9270	7.1% (3.5%–10.7%)	11.5% (8.2%–14.7%)	0.027	11.0% (6.9%–15.2%)	1.3E-15
DDD cases with seizures versus UKHLS	1319	9270	2.6% (-8.2%–13.4%)	14.2% (4.1%–24.4%)	0.5	16.7% (4.8%–28.6%)	6.6E-05
meta-analysis of DDD and GEL	10015	22937	3.7% (1.7%–5.7%)	11.2% (8.5%–13.9%)	NA	10.7% (7.4%–14.0%)	NA
meta-analysis (ID/DD only)	9188	22937	4.4% (2.2%–6.6%)	10.9% (8.1%–13.8%)	NA	10.8% (7.4%–14.2%)	NA
meta-analysis (seizures only)	2218	22937	0.9% (-1.1%–2.8%)	15.2% (6.3%–24.1%)	NA	14.8% (4.7%–25.0%)	NA

Figure R1

Figure R1 Enrichment of NDC risk genes (from GWAS of educational attainment or the DDG2P gene panel) in brain cell types. (Now **Supplementary Figure 18** in the manuscript). We focused on 19,130 autosomal protein coding genes. To represent the genes implicated via common variants, we used EA GWAS genes (N=1,722) prioritised by Lee *et al.* 2018. To represent the genes implicated via rare variants, we took 788 DDG2P genes in which a diagnosis has been found for any DDD proband with an NDC. **(A)** shows enrichment of genes that are expressed in prenatal or postnatal brain tissues according to Li *et al.*⁸ compared to all other autosomal genes. **(B)** shows enrichment of the indicated gene set within genes that show particularly high expression in the indicated cell types relative to other cell types in the prenatal brain. We used single cell RNA sequence data of prenatal brain samples from Li *et al.* to define genes showing particularly high expression in the given cell type (**Supplementary Methods**), as shown on the x-axis. Significant enrichment or depletion relative to all other genes is highlighted by colored asterisks. Black asterisks indicate cell types that showed stronger enrichment in DDG2P genes than EA GWAS genes, or *vice versa*. This was obtained by comparing enrichment estimates using two-sided z-score tests. **(C)** shows enrichment of heritability of educational attainment attributable to SNPs in or near genes that show particularly high expression in the indicated cell types relative to other cell types in the prenatal brain. We applied stratified LD score regression to GWAS summary statistics of educational attainment. Enrichment was estimated as the proportion of heritability explained by SNPs in or near prenatal brain cell type-enriched genes, which is shown in **(D)** divided by the proportion of SNPs mapping to these regions. Error bars indicate 95% confidence intervals. Significant enrichment or significantly different enrichment that passed Bonferroni correction of 52 tests (26 gene sets and 2 target gene lists) is indicated by two asterisks, and nominally significant evidence is indicated by one asterisk. IPC: intermediate progenitor cells; NEPRGC: neural epithelial progenitor/radial glial lineage; ExN: excitatory neurons; InN: interneurons; NasN: nascent neurons; Astro: astroglial lineage; Oligo: oligodendrocytes; OPC: oligodendrocyte progenitor cells; Endo: endothelial cells.

Figure R2

Figure R2 (similar to Figure 2 in the manuscript). Genetic correlations between brain-related traits and conditions and neurodevelopmental conditions (NDCs) before (blue) and after (red) restricting to patients with ID/DD. Genetic correlation estimates were calculated using Linkage Disequilibrium Score Regression for the GWAS meta-analysis between DDD and GEL. There were 9,188 NDC patients who had intellectual disability or developmental delay (ID/DD), which were 92% of all NDC patients in the original GWAS meta-analysis. Error bars show 95% confidence intervals.

Figure R3

Figure R3 (compare to Supplementary Table 2 from the manuscript). Variance explained on the liability scale by polygenic scores before and after restricting to patients with ID/DD or seizures. We tested PGSs for educational attainment (EA), cognitive performance (CP), the non-cognitive component of educational attainment (NonCogEA), schizophrenia (SCZ), and rare neurodevelopmental conditions (NDCs). $PGS_{NDC,DDD}$ was constructed using a GWAS comparing all NDC patients in DDD to controls from UKHLS (i.e the same one as used in the original manuscript), and $PGS_{NDC-ID/DD,DDD}$ was constructed using a GWAS comparing NDC patients who had ID/DD (89%) to the same controls. These two PGSs were tested in GEL only. Performance of PGSs comparing GEL cases with controls is in purple, and orange indicates that comparing DDD with UKHLS. The plot shows results before and after restricting the cases in the target sample to those with ID/DD or those with seizures. Three PGSs are not significantly associated with NDC risk in the subgroup of patients with seizures, indicated by a cross. All the other associations are significant with the logistic regression P-value ranging from 0.0067 to 1.3×10^{-38} .

Figure R4

Figure R4. Difference in average polygenic scores between patients with and without a monogenic diagnosis, before and after restricting to patients with ID/DD or seizure. We compared the differences between 3,821 diagnosed and 6,345 undiagnosed NDC patients (dark pink). In Figure 3A, the average PGSs of these two groups are shown by blue and red squares, respectively. We also did the comparison in NDC patients with intellectual disability or developmental delay (ID/DD; light pink; N=3,570 diagnosed versus N=5,775 undiagnosed) and in NDC patients with seizure (grey; N=922 diagnosed versus N=1,340 undiagnosed). Significant differences are indicated by asterisks. Error bars indicate 95% confidence intervals. PGSs for NDCs were derived from GWASs in DDD CoreExome samples, so these two PGSs were tested in GEL and DDD Omni samples (N=1,130 and 2,750 diagnosed and undiagnosed NDC patients, respectively; N=1,082 and 2,628 diagnosed and undiagnosed patients with ID/DD, respectively; N=319 and 633 diagnosed and undiagnosed patients with seizures, respectively).

Figure R5

Figure R5 (compare to Figure 4A from the main manuscript). Polygenic transmission disequilibrium test (pTDT) in undiagnosed probands with unaffected parents before and after restricting to patients with ID/DD or seizure. We used trios (N=2,866, or 1,567 for testing $PGS_{NDC,DDD}$ and $PGS_{NDC-ID/DD,DDD}$) in which both parents were unaffected and the proband did not have a monogenic diagnosis (same plotted in Figure 4A). We ran pTDT in a subset of ID/DD trios (N=2,568, or 1,479 for testing $PGS_{NDC,DDD}$ and $PGS_{NDC-ID/DD,DDD}$) as well as in a subset of trios in which the proband had seizures (N=632, or 362 for testing $PGS_{NDC,DDD}$ and $PGS_{NDC-ID/DD,DDD}$). The mean pTDT deviation is the difference between the child's PGS and the mean parental PGS, in units of the SD of the latter. Error bars show 95% confidence intervals.

Figure R6

Figure R6. Regressions comparing undiagnosed probands with neurodevelopmental conditions to controls, with and without controlling for parental PGSs, before and after restricting to patients with ID/DD or seizure. The plot shows effect sizes of PGSs on case/control status obtained from a logistic regression, testing either the child's PGS alone (proband only model) amongst trio probands, or while additionally controlling for the parents' PGSs (trio model). We compared trios where the proband did not have a monogenic diagnosis and both parents were unaffected to control trios from GEL, ALSPAC, and MCS (top panel). We also compared trios affected by ID/DD (middle panel) or seizure (bottom panel) with the same control trios. Error bars indicate 95% confidence intervals.

Figure R7

Figure R7. Correlation between rare variant burden scores and polygenic scores in patients with neurodevelopmental conditions and their parents, before and after restricting to patients with ID/DD or seizure. Correlation coefficients between the number of inherited rare damaging coding (A) or synonymous variants (B; negative control) in constrained genes and polygenic scores within/between different sets of individuals. In blue are the correlations within probands with neurodevelopmental conditions regardless of diagnostic status whose parents are unaffected (i.e. the child's rare variant burden score, RVBS, with their own polygenic score, PGS), and in purple are the correlations within their parents. In orange is the cross-parental correlation i.e. one parent's RVBS correlated with the other parent's PGS. We calculated the correlations in trios with neurodevelopmental conditions from DDD and GEL (N=3,999, or 2,553 for $PGS_{NDC,DDD}$ and $PGS_{NDC-ID/DD,DDD}$), as well as in two subgroups of trios affected by ID/DD (N=3,643, or 2,422 for $PGS_{NDC,DDD}$ and $PGS_{NDC-ID/DD,DDD}$) or seizure (N=946, or 638 for $PGS_{NDC,DDD}$ and $PGS_{NDC-ID/DD,DDD}$). Note that both the RVBSs and PGSs have been corrected for 20 genetic principal components. Error bars represent 95% confidence intervals.

Figure R8

Figure R8. Distribution of PCs that showed at least one significant difference between cohorts. We used two-sided t-tests to compare differences in the first 20 genetic principal components (PCs) between all cohorts. Red estimates indicate average values and error bars indicate \pm one standard deviation. We included NDC patients and parents in DDD and GEL cohorts. GEL_ctrl indicates individuals in GEL that were used as controls in the GWAS. PC1 showed a significant difference in all pairwise comparisons except for GEL controls vs UKHLS comparison. Brackets in other PCs indicate significant differences that passed the Bonferroni correction of 120 (20 PCs * 6 cohorts).

Figure R9

Figure R9. Results from the trio model fitted in GEL before and after controlling for the Index of Multiple deprivation (IMD). Similar to Figure 5 from the paper. The plot shows effect sizes of PGSs on case/control status, testing either the child's PGS alone ("proband only") amongst trio probands, or while additionally controlling for the parents' PGSs (trio model) either with or without additionally controlling for IMD. These were obtained from a logistic regression comparing undiagnosed proband with neurodevelopmental conditions to controls from GEL (1196 cases versus 801 controls). Note that this analysis was restricted to GEL since it was the only cohort for which a quantitative measure of IMD was available.

Decrease in coefficients

Maternal NTC 24%
Paternal NTC 22%

Z test not significant ($p=0.71$);
beta will have to be >0.034 to achieve a
significant p given the current se

- child_PGS; proband only
- child_PGS
- ▲ mother_PGS
- ▲ father_PGS
- imd (The original, quantitative measurement)
- Maternal NTC 17%
- Paternal NTC 23%

Figure R10

Figure R10. The association between proband's PGS_{EA} and diagnostic status in DDD, with or without correcting for technical, clinical and prenatal factors, assessed via logistic regression. An odds ratio greater than 1 indicates that higher PGS_{EA} is associated with a higher chance of getting a monogenic diagnosis. Pink indicates results after correcting for each indicated factor individually. Dark purple indicates associations obtained after correcting for multiple factors in a joint model including either trio status and prematurity or all factors.

Figure R11

Figure R11. Variance explained in risk of neurodevelopmental conditions (NDCs) by polygenic scores for relevant traits. Variance explained was calculated on the liability scale assuming a population prevalence of 1%. We compared DDD patients with control individuals from the UKHLS cohort (orange), and GEL patients with control individuals from GEL (purple). In addition to the individual PGSs for NDCs and relevant traits, we calculated composite PGS combining PGS_{EA} , PGS_{CP} , and PGS_{SCZ} . Weights were calculated in one cohort and tested in the other. In GEL, we also calculated a PGS combining PGS_{EA} , PGS_{CP} , PGS_{SCZ} , and $PGS_{NDC,DDD}$. We could not use DDD samples to train the weights for this score because $PGS_{NDC,DDD}$ was constructed from the DDD-derived GWAS; instead we used five-fold cross validation to estimate the variance explained in GEL.

Figure R12

Figure R12. Difference in average individual and composite polygenic scores between patients with versus without a monogenic diagnosis. We compared the differences between 3,821 diagnosed and 6,345 undiagnosed NDC patients from DDD and GEL combined. In Figure 3A, the average scores of individual PGSs of these two groups are shown by blue and red squares, respectively. In addition, we used a PGS combining PGS_{EA} , PGS_{CP} , and PGS_{SCZ} , and a higher score indicates lower risk for NDCs. We showed results of a DDD-trained composite PGS in GEL, a GEL-trained composite PGS in DDD, and a meta-analysis of the two cohorts. Significant differences are indicated by asterisks. Error bars indicate 95% confidence intervals.

Figure R13

Figure R13. Polygenic transmission disequilibrium test (pTDT) in undiagnosed probands with unaffected parents using a combined PGS. We used trios (N=2,866, or 1,567 for testing PGS_{NDC,DDD}) in which both parents were unaffected and the proband did not have a monogenic diagnosis. In addition to individual PGSs (same plotted in Figure 4A), we constructed a PGS combining PGS_{EA}, PGS_{CP}, and PGS_{SCZ}. We showed results of a DDD-trained composite PGS in GEL, a GEL-trained composite PGS in DDD, and a meta-analysis of the two cohorts. The mean pTDT deviation is the difference between the child's PGS and the mean parental PGS, in units of the SD of the latter. Error bars show 95% confidence intervals.

Figure R14

Figure R14. Regressions comparing undiagnosed probands with neurodevelopmental conditions to controls, with and without controlling for parental PGSs. The plot shows effect sizes of PGSs on case/control status obtained from a logistic regression, testing either the child's PGS alone (proband only model) amongst trio probands, or while additionally controlling for the parents' PGSs (trio model). We compared trios where the proband did not have a monogenic diagnosis and both parents were unaffected to control trios from GEL, ALSPAC, and MCS. In addition to individual PGSs (same plotted in Figure 5), we make a composite PGS combining PGS_{EA}, PGS_{CP}, and PGS_{SCZ}, and a lower score indicates higher risk for NDCs. We showed results of a DDD-trained composite PGS in GEL, a GEL-trained composite PGS in DDD, and a meta-analysis of the two cohorts. Error bars indicate 95% confidence intervals.

Figure R15

Figure R15. Correlation between rare variant burden scores and polygenic scores in patients with neurodevelopmental conditions and their parents. Correlation coefficients between the number of inherited rare damaging coding or synonymous variants (a negative control) in constrained genes and polygenic scores within/between different sets of individuals. In addition to individual PGS (same plotted in Figure 6), we used a PGS combining PGS_{EA} , PGS_{CP} , and PGS_{SCZ} , and a lower score indicates higher risk for NDCs. In blue are the correlations within probands with neurodevelopmental conditions regardless of diagnostic status whose parents are unaffected (i.e. the child's rare variant burden score, RVBS, with their own polygenic score, PGS), and in purple are the correlations within their parents. In orange is the cross-parental correlation i.e. one parent's RVBS correlated with the other parent's PGS. We calculated the correlations in trios with neurodevelopmental conditions from DDD and GEL (N=3,999, or 2,553 for $PGS_{NDC,DDD}$ and $PGS_{NDC-ID/DD,DDD}$). Note that both the RVBSs and PGSs have been corrected for 20 genetic principal components. Error bars represent 95% confidence intervals.

References for the Response document

1. Howe, L. J. *et al.* Within-sibship genome-wide association analyses decrease bias in estimates of direct genetic effects. *Nat. Genet.* **54**, 581–592 (2022).
2. Lee, J. J. *et al.* Gene discovery and polygenic prediction from a 1.1-million-person GWAS of educational attainment. *Nat. Genet.* **50**, 1112 (2018).
3. Niemi, M. E. K. *et al.* Common genetic variants contribute to risk of rare severe neurodevelopmental disorders. *Nature* **562**, 268–271 (2018).
4. Fry, A. *et al.* Comparison of Sociodemographic and Health-Related Characteristics of UK Biobank Participants With Those of the General Population. *Am. J. Epidemiol.* **186**, 1026–1034 (2017).
5. Wright, C. F. *et al.* Assessing the Pathogenicity, Penetrance, and Expressivity of Putative Disease-Causing Variants in a Population Setting. *Am. J. Hum. Genet.* **104**, 275–286 (2019).
6. Minikel, E. V. *et al.* Ascertainment bias causes false signal of anticipation in genetic prion disease. *Am. J. Hum. Genet.* **95**, 371–382 (2014).
7. Kingdom, R., Beaumont, R. N., Wood, A. R., Weedon, M. N. & Wright, C. F. Genetic modifiers of rare variants in monogenic developmental disorder loci. *Nat. Genet.* **56**, 861–868 (2024).
8. Li, M. *et al.* Integrative functional genomic analysis of human brain development and neuropsychiatric risks. *Science* **362**, (2018).
9. Finucane, H. K. *et al.* Partitioning heritability by functional annotation using genome-wide association summary statistics. *Nat. Genet.* **47**, 1228–1235 (2015).
10. Nivard, M. G. *et al.* More than nature and nurture, indirect genetic effects on children's academic achievement are consequences of dynastic social processes. *Nat Hum Behav* **8**, 771–778 (2024).
11. Hill, W. D. *et al.* Molecular Genetic Contributions to Social Deprivation and Household

Income in UK Biobank. *Curr. Biol.* **26**, 3083–3089 (2016).

12. Chuong, M. *et al.* Methodological Considerations When Using Polygenic Scores to Explore Parent-Offspring Genetic Nurturing Effects. *bioRxiv* 2023.03.10.532118 (2023)
doi:10.1101/2023.03.10.532118.
13. Wright Caroline F. *et al.* Genomic Diagnosis of Rare Pediatric Disease in the United Kingdom and Ireland. *N. Engl. J. Med.* **388**, 1559–1571 (2023).

Referee #1 (Remarks to the Author):

Thank you again for asking me to review this manuscript from Huang, Martin and colleagues. I thank the authors for their extensive response to my comments on the previous review. I had a few key aspects that I thought needed further attention, thought, and analysis, and I am very happy that the authors have done exactly that in the revision. I am particularly impressed with the thoroughness and due diligence paid to assessing the validity of using PGS for "neurodevelopmental condition", a catch-all term for a heterogeneous condition, and I am surprised (perhaps, as are the authors) that it works well, even when conditioned for different criteria. I am happy that this was done. The next point on a detailed analysis and description of non-transmitted alleles was also well done. The other points regarding the mechanistic relevance and penetrance of common variants were also handled satisfactorily. I feel that that revisions made to reflect these analyses are important and clear.

I also read through the responses to other reviewers' comments, and find that the authors have responded appropriately.

We are glad to hear we addressed the comments appropriately. We thank the reviewer for their kind comments.

Referee #2 (Remarks to the Author):

The authors have thoroughly responded to each of my concerns from my previous review and made several substantial changes that improved the manuscript. Of note, the organization of the manuscript into aims (including a very clear schematic overview Figure 1) has made the contributions of the work much more clear and helped me as a reader to follow the reasoning behind each of the analyses carried out as part of this paper. I appreciate the time they have taken to do this.

Like the other reviewers, I also really appreciated the FAQ. It is carefully written and I expect that it will be useful to the broader public. If this work published, will the FAQ be provided online as Supplementary Materials, or do the authors plan to post it in some online repository? If possible, I recommend the former since that will likely increase its impact.

We thank the reviewer for their kind comments. We are pleased to hear that the manuscript has improved and the aims helped highlight the contributions. We have included FAQ in the Supplementary Information as suggested.

My remaining comments are all mostly-minor expositional issues.

1) The authors have made changes to the manuscript in several places to clarify that the association between the non-transmitted parental alleles and the child's outcome is not necessarily due to indirect genetic effects, but several places remain where it is not clear. For example, in the abstract, the authors say that "it is also unclear whether parents' polygenic

background contributes to their children's risk beyond the direct effect of variants transmitted to the child." This implies to me that the authors will be able to assess when the parents' polygenic background contributes to their children's risk, but ultimately they cannot. Or in Figure 1, the arrow from the untransmitted alleles to the child's phenotype is labeled as an "indirect genetic effect" when it should be characterized as the non-transmitted coefficient. Also, in line 1145 of the Online Methods, the authors state that θ_{NT} captures the effect of non-transmitted alleles due to genetic nurture. (They subsequently clarify later in the paragraph that other factors could be included, but it seems like they should be more upfront about it.) There are several other places through the text where it's not clear that the authors aren't estimating indirect effects, so it would be good to go through and clean those up.

We thank the reviewer for highlighting the texts that need clarification. We have now updated the following places:

- We have now emphasised testing for direct genetic effects in the abstract:
"It is also unclear whether polygenic background affects risk directly via alleles transmitted from parents to children, or whether indirect genetic effects mediated through the family environment also play a role."

While we acknowledge that we cannot infer indirect genetic effects, and try to make this clear elsewhere, including later in the abstract, we hope that the reviewer will agree that it is useful to mention 'indirect genetic effects' at this point to make clearer, by elimination, what we mean by 'direct genetic effects'. If they feel strongly, we can remove the second clause.

- In the outline figure (now Extended Data Figure 1 in order to reduce the length of the main manuscript), "indirect genetic effect" is listed as one of the potential reasons that explain the significant non-transmitted coefficient, along with assortative mating. (The arrow is intended to indicate causality whereas the dotted line alongside the 'assortative mating' point is intended to indicate correlation.) Thus, we have not changed the figure but have instead updated the figure legend to emphasise that we did not find evidence for indirect effects. Again, if the reviewer feels this is inadequate, we can change the figure.
- We have now updated the Methods section to make it clear that θ_{NT} captures also confounding effects:
" θ_{NT} indicates the effect of parental non-transmitted alleles, which capture both the indirect genetic effects and potential confounding factors"
- We modified the second last paragraph in Introduction so that we avoid suggesting that the non-transmitted alleles necessarily have causal effects:
"One possible explanation for this is that variants associated with these traits have indirect genetic effects, i.e. they have some effect on the parents, and this then affects the offspring through the family or prenatal environment^{4,26-28}. However, confounding factors may also contribute to population-based genetic effect estimates^{4,29,30}. Studies of rare diseases have typically assumed implicitly that variants impacting risk have direct genetic effects on the affected individual. Given the genetic overlap with educational attainment and cognition, we

hypothesized that the common variants associated with risk of rare neurodevelopmental conditions might not only reflect direct genetic effects. ”

- *In the fifth paragraph of the Discussion, we changed “no significant evidence that prematurity mediates indirect genetic effects of common alleles” to “no significant evidence that prematurity mediates the effects of non-transmitted common parental alleles”*

2) In the section on "Exploring the role of prenatal factors", the authors conduct a Mendelian randomization study. They state "In theory, the genetic correlation between educational attainment and premature delivery could reflect a causal effect of lower educational attainment on premature birth, and/or a causal effect of premature birth on 427 lower educational attainment." They fail to note that a significant MR result could reflect that the SNPs used in the analysis affect some third factor which affects both premature birth and educational attainment. Also both associations could be driven by confounds like population stratification. Indeed, the authors point out that the traditional MR interpretation that educational attainment is affecting premature birth doesn't make sense since the education would have to take place after birth or after diagnosis of a neurodevelopmental condition, concluding that it is "more likely to reflect a causal effect of parents' educational attainment on their children's risk, consistent with the presence of indirect genetic effects." While this is technically true, this evidence is no more evidence of indirect effects than it is evidence of population stratification or other confounds. Ultimately, given that the assumptions of MR are clearly not satisfied, I don't know what is gained from the MR analyses beyond what is learned from the analyses in Supplementary Figure 5 where they show the association between the EA PGS and neurodevelopmental conditions is unchanged when controlling for prematurity or excluding probands who were born prematurely. I'd recommend removing the MR results entirely, but if they authors would like to keep it, they need to argue what additional information is gained from the MR analyses above the more interpretable results that don't imply causality.

We thank the reviewer for the suggestion. We totally agree with the reviewer that the assumptions underlying MR are probably not entirely valid, and there are potential caveats in interpreting the results. Following the reviewer's suggestion, we have removed the MR analysis and Extended Data Figure 9 from the manuscript.

Referee #3 (Remarks to the Author):

I am satisfied that they authors have address my points well and thank them for their careful analyses. I believe this paper will be a substantial addition to the literature.

We thank the reviewer for these kind comments.